

1 2
# Implementation of nitrogen cycle in the CLASSIC land model

## Ali Asaadi and Vivek. K. Arora

Canadian Centre for Climate Modelling and Analysis, Environment Canada, University of Victoria, Victoria, B.C., V8W 2Y2, Canada



**Abstract**

A terrestrial nitrogen (N) cycle model is coupled to carbon (C) cycle in the framework of the Canadian Land Surface Scheme Including biogeochemical Cycles (CLASSIC). CLASSIC currently models physical and biogeochemical processes and simulates fluxes of water, energy, and $CO_2$ at the land-atmosphere boundary. Similar to most models, gross primary productivity in CLASSIC increases in response to increasing atmospheric $CO_2$ concentration. In the current model version, a downregulation parameterization emulates the effect of nutrient constraints and scales down potential photosynthesis rates, using a globally constant scalar, as a function of increasing $CO_2$. In the new model when N and C cycles are coupled, cycling of N through the coupled soil-vegetation system facilitates the simulation of leaf N content and maximum carboxylation capacity ($V_{cmax}$) prognostically. An increase in atmospheric $CO_2$ decreases leaf N content, and therefore $V_{cmax}$, allowing the simulation of photosynthesis downregulation as a function of N supply. All primary N cycle processes, that represent the coupled soil-vegetation system, are modelled explicitly. These include biological N fixation, treatment of externally specified N deposition and fertilization application, uptake of N by plants, transfer of N to litter via litterfall, mineralization, immobilization, nitrification, ammonia volatilization, leaching, and the gaseous fluxes of NO, $N_2O$, and $N_2$. The interactions between terrestrial C and N cycles are evaluated by perturbing the coupled soil-vegetation system in CLASSIC with one forcing at a time over the 1850-2017 historical period. These forcings include the increase in atmospheric $CO_2$, change in climate, increase in N deposition, and increasing crop area and fertilizer input, over the historical period. The model response to these forcings is consistent with conceptual understanding of the coupled C and N cycles. The simulated terrestrial carbon sink over the 1959-2017 period, from the simulation with all forcings, is 2.0 Pg C/yr and compares reasonably well with the quasi observation-based estimate from the 2019 Global Carbon Project (2.1 Pg C/yr). The contribution of increasing $CO_2$, climate change, and N deposition to carbon uptake by land over the historical period (1850-2017) is calculated to be 84%, 2%, and 14%, respectively.



## 1. Introduction

The uptake of carbon (C) by land and ocean in response to the increase in anthropogenic fossil fuel emissions of $CO_2$ has served to slow down the growth rate of atmospheric $CO_2$ since the start of the industrial revolution. At present, about 55% of total carbon emitted into the atmosphere is taken up by land and ocean (Le Quéré et al., 2018; Friedlingstein et al., 2019). It is of great policy, societal, and scientific relevance whether land and ocean will continue to provide this ecosystem service. Over land, the uptake of carbon in response to increasing anthropogenic $CO_2$ emissions is driven by two primary factors, 1) the $CO_2$ fertilization of the terrestrial biosphere, and 2) the increase in temperature, both of which are associated with increasing $[CO_2]$. The $CO_2$ fertilization effect increases photosynthesis rates for about 80% of the world's $C_3$ vegetation since photosynthesis for plants that use the $C_3$ photosynthetic pathway is currently limited by $[CO_2]$ (Still et al., 2003; Zhu et al., 2016). The remaining 20% of vegetation uses the $C_4$ photosynthetic pathway that is much less sensitive to $[CO_2]$. Warming increases carbon uptake by vegetation in mid-high latitude regions where growth is currently limited by low temperatures (Zeng et al., 2011).

Even when atmospheric $CO_2$ is not limiting for photosynthesis, and near surface air temperature is optimal, vegetation cannot photosynthesize at its maximum possible rate if available water and nutrients (most importantly nitrogen (N) and phosphorus (P)) constrain photosynthesis (Vitousek and Howarth, 1991; Reich et al., 2006b). In the absence of water and nutrients, photosynthesis simply cannot occur. N is a major component of chlorophyll (the compound through which plants photosynthesize) and amino acids (that are the building blocks of proteins). The constraint imposed by available water and nutrients implies that the carbon



uptake by land over the historical period in response to increasing [$CO_2$] is lower than what it
would have been if water and nutrients were not limiting. This lower than maximum theoretically
possible rate of increase of photosynthesis in response to increasing atmospheric $CO_2$ is referred
to as downregulation (Faria et al., 1996; Sanz-Sáez et al., 2010). Typically, however, the term
downregulation of photosynthesis is used only in the context of nutrients and not water. McGuire
et al. (1995) define downregulation as a decrease in photosynthetic capacity of plants grown at
elevated $CO_2$ in comparison to plants grown at baseline $CO_2$, although the rate of photosynthesis
for plants grown and measured at elevated $CO_2$ is still higher than the rate for plants grown and
measured at baseline $CO_2$.

Earth system models (ESMs) that explicitly represent coupling of the global carbon cycle

and physical climate system processes are the only tools available at present that, in a physically
consistent way, are able to project how land and ocean carbon cycles will respond to future
changes in [$CO_2$]. Such models are routinely compared to one another under the auspices of the
Coupled Model Intercomparison Project (CMIP) every 6-7 years. The most recent and sixth phase
of CMIP (CMIP6) is currently underway (Eyring et al., 2016). Interactions between carbon cycle
and climate in ESMs have been compared under the umbrella of the Coupled Climate-Carbon
Cycle Model Intercomparison Project ($C^4$MIP) (Jones et al., 2016) which is an approved MIP of
the CMIP. Comparison of land and ocean carbon uptake in $C^4$MIP studies (Friedlingstein et al.,
2006; Arora et al., 2013, 2019) indicate that the future land carbon uptake across ESMs varies
widely and more than three times as much for the ocean carbon uptake. The reason for widely
varying estimates of future land carbon uptake across models is that our understanding of
biological processes that determine land carbon uptake is much less advanced than the physical



processes which primarily determine carbon uptake over the ocean. In the current generation of
terrestrial ecosystem models, other than photosynthesis for which a theoretical framework
exists, almost all of the other biological processes are represented on the basis of empirical
observations and parameterized in one way or another. In addition, not all models include N and
P cycles.  In the absence of an explicit representation of nutrient constraints on photosynthesis,
land models in ESMs parameterize downregulation of photosynthesis in other ways that reduce
the rate of increase of photosynthesis to values below its theoretically maximum possible rate,
as [$CO_2$] increases (e.g. Arora et al., 2009). Comparison of models across 5th and 6th phase of CMIP
shows that the fraction of models with land N cycle is increasing (Arora et al., 2013, 2019). The
nutrient constraints on photosynthesis are well recognized (Vitousek and Howarth, 1991; Arneth
et al., 2010). Terrestrial carbon cycle models neglect of nutrient limitation on photosynthesis has
been questioned from an ecological perspective (Reich et al., 2006a) and it has been argued that
without nutrients constraints these models will overestimate future land carbon uptake (Hungate
et al., 2003). Since in the real world photosynthesis downregulation does indeed occur due to
nutrient constraints, it may be argued that more confidence can be placed in future projections
of models that explicitly model the interactions between the terrestrial C and N cycles rather
than parameterize it in some other way.

Here, we present the implementation of N cycle in the Canadian Land Surface Scheme

Including biogeochemical Cycles (CLASSIC) model, which serves as the land component in the
family of Canadian Earth System Models (Arora et al., 2009, 2011; Swart et al., 2019). Section 2
briefly describes existing physical and carbon cycle components and processes of the CLASSIC
model. The conceptual basis of the new N cycle model and its parameterizations are described





in Section 3 and in the appendix. Section 4 outlines the methodology and data sets that we have
used to perform various simulations over the 1850-2017 historical period to assess the realism
of the coupled C and N cycles in CLASSIC in response to various forcings. Results from these
simulations over the historical period are presented in Section 5 and finally discussion and
conclusions are presented in Section 6.
**2. The CLASSIC land model**
**2.1 The physical and carbon biogeochemical processes**

The CLASSIC model is the successor to, and based on, the coupled Canadian Land Surface

Scheme (CLASS; Verseghy, 1991; Verseghy et al., 1993) and Canadian Terrestrial Ecosystem
Model (CTEM; Arora and Boer, 2005; Melton and Arora, 2016). CLASS and CTEM model physical
and biogeochemical processes in CLASSIC, respectively. Both CLASS and CTEM have a long history
of development as described in Melton et al. (2019) who also provide an overview of the CLASSIC
land model and describe its new technical developments that launched CLASSIC as a community
model. CLASSIC simulates land-atmosphere fluxes of water, energy, momentum, $CO_2$, and $CH_4$.
The CLASSIC model can be run at a point scale, e.g. using meteorological and geophysical data
from a FluxNet site, or over a spatial domain, that may be global or regional, using gridded data.
We briefly summarize the primary physical and carbon biogeochemical processes of CLASSIC here
that are relevant in the context of implementation of the N cycle in the model.
**2.1.1 Physical processes**

The physical processes of CLASSIC that simulate fluxes of water, energy and momentum,

based on CLASS, operate at a sub-daily time step. A time step of 30 minutes is typically used to



avoid numerical instabilities. Water, energy, and momentum fluxes are calculated over
vegetated, snow, and bare fractions, and the fractional vegetation cover is specified for each grid
cell. The vegetation is described in terms of four plant functional types (PFTs) in the operational
version of the model: needleleaf trees, broadleaf trees, crops, and grasses. The fractional
coverage of these four PFTs are either specified or may be dynamically simulated using
competition between PFTs, calculations for which are performed in the biogeochemical module
(CTEM). The structural attributes of vegetation are described by leaf area index (LAI), vegetation
height, canopy mass, and rooting depth and distribution that determine the fraction of roots in
each of the model's soil layers. These structural vegetation attributes may be specified or
simulated dynamically by the biogeochemical module of CLASSIC as a function of the driving
meteorological data and [$CO_2$]. The number of permeable soil and non-permeable bedrock layers
in CLASSIC can be varied depending on its application. The standard offline model that is driven
with reanalysis meteorological data, like in this study, uses 20 ground layers starting with 10
layers of 0.1 m thickness, gradually increasing to a 30 m thick layer for a total ground depth of
over 61 m. For application within the Canadian Earth system model currently three ground layers
with thicknesses of 0.1, 0.25 and 3.75 m are used. The depth to bedrock varies geographically
and is specified based on a soil depth data set. Above this depth, the layers are considered soil
and therefore permeable allowing movement of water between the layers for which liquid and
frozen soil moisture contents are determined prognostically. Below the permeable soil, the
bedrock rock layers are considered impermeable and therefore their soil moisture content is
zero. Soil and bedrock temperatures are found for each ground layer. CLASSIC also prognostically
models the temperature, mass, albedo, and density of a single layer snow pack (when the climate



permits snow to exist), the temperature and depth of ponded water on the soil, and the
temperature of the vegetation canopy. Interception and throughfall of rain and snow by the
canopy, and the subsequent unloading of snow, are also modelled. Energy and water balance of
each grid cell evolves independently and there is no lateral transfer of heat or moisture between
them.
**2.1.2 Biogeochemical processes**
The biogeochemical processes in CLASSIC are based on CTEM, and described in detail in
the appendix of Melton and Arora (2016). The biogeochemical component of CLASSIC simulates
the land-atmosphere exchange of $CO_2$ and while doing so simulates vegetation as a dynamic
component. The physics module (CLASS) provides the biogeochemical module (CTEM) with
physical land surface information including net radiation, and liquid and frozen soil moisture
contents of all the soil layers. The biogeochemical module of CLASSIC uses this information along
with air temperature to simulate photosynthesis and prognostically calculates amount of carbon
in the model's three live (leaves, stem, and root) and two dead (litter and soil) carbon pools.
Photosynthesis in CLASSIC is modelled at the same sub-daily time as the physical processes. The
remainder of the biogeochemical processes are modelled at a daily time step. These include: 1)
autotrophic and heterotrophic respirations from all the live and dead carbon pools, respectively,
2) allocation of photosynthate from leaves to stem and roots, 3) leaf phenology, 4) turnover of
live vegetation components that generates litter, 5) mortality, 6) land use change (LUC), 7) fire
(Arora and Melton, 2018), and 8) competition between PFTs for space (not switched on in this
study).






Figure 1 shows the existing structure of CLASSIC's carbon pools along with the addition
of non-structural carbohydrate pools for each of the model's live vegetation components. The
non-structural pools are not yet represented in the current operational version of CLASSIC
(Melton et al., 2019). The addition of non-structural carbohydrate pools is explained in Asaadi et
al. (2018) and helps improve leaf phenology for cold deciduous tree PFTs. The N cycle model
presented here is built on the research version of CLASSIC that consists of non-structural and
structural carbon pools for the leaves (L), stem (S), and root (R) components and the two dead
carbon pools in litter or detritus (D) and soil or humus (H) (Figure 1). We briefly describe these
carbon pools and fluxes between them, since N cycle pools and fluxes are closely tied to carbon
pools and fluxes. The gross primary productivity (GPP) flux enters the leaves from the
atmosphere. This non-structural photosynthate is allocated between leaves, stem, and roots. The
non-structural carbon then moves into the structural carbohydrates pool. Once this conversion
occurs structural carbon cannot be converted back to non-structural labile carbon. The model
attempts to maintain a minimum fraction of non-structural to total carbon in each component of
about 0.05 (Asaadi et al., 2018). Non-structural carbon is moved from stem and root components
to leaves, at the time of leaf onset for deciduous PFTs, and this is termed reallocation. The
movement of non-structural carbon is indicated by red arrows. Maintenance and growth
respiration (indicated by subscript $m$ and $g$ in Figure 1), which together constitute autotrophic
respiration, occur from the non-structural components of the three live vegetation components.
Litterfall from the structural and non-structural components of the vegetation components
contributes to the litter pool. Leaf litterfall is generated due to normal turnover of leaves as well



as cold and drought stress, and reduction in day length. Stem and root litter is generated due to
their turnover based on their specified life spans. Heterotrophic respiration occurs from the litter
and soil carbon pools depending on soil moisture and temperature, and humified litter is moved
from litter to the soil carbon pool.

All these terrestrial ecosystem processes and the amount of carbon in the live and dead

carbon pools are modelled explicitly for nine PFTs that map directly onto the four base PFTs used
in the physics module of CLASSIC. Needleleaf trees are divided into their deciduous and
evergreen phenotypes, broadleaf trees are divided into cold deciduous, drought deciduous, and
evergreen phenotypes, and crops and grasses are divided based on their photosynthetic
pathways into $C_3$ and $C_4$ versions. The sub-division of PFTs is required for modelling
biogeochemical processes. For instance, simulating leaf phenology requires the distinction
between evergreen and deciduous phenotypes of needleleaf and broadleaf trees. However, once
LAI is known, a physical process (such as the interception of rain and snow by canopy leaves) does
not need to know the underlying evergreen or deciduous nature of leaves.

The prognostically determined biomasses in leaves, stem, and roots are used to calculate

structural vegetation attributes that are required by the physics module. Leaf biomass is used to
calculate LAI using PFT-dependent specific leaf area. Stem biomass is used to calculate vegetation
height for tree and crop PFTs, and LAI is used to calculate vegetation height for grasses. Finally,
root biomass is used to calculate rooting depth and distribution which determines the fraction of
roots in each soil layer. Other than these structural vegetation attributes the biogeochemical
module also calculates canopy resistance (in conjunction with photosynthesis) that is used by the
physics module in calculating transpiration.



The approach for calculating photosynthesis in CLASSIC is based on the standard Farquhar
et al. (1980) model for $C_3$ photosynthetic pathway, and Collatz et al. (1992) for the $C_4$
photosynthetic pathway and presented in detail in Arora (2003). The model calculates gross
photosynthesis rate that is co-limited by the photosynthetic enzyme Rubisco, by the amount of
available light, and by the capacity to transport photosynthetic products for $C_3$ plants or the $CO_2$-
limited capacity for $C_4$ plants. In the real world, the maximum Rubsico limited rate ($V_{cmax}$) depends
on the leaf N content since photosynthetic capacity and leaf N are strongly correlated (Evans,
1989; Field and Mooney, 1986; Garnier et al., 1999). In the current operational version of
CLASSIC, the N cycle is not represented and the PFT-dependent values of $V_{cmax}$ are therefore
specified based on Kattge et al. (2009) who compile $V_{cmax}$ values using observation-based data
from more than 700 measurements. Along with available light, and the capacity to transport
photosynthetic products, the GPP in the model is strongly determined by specified PFT-
dependent values of $V_{cmax.}$ Also, in the current CLASSIC version a parameterization of
photosynthesis downregulation is included which, in the absence of the N cycle, implicitly
attempts to simulate the effects of nutrient constraints. This parameterization is explained in
detail in Arora et al. (2009) and briefly summarized here.
Following earlier simpler approaches (Cao et al., 2001; Alexandrov and Oikawa, 2002),
GPP can be expressed as a logarithmic function of [$CO_2$]

$$G_p(t) = G_0 \left( 1 + \gamma_p \ln \frac{c(t)}{c_0} \right) \qquad (1)$$

where the unconstrained or potential GPP at any given time, $G_p(t)$, is a function of its initial value
$G_0$, [$CO_2$] at time $t$, $c(t)$, and its initial value $c_0$. The rate of increase of GPP is determined by the





parameter $\gamma_p$  (where p indicates the "potential" rate of increase of GPP with [$CO_2$]). The
parameter $\gamma_p$ is calculated by fitting equation (1) to simulated GPP over the historical period. In
the absence of any nutrient constraints, the rate of increase of carbon uptake per unit area of
leaves is determined by the theoretical framework of Farquhar et al. (1980) and Collatz et al.
(1992) for $C_3$ and $C_4$ photosynthetic pathways, respectively. The rate of increase of global GPP,
however, also depends on how the model simulated LAI increases in response to increasing [$CO_2$],
which in turn depends on how photosynthate is allocated between leaves, stem, and root. Arora
et al. (2009) compared the unconstrained simulated rate of increase of GPP per unit increase in
[$CO_2$] (their Figure 3) with that based on the theoretical framework to show that the model's
response to increasing [$CO_2$] over the historical period is consistent with the theoretical
framework, given specified time-independent $V_{cmax}$ values for different PFTs. To parameterize
downregulation of photosynthesis with increasing [$CO_2$] for emulating nutrient constraints, the
unconstrained or potential GPP (for each time step and each PFT in a grid cell) is multiplied by
the global scalar $\xi(c)$

$$G = \xi(c)\ G_p \tag{2}$$

$$\xi(c) = \frac{1+\gamma_d \ln(c/c_0)}{1+\gamma_p \ln(c/c_0)} \tag{3}$$

where $t$ is omitted for clarity and the parameter $\gamma_d$ represents the downregulated rate of
increase of GPP with [$CO_2$] (indicated by the subscript $d$). When $\gamma_d < \gamma_p$ the modelled gross
primary productivity ($G$) increases in response to [$CO_2$] at a rate determined by the value of $\gamma_d$.
In the absence of the N cycle, the term $\xi(c)$ thus emulates down-regulation of photosynthesis as
$CO_2$ increases. For example, values of $\gamma_d$=0.42 and $\gamma_p$=0.90, from  Arora et al. (2009), yield a value



of $\xi(c)$ = 0.94 (indicating a 6% downregulation) for $c$=390 ppm (corresponding to year 2010) and
$c_0$=285 ppm.

Note that while the original model version does not include a N cycle, it is capable of

simulating realistic geographical distribution of GPP that partly comes from specification of
observation-based $V_{cmax}$ rates (which implicitly take into account C and N interactions in a non-
dynamic way) but more so the fact that the geographical distribution of GPP (and therefore net
primary productivity, NPP), to the first order, depends on climate. The Miami NPP model, for
instance, is able to simulate the geographical distribution of NPP using only mean annual
temperature and precipitation (Leith, 1975) since both the C and N cycles are governed primarily
by climate. The current version of CLASSIC is also able to reasonably simulate the terrestrial C
sink over the second half of the 20$^{th}$ century and early 21$^{st}$ century. CLASSIC (with the CLASS-
CTEM name) has regularly contributed to the annual Trends in Net Land–Atmosphere Carbon
Exchange (TRENDY) model intercomparison since 2016 which contributes results to the Global
Carbon Project's annual assessments – the most recent one being Friedlingstein et al. (2019).
What is then the purpose of coupling C and N cycles?

**3. Implementation of the N cycle in CLASSIC**

The primary objective of implementation of the N cycle is to model $V_{cmax}$ as a function of

leaf N content so as to make the use of multiplier $\xi(c)$ obsolete in the model, and allow to project
future carbon uptake that is constrained by available N. Modelling of leaf N content as a
prognostic variable, however, requires modelling the full N cycle over land. N enters the soil in



the inorganic mineral form through biological fixation of N, fertilizer application, and atmospheric
N deposition in the form of ammonium and nitrate. N cycling through plants implies uptake of
inorganic mineral N by plants, its return to soil through litter generation in the organic form, and
its conversion back to mineral form during decomposition of organic matter in litter and soil.
Finally, N leaves the coupled soil-vegetation system through leaching in runoff and through
various gaseous forms to the atmosphere. This section describes how these processes are
implemented and parameterized in the CLASSIC modelling framework. While the first order
interactions between C and N cycles are described well by the current climate, their temporal
dynamics over time require to explicitly model these processes.

Globally, terrestrial N cycle processes are even less constrained than the C cycle

processes. As a result, the model structure and parameterizations are based on conceptual
understanding and mostly empirical observations of N cycle related biological processes. We
attempt to achieve balance between a parsimonious and simple model structure and the ability
to represent the primary feedbacks and interactions between different model components.
**3.1 Model structure, and N pools and fluxes**

N is associated with each of the model's five live vegetation components and the two

dead carbon pools (shown in Figure 1). In addition, separate mineral pools of ammonium ($NH_4^+$)
and nitrate ($NO_3^-$) are considered. Figure 2 shows the C and N pools together in one graphic along
with the fluxes of N and C between various pools. These fluxes characterize the prognostic nature
of the pools as defined by the rate change equations below. The model structure allows the C:N
ratio of the live leaves ($C{:}N_L = C_L/N_L$), stem ($C{:}N_S = C_S/N_S$), and root ($C{:}N_R = C_R/N_R$)



components, and the dead litter (or debris) pool ($C:N_D = C_D/N_D$) to evolve prognostically. The
C:N ratio of soil organic matter ($C:N_H = C_H/N_H$), however, is assumed to be constant at 13
following Wania et al. (2012) (see also references therein) . The implications of this assumption
are discussed later.

The rates of change of N in the NH$_4^+$ and NO$_3^-$ pools (in gN m$^{-2}$), $N_{NH4}$ and $N_{NO3}$,

respectively, are given by
$$\frac{d\,N_{NH4}}{dt} = B_{NH4} + F_{NH4} + P_{NH4} + M_{D,NH4} + M_{H,NH4}$$

$$-U_{NH4} - (I_{NO3} + I_{N2O} + I_{NO}) - V_{NH3} - O_{NH4} \qquad (4)$$

$$\frac{d\,N_{NO3}}{dt} = P_{NO3} + I_{NO3} - L_{NO3} - U_{NO3} - (E_{N2} + E_{N2O} + E_{NO}) - O_{NO3} \qquad (5)$$

and all fluxes are represented in units of gN m$^{-2}$ day$^{-1}$. $B_{NH4}$  is the rate of biological N fixation
which solely contributes to the NH$_4^+$ pool, $F_{NH4}$ is the fertilizer input which is assumed to
contribute only to the NH$_4^+$ pool, and $P_{NH4}$ and $P_{NO3}$ are atmospheric deposition rates that
contribute to the NH$_4^+$ and NO$_3^-$ pools, respectively. Biological N fixation, fertilizer input, and
atmospheric deposition are the three routes through which N enters the coupled soil-vegetation
system. $M_{D,NH4}$ and $M_{H,NH4}$ are the mineralization flux from the litter and soil organic matter
pools, respectively, associated with their decomposition. We assume mineralization of humus
and litter pools only contributes to the NH$_4^+$ pool. $O_{NH4}$ and $O_{NO3}$  indicate immobilization of N
from the NH$_4^+$ and NO$_3^-$ pools, respectively, to the humus N pool which implies microbes (that
are not represented explicitly) are part of the humus pool. Combined together the terms
($M_{D,NH4} + M_{H,NH4} - O_{NH4} - O_{NO3}$) yield the net mineralization rate. $V_{NH3}$ is the rate of





ammonia (NH₃) volatilization and $L_{NO3}$ is the leaching of N that occurs only from the NO₃⁻ pool.
The positively charged ammonium ions are attracted to the negatively charged soil particles and
as a result it is primarily the negatively charged nitrate ions that leach through the soil (Porporato
et al., 2003; Xu-Ri and Prentice, 2008). $U_{NH4}$ and $U_{NO3}$ are uptakes of NH₄⁺ and NO₃⁻ by plants,
respectively. The nitrification flux from NH₄ to NO₃ pool is represented by $I_{NO3}$ which also results
in the release of the nitrous oxide (N₂O), a greenhouse gas, and nitric oxide (NO) through nitrifier
denitrification represented by the terms $I_{N2O}$ and $I_{NO}$, respectively. Finally, $E_{N2}$, $E_{N2O}$, and $E_{NO}$
are the gaseous losses of N₂ (nitrogen gas), N₂O, and NO from the NO₃⁻ pool associated with
denitrification. N is thus lost through the soil-vegetation system via leaching in runoff and
through gaseous losses of $I_{N2O}$, $I_{NO}$, $E_{N2}$, $E_{N2O}$, $E_{NO}$, and $V_{NH3}$.
The structural and non-structural N pools in root are written as $N_{R,S}$ and $N_{R,NS}$,
respectively, and similarly for stem ($N_{S,S}$ and $N_{S,NS}$) and leaves ($N_{L,S}$ and $N_{L,NS}$), and together the
structural and non-structural pools make the total N pool in leaf ($N_L = N_{L,S} + N_{L,NS}$), root ($N_R = $
$N_{R,S} + N_{R,NS}$), and stem ($N_S = N_{S,S} + N_{S,NS}$) components. The rate change equation for
structural and non-structural N pools in root are given by
$$\frac{d\,N_{R,NS}}{dt} = U_{NH4} + U_{NO3} + R_{L2R} - R_{R2L} - A_{R2L} - A_{R2S} - LF_{R,NS} - T_{R,NS2S} \qquad (6)$$
$$\frac{d\,N_{R,S}}{dt} = T_{R,NS2S} - LF_{R,S} \qquad (7)$$
Similar to the uptake of carbon by leaves and its subsequent allocation to root and stem
components, N is taken up by roots and then allocated to leaves and stem. $A_{R2L}$ and $A_{R2S}$
represent the allocation of N from roots to leaves and stem, respectively. The terms $R_{L2R}$ and
$R_{R2L}$ represent the reallocation of N between the non-structural components of root and leaves.





$R_{L2R}$ is the N reallocated from leaves to root representing resorption of a fraction of leaf N during
leaf fall for deciduous tree PFTs. $R_{R2L}$ indicates reallocation of N from roots to leaves (termed
reallocation in Figure 2) at the time of leaf-out for deciduous tree PFTs. At times other than leaf-
out and leaf-fall and for other PFTs these two terms are zero. $T_{R,NS2S}$ is the one way transfer of
N from the non-structural to the structural root pool, and similar to the carbon pools, once N is
converted to its structural form it cannot be converted back to its non-structural form. Finally,
the litterfall due to turnover of roots occurs from both the structural ($LF_{R,S}$) and non-structural
($LF_{R,NS}$) N pools.
The rate change equations for non-structural and structural components of leaves are
written as
$$\frac{d\,N_{L,NS}}{dt} = A_{R2L} - R_{L2R} - R_{L2S} + R_{R2L} + R_{S2L} - LF_{L,NS} - T_{L,NS2S} \tag{8}$$

$$\frac{d\,N_{L,S}}{dt} = T_{L,NS2S} - LF_{L,S} \tag{9}$$

where $T_{L,NS2S}$ is the one way transfer of N from the non-structural leaf component to its
structural N pool and $R_{S2L}$ indicates reallocation of N from stem to leaves (similar to $R_{R2L}$) at the
time of leaf out for deciduous tree PFTs. Litterfall occurs from both the structural ($LF_{L,S}$) and non-
structural ($LF_{L,NS}$) N pools of leaves, and all other terms have been previously defined.
Finally, the rate change equations for non-structural and structural components of stem
are written as
$$\frac{d\,N_{S,NS}}{dt} = A_{R2S} + R_{L2S} - R_{S2L} - LF_{S,NS} - T_{S,NS2S} \tag{10}$$





$$\frac{d\,N_{S,S}}{dt} = T_{S,NS2S} - LF_{S,S} \tag{11}$$

where $LF_{S,NS}$ and $LF_{S,S}$ represent stem litter from the non-structural and structural components,
$T_{S,NS2S}$ is the one way transfer of N from the non-structural stem component to its structural N
pool. All other terms have been previously defined.

Adding equations (6) through (11) yields rate of change of N in the entire vegetation pool

($N_V$) as
$$\frac{d\,N_V}{dt} = \frac{d\,N_{R,NS}}{dt} + \frac{d\,N_{R,S}}{dt} + \frac{d\,N_{L,NS}}{dt} + \frac{d\,N_{L,S}}{dt} + \frac{d\,N_{S,NS}}{dt} + \frac{d\,N_{S,S}}{dt} = \frac{d\,N_R}{dt} + \frac{d\,N_L}{dt} + \frac{d\,N_S}{dt}$$

$$\frac{d\,N_V}{dt} = U_{NH4} + U_{NO3} - LF_{R,NS} - LF_{R,S} - LF_{L,NS} - LF_{L,S} - LF_{S,NS} - LF_{S,S}$$

$$= U_{NH4} + U_{NO3} - LF_R \qquad\quad - LF_L \qquad\qquad - LF_S \tag{12}$$

which indicates how the dynamically varying vegetation N pool is governed by mineral N uptake
from the $NH_4^+$ and $NO_3^-$ pools and litterfall from the structural and non-structural components of
the leaves, stem, and root pools. $LF_R$ is the total N litter generation from the root pool and sum
of litter generation from its structural and non-structural components ($LF_R = LF_{R,S} + LF_{R,NS}$),
and similarly for the leaves ($LF_L$) and the stem ($LF_S$) pools.

The rate change equations for the organic N pools in the litter ($N_D$) and soil ($N_H$) pools

are written as follows.
$$\frac{d\,N_D}{dt} = LF_R + LF_L + LF_S - H_{N,D2H} - M_{D,NH4} \tag{13}$$

$$\frac{d\,N_H}{dt} = H_{N,D2H} + O_{NH4} + O_{NO3} - M_{H,NH4} \tag{14}$$



where $H_{N,D2H}$ is the transfer of humidified organic matter from litter to the soil organic matter
pool, and all other terms have been previously defined.
Sections A.1, A.2, and A.3 in the appendix describe how the individual terms of the rate
change equations of the 10 prognostic N pools (equations 4 through 11, and equations 13 and
14) are specified or parameterized. The treatment of these terms are briefly described here.
Biological N fixation (BNF, $B_{NH4}$) is parameterized as a function of soil moisture and temperature
with higher fixation rate per unit area for agricultural areas than natural vegetation. If externally
specified information for ammonium ($NH_4^+$) and nitrate ($NO_3^-$) deposition rates is available then
it is used otherwise deposition is assumed to be split equally between $NH_4^+$ and $NO_3^-$. Externally
specified fertilizer application rates are same throughout the year in the tropics (between 30°S
and 30°N), given multiple crop rotations in a given year in tropical regions. Between 30° and 90°
latitudes in both northern and southern hemispheres, we assume that fertilizer application starts
on the spring equinox and ends on the fall equinox. Plant N demand is calculated on the basis of
the fraction of NPP allocated to leaves, stem, and root components and their specified minimum
PFT-dependent C:N ratios. Both passive and active root uptakes of N are modelled. Passive
uptake depends on transpiration and concentration of $NH_4^+$ and $NO_3^-$ in the root zone water
column. When passive N uptake cannot meet the N demand, active uptake compensates for
reduced passive uptake though eventually they both depend on the amount of available N in the
mineral pools. Plant N uptake by roots is allocated to stem and leaf components, which allows to
model leaf N content ($N_L$) as a prognostic variable. N contributions to litter through litterfall are
based on C:N ratios of the vegetation components and the litterfall rates. Resorption of N before
litterfall for deciduous tree species is also modelled. Decomposition of litter and soil organic



matter releases C to the atmosphere as $CO_2$ and the mineralized N is moved to the $NH_4^+$ pool.
Immobilization of mineral N from $NH_4^+$ and $NO_3^-$ pools into the soil organic matter pool is meant
to keep the soil organic matter C:N ratio ($C:N_H$) at its specified constant value of 13 for all PFTs.

Nitrification, the process converting ammonium to nitrate, is driven by microbial activity

and depends both on soil temperature and moisture such that it is constrained both at high and
low soil moisture contents. Gaseous fluxes of NO ($I_{NO}$) and $N_2O$ ($I_{N2O}$) are associated with
nitrification and assumed to be directly proportional to the nitrification flux. Denitrification is
modelled to reduce $NO_3^-$ to NO, $N_2O$, and ultimately to $N_2$. Unlike nitrification, however,
denitrification is primarily an anaerobic process and therefore occurs primarily when soil is
saturated. Leaching of $NO_3^-$ ($L_{NO3}$) is parameterized to be directly proportional to baseflow from
the bottommost soil layer and the size of the $NO_3^-$ pool. Finally, $NH_3$ volatilization ($V_{NH3}$) is
parametrized as a function of $NH_4^+$ pool size, soil temperature, soil pH, and aerodynamic and
boundary layer resistances.
**3.2 Coupling of C and N cycles**

As mentioned earlier in Section 2.1.2, the primary objective of coupling of C and N cycles

is to be able to simulate $V_{cmax}$ as a function of leaf N content ($N_L$) for each PFT. This coupling is
represented through the following relationship

$$V_{cmax} = \Gamma_1 N_L + \Gamma_2 \qquad\qquad (15)$$

where $\Gamma_1$ (13 µmol $CO_2$ $gN^{-1}$ $s^{-1}$) and $\Gamma_2$ (8.5 µmol $CO_2$ $m^{-2}$ $s^{-1}$) are global constants, except for
the broadleaf evergreen tree PFT for which a lower value of $\Gamma_1$ (5.1 µmol $CO_2$ $gN^{-1}$ $s^{-1}$) is used as
discussed below. A linear relationship between photosynthetic capacity and $N_L$ (Evans, 1989;



Field and Mooney, 1986; Garnier et al., 1999) and between photosynthetic capacity and leaf
chlorophyll content (Croft et al., 2017) is empirically observed. The modelled differences in PFT
specific values of $V_{cmax}$, in our framework, come through differences in simulated $N_L$ values that
depend on BNF, given that BNF is the primary natural source of N input into the coupled soil-
vegetation system. $N_L$ values, however, also depend on leaf phenology, allocation of carbon and
nitrogen, turnover rates, transpiration (which brings in N through passive uptake), and almost
every aspect of plant biogeochemistry which affects a PFT's net primary productivity and
therefore N demand. We have avoided using PFT-dependent values of $\Gamma_1$ and $\Gamma_2$ for easy
optimization of these parameter values but also because such an optimization can potentially
hide other model deficiencies. More importantly, using PFT-independent values of $\Gamma_1$ and $\Gamma_2$
yields a more elegant framework whose successful evaluation will provide confidence in the
overall model structure.
As shown later in the results section, using $\Gamma_1$ and $\Gamma_2$ as global constants yields GPP values
that are higher in the tropical region than an observation-based estimate. This is not surprising
since tropical regions are well known to be limited by P (Vitousek, 1984; Aragão et al., 2009;
Vitousek et al., 2010) and our framework currently doesn't model a P cycle explicitly. An
implication of productivity that is limited by P is that changes in $N_L$ are less important. In the
absence of explicit treatment of the P cycle, we therefore simply use a lower value of $\Gamma_1$ for the
broadleaf evergreen tree PFT which, in our modelling framework, exclusively represents a
tropical PFT. Although, a simple way to express P limitation, this approach yields the best
comparison with observation-based GPP, as shown later.





The second pathway of coupling between the C and N cycles occurs through
mineralization of litter and soil organic matter. During periods of higher temperature,
heterotrophic C respiration fluxes increase from the litter and soil organic matter pools and this
in turn implies an increased mineralization flux (via equation A14 in the appendix) leading to
more mineral N available for plants to uptake.
**4.0 Methodology**
**4.1 Model simulations and input data sets**
We perform CLASSIC model simulations with the N cycle for the pre-industrial period
followed by several simulations for the historical 1851-2017 period to evaluate the model's
response to different forcings, as summarized below. The simulation for the pre-industrial period
uses forcings that correspond to year 1850 and the model is run for thousands of years until its
C and N pools come into equilibrium. The pre-industrial simulation, therefore, yields the initial
conditions from which the historical simulations for the period 1851-2017 are launched.
For the historical period, the model is driven with time-varying forcings that include $CO_2$
concentration, population density (used by the fire module of the model for calculating
anthropogenic fire ignition and suppression), land cover, and meteorological data. In addition,
for the N cycle module, the model requires time-varying atmospheric N deposition and fertilizer
data. The atmospheric $CO_2$ and meteorological data (CRU-JRA) are same as those used for the
TRENDY model intercomparison project for terrestrial ecosystem models for year 2018 (Le Quéré
et al., 2018). The CRU-JRA meteorological data is based on 6-hourly Japanese Reanalysis (JRA)
adjusted for monthly values based on the Climate Research Unit (CRU) data and available for the



period 1901-2017. Since no meteorological data are available for the 1850-1900 period, we use
1901-1925 meteorological data repeatedly for this duration and also the pre-industrial spin up.
The assumption is that since there is no significant trend in the CRU-JRA data over this period,
these data can be reliably used to spin up the model to equilibrium. The land cover data used to
force the model are based on a geographical reconstruction of the historical land cover driven by
the increase in crop area following Arora and Boer (2010) but using the crop area data prepared
for the Global Carbon Project (GCP) 2018 following Hurtt et al. (2006). Since land cover is
prescribed, the competition between PFTs for space for the simulations reported here is switched
off. The population data for the period 1850-2017 are based on Klein Goldewijk et al. (2017) and
obtained from ftp://ftp.pbl.nl/../hyde/hyde3.2/baseline/zip/. The time-independent forcings
consist of soil texture and permeable depth data.

Time-varying atmospheric N deposition and fertilizer data used over the historical period

are also specified as per the TRENDY protocol. The fertilizer data are based on the $N_2O$ model
intercomparison project (NMIP) (Tian et al., 2018) and available for the period 1860-2014.  For
the period before 1860, 1860 fertilizer application rates are used. For the period after 2014,
fertilizer application rates for 2014 are used. Atmospheric N deposition data are from input4MIPs
(https://esgf-node.llnl.gov/search/input4mips/) and are the same as used by models
participating in CMIP6 for the historical period (1850-2014). For years 2015-2017 the N
deposition data corresponding to those from representative concentration pathway (RCP) 8.5
scenario are used.

To evaluate the model's response to various forcings over the historical period we

perform several simulations turning on one forcing at a time as summarized in Table 1. The



objective of these simulations is to see if the model response to individual forcings is consistent
with expectations. For example, in the CO2-only simulation only atmospheric $CO_2$ concentration
increases over the historical period, while all other forcings stay at their 1850 levels. In the N-
DEP-only simulation only N deposition increases over the historical period, and similarly for other
runs in Table 1. A "FULL" simulations with all forcings turned on is then also performed which we
compare to the original model without a N cycle which uses the photosynthesis downregulation
parameterization (termed "ORIGINAL" in Table 1).

Finally, a separate pre-industrial simulation is also performed that uses the same $\Gamma_1$ and

$\Gamma_2$ globally (FULL-no-implicit-P-limitation). This simulation is used to illustrate the effect of
neglecting P limitation for the broadleaf evergreen tree PFT in the tropics.
**4.2 Evaluation data sources**

We compare globally-summed annual values of N pools and fluxes with observations and

other models, and where available their geographical distribution and seasonality. In general,
however, much less observation-based data are available to evaluate simulated terrestrial N
cycle components than for C cycle components. As a result, N pools and fluxes are primarily
compared to results from both observation-based studies and other modelling studies
(Bouwman et al., 2013; Fowler et al., 2013; Galloway et al., 2004; Vitousek et al., 2013; Zaehle,
2013). Since the primary purpose of the N cycle in our framework is to constrain the C cycle, we
also compare globally-summed annual values of GPP and net atmosphere-land $CO_2$ flux, and their
zonal distribution with available observation-based and other estimates. The observation-based
estimate of GPP is from Beer et al. (2010), who apply diagnostic models to extrapolate ground-



based carbon flux tower observations from about 250 stations to the global scale. Observation-
based net atmosphere-land $CO_2$ flux is from Global Carbon Project's 2019 assessment
(Friedlingstein et al., 2019).
**5.0 Results**
**5.1 N inputs**

Figure 3, panel a, shows the global values of simulated BNF from the six primary

simulations summarized in Table 1. BNF stays at its pre-industrial value of around 80 Tg N $yr^{-1}$ in
the CO2-only, FIRE-only, and N-DEP-only simulations. In the CLIM-only (indicated by magenta
coloured line) and the FULL-no-LUC (blue line) simulations the change in climate, associated with
increases in temperature and precipitation over the 1901-2017 period (see Figure A1 in the
appendix), increases BNF to about 85 Tg N $yr^{-1}$. In our formulation (equation A1) BNF is positively
impacted by increases in temperature and precipitation. The values in parenthesis in Figure 3a
legend, and in subsequent panels of this and other figures, show average values over the 1850s,
the last 20 years (1998-2017) of the simulations, and the change between these two periods. In
the LUC+FERT-only simulation (dark green line) the increase in crop area contributes to an
increase in global BNF with a value around 110 Tg N $yr^{-1}$ for the present day, since a higher BNF
per unit crop area is assumed than for natural vegetation. Finally, in the FULL simulation (red line)
the 1998-2017 average value is around 117 Tg N $yr^{-1}$ due both to changes in climate over the
historical period and the increase in crop area. Our present day value of global BNF is broadly
consistent with other modelling and data-based studies as summarized is Table 2. Panels c and e
in Figure 3 show the decomposition of the total terrestrial BNF into its natural (over non-crop



PFTs) and anthropogenic (over $C_3$ and $C_4$ crop PFTs) components. The increase in crop area over
the historical period decreases natural BNF from its pre-industrial value of 59 to 54 Tg N yr$^{-1}$ for
the present day as seen for the LUC+FERT-only simulation (green line) in Figure 3c, while
anthropogenic BNF over agricultural area increases from 21 to 56 Tg N yr$^{-1}$ (Figure 3e). Figure 3c
and 3e show that the increase in BNF (Figure 3a) in the FULL simulation is caused primarily by an
increase in crop area. Our present day values of natural and anthropogenic BNF are also broadly
consistent with other modelling and data-based studies as summarized in Table 2.

Figure 3, panels b, d, and f, show the global values of externally specified fertilizer input,

and deposition of ammonium and nitrate, based on the TRENDY protocol, for the six primary
simulations. Ammonium and nitrate deposition, and fertilizer input stay at their pre-industrial
level for simulations in which these forcings do not increase over the historical period. As
mentioned earlier, N deposition is split evenly into ammonium and nitrate. The present day
values of fertilizer input and N deposition are consistent with other estimates available in the
literature (Table 2). The fertilizer input rate in the simulation with all forcings except land use
change (FULL-no-LUC, blue line), that is with no increase in crop area over its 1850 value, is 50 Tg
N yr$^{-1}$ compared to 91 Tg N yr$^{-1}$ in the FULL simulation, averaged over the 1998-2017 period. The
additional 41 Tg N yr$^{-1}$ of fertilizer input occurs in the FULL simulation due to the increase in crop
area but also due to the increasing fertilizer application rates over the historical period.

Figure 4 shows the geographical distribution of simulated BNF, and specified fertilizer

application and N deposition rates. The geographical distribution of BNF (Figure 4a) looks very
similar to the current distribution of vegetation (not shown) with warm and wet regions showing
higher values than cold and dry regions since BNF is parameterized as a function of temperature





and soil moisture. Figures 4c and 4e show the split of BNF into its natural and anthropogenic
components. Anthropogenic BNF only occurs in regions where crop area exists according to the
specified land cover and it exhibits higher values than natural BNF in some regions because of its
higher value per unit area (see section A.1.1 in the appendix). In Figure 4b, the fertilizer
application rates are concentrated in regions with crop area and with values as high as 16 gN m$^-$
$^2$ especially in eastern China.  The N deposition rates are more evenly distributed than fertilizer
applications rates, as would be expected, since emissions are transported downstream from their
point sources. Areas with high emissions like the eastern United States, India, eastern China, and
Europe, however, still stand out as areas that receive higher N deposition.
At the global scale, and for the present day, natural BNF (59 Tg N yr$^{-1}$) is overwhelmed by
anthropogenic sources: anthropogenic BNF (60 Tg N yr$^{-1}$), fertilizer input (91.7 Tg N yr$^{-1}$), and
atmospheric N deposition increase since the pre-industrial era (~45 Tg N yr$^{-1}$). Currently humanity
fixes more N than the natural processes  (Vitousek, 1994).
**5.2 C and N pools, fluxes response to historical changes in forcings**
To understand the model response to changes in various forcings over the historical
period we first look at the evolution of global values of primary C and N pools, and fluxes, shown
in Figures 5 through 9. Figure 5a shows the time evolution of global annual GPP values, the
primary flux of C into the land surface, for the six primary simulations, the ORIGINAL simulation
performed with the model version with no N cycle, and the ORIG-UNCONST simulation with no
photosynthesis downregulation (see Table 1). The unconstrained rate of increase in GPP (35.6 Pg
C yr$^{-1}$ over the historical period) in the ORIG-UNCONST simulation (dark cyan line) is governed by



the standard photosynthesis model equations following Farquhar et al. (1980) and Collatz et al.
(1992) for $C_3$ and $C_4$ plants, respectively. Downregulation of photosynthesis in the ORIGINAL
simulation (purple line) is modelled on the basis of equation (1), while in the FULL simulation (red
line) photosynthesis downregulation results from a decrease in $V_{cmax}$ values (Figure 6d) due to a
decrease in leaf N content (Figure 6b). We will compare the FULL and ORIGINAL simulations in
more detail later. The simulations with individual forcings, discussed below, provide insight into
the combined response of GPP to all forcings in the FULL simulation.

**5.2.1 Response to increasing $CO_2$**

The response of C and N cycles to increasing $CO_2$ in the CO2-only simulation (orange line

in Figure 5) is the most straightforward to interpret. A $CO_2$ increase causes GPP to increase by 7.5
Pg C $yr^{-1}$ over its pre-industrial value (Figure 5a), which in turn causes vegetation (Figure 5b), leaf
(Figure 5c), and soil (Figure 5d) carbon mass to increase as well. The vegetation and leaf N
amounts (orange line, Figures 6a and 6b), in contrast, decrease in response to increasing $CO_2$.
This is because N gets locked up in the soil organic matter pool (Figure 6c) in response to an
increase in the soil C mass (due to the increasing GPP), litter inputs which are now rich in C (due
to $CO_2$ fertilization) but poor in N (since N inputs are still at their pre-industrial level), and the fact
that the C:N ratio of the soil organic matter is fixed at 13. This response to elevated $CO_2$ which
leads to increased C and decreased N in vegetation is consistent with meta-analysis of 75 field
experiments of elevated $CO_2$ (Cotrufo et al., 1998). A decrease in N in leaves (orange line, Figure
6b) leads to a concomitant decrease in maximum carboxylation capacity ($V_{cmax}$) (orange line,
Figure 6d) and as a result GPP increases at a much slower rate in the CO2-only simulation than in





the ORIG-UNCONST simulation (Figure 5a). Due to the N accumulation in the soil organic matter
pool, the $NH_4^+$ and $NO_3^-$ (Figures 6e and 6f) pools also decrease in size in the CO2-only simulation.

Figure 7 shows the time series of N demand, plant N uptake and its split between passive

and active N uptakes. The plant N demand in the CO2-only simulation (Figure 7a, orange line)
increases from its pre-industrial value of 1512 Tg N/yr to 1639 Tg N/yr for the present day since
the increasing C input from increasing GPP requires higher N input to maintain preferred
minimum C:N ratio of plant tissues. However, since mineral N pools decrease in size over the
historical period (Figures 6e and 6f), the total plant N uptake (Figure 7b) reduces. Passive plant N
uptake is directly proportional to pool sizes of $NH_4^+$ and $NO_3^-$ and therefore it reduces in response
to increasing $CO_2$. Active plant N uptake, which compensates for insufficient passive N uptake
compared to the N demand, also eventually starts to decline as it also depends on mineral N pool
sizes. The eventual result of increased C supply and reduced N supply is an increase in the C:N
ratio of all plant components and litter (Figure 8).

Figure 9 shows the net mineralization flux (the net transfer of mineralized N from litter

and humus pools to the mineral N pools as a result of the decomposition of organic matter),
nitrification (N flux from $NH_4^+$ to the $NO_3^-$ pool), and the gaseous and leaching losses from the
mineral pools. The net mineralization flux reduces in the CO2-only simulation (Figure 9a, orange
line) as N gets locked up in the soil organic matter. A reduction in the $NH_4^+$ pool size implies a
reduction in the nitrification flux over the historical period (Figure 9b, orange line). Finally,
leaching from the $NO_3^-$ pool (Figure 9c), $NH_3$ volatilization (Figure 9d), and the gaseous losses
associated with nitrification from the $NH_4^+$ pool (Figure 9e) and denitrification from the $NO_3^-$ pool





(Figure 9f) all reduce in response to reduction in pool sizes of $NH_4+$ and $NO_3-$ in the CO2-only
simulation.

**5.2.2 Response to changing climate**

The perturbation due to climate change alone over the historical period in the CLIM-only
(magenta coloured lines in Figures 5-9) simulation is smaller than that due to increasing $CO_2$. In
Figure 5a, changes in climate over the historical period increase GPP slightly by 3.60 Pg C $yr^{-1}$
which in turn slightly increases vegetation (including leaf) C mass (Figure 5b,c). The litter and soil
carbon mass (Figure 5d), however, decrease slightly due to increased decomposition rates
associated with increasing temperature (see Figure A1). Both the increase in BNF due to
increasing temperature (magenta line in Figure 3a), and the reduction in litter and soil N mass
(Figure 6c) due to increasing decomposition and higher net N mineralization (Figure 9a, magenta
line), make more N available.  This results in a slight increase in vegetation and leaf N mass
(Figures 6a and 6b) and the $NH_4^+$ (Figure 6e) pool which is the primary mineral pool in soils under
vegetated regions. The global $NO_3^-$ pool, in contrast, decreases in the CLIM-only simulation
(Figure 6f) with the reduction primarily occurring in arid regions where the $NO_3^-$ amounts are very
large (see Figure 10b that shows the geographical distribution of the primary C and N pools). The
geographical distribution of $NH_4^+$ (Figure 10a) generally follows the geographical distribution of
BNF, but with higher values in areas where cropland exists and where N deposition is high. The
geographical distribution of $NO_3^-$ (Figure 10b) generally shows lower values than $NH_4^+$ except in
the desert regions where lack of denitrification leads to a large buildup of the $NO_3^-$ pool (see
section A3.2 in the appendix). Although Figure 10 shows the geographical distribution of mineral
N pools from the FULL simulation, the geographical distribution of pools are broadly similar





between different simulations with obvious differences such as lack of hot spots of N deposition
and fertilizer input in simulations in which these forcings stay at their pre-industrial levels. Figure
10 also shows the simulated geographical distribution of C and N pools in the vegetation and soil
organic matter. The increase in GPP due to changing climate increases the N demand (Figure 7a,
magenta line) but unlike the CO2-only simulation, the plant N uptake increases since the $NH_4^+$
and $NO_3^-$ pools increase in size over the vegetated area in response to increased BNF (Figure 3a,
magenta line). The increase in plant N uptake comes from the increase in passive plant N uptake
(Figure 7c) while the active plant N uptake reduces (Figure 7d). Active and passive plant N uptakes
are inversely correlated. This is by design since active plant N uptake increases when passive
plant N uptake reduces and vice-versa, although eventually both depend on the size of available
mineral N pools. Enhancement of plant N uptake due to changes in climate, despite increases in
GPP associated with a small increase in $V_{cmax}$ (Figure 6d), leads to a small reduction in the C:N
ratio of all plant tissues (Figure 8). The litter C:N, in contrast, shows a small increase since not all
N makes its way to the litter as a fraction of leaf N is resorbed from deciduous trees leaves prior
to leaf fall (Figure 8e). Finally, the small increase in pool sizes of $NH_4^+$ and $NO_3^-$ leads to a small
increase in leaching, volatilization, and gaseous losses associated with nitrification and
denitrification (Figure 9).
**5.2.3 Response to N deposition**

The simulated response of GPP to changes in N deposition (brown line) over the historical

period is smaller than that for $CO_2$ and climate (Figure 5a). The small increase in GPP of 2.0 Pg C
$yr^{-1}$ leads to commensurately small increases in vegetation (Figure 5b) and litter plus soil (Figure
5d) C mass.  Vegetation and leaf N mass (Figure 6a,b) also increase in response to N deposition



and so do mineral pools of $NH_4^+$ and $NO_3^-$ (Figure 6e,f). The increase in GPP in the simulation with
N deposition results from an increase in $V_{cmax}$ rates (Figure 6d) associated with an increase in leaf
N content (Figure 6b). N demand increases marginally and so does plant N uptake in response to
N deposition (Figure 7). As would be intuitively expected, the C:N ratio of the whole plant, its
components of leaves, stem, and root, and litter decreases slightly in response to N deposition
(Figure 8). Net N mineralization, nitrification, leaching, volatilization, and gaseous losses
associated with nitrification and denitrification all increase in response to N deposition (Figure

9).

**5.2.4 Response to LUC and fertilizer input**

The simulated response to LUC, which reflects an increase in crop area, and increased

fertilizer deposition rates over the historical period is shown by dark green lines in Figures 5
through 9. The increase in fertilizer input is a much bigger perturbation to the N cycle system
than N deposition. Figure 3 shows that at the global scale the fertilizer inputs increase from 0 to
~92 Tg N/yr over the historical period, while the combined $NH_4^+$ and $NO_3^-$ N deposition rate
increases from around 20 to 65 Tg N/yr. In addition, because of higher per unit area BNF rates
over crop area than natural vegetation, the increase in crop area in this simulation leads to an
increase in anthropogenic BNF from about 20 to 56 Tg N/yr over the historical period. All together
increasing crop area and fertilizer inputs imply an additional ~130 Tg N/yr being input into the
terrestrial N cycle at the present day since the pre-industrial period, compared to an increase of
only 45 Tg N/yr for the N deposition forcing.



The global increase in fertilizer input over the historical period leads to higher $NH_4^+$ and
$NO_3^-$ pools (Figures 6e and 6f). Although both fertilizer and BNF contribute to the $NH_4^+$ pool, the
$NO_3^-$ pool also increases through the nitrification flux (Figure 9b). An increase in crop area over
the historical period results in deforestation of natural vegetation that reduces vegetation
biomass but also soil carbon mass, since a higher soil decomposition rate is assumed over
cropland area (Figures 5b and 5d), consistent with empirical measurements (Wei et al., 2014).
Fertilizer application only occurs over crop areas which increases the $V_{cmax}$ rates for crops and, as
expected, this yields an increase in globally-averaged $V_{cmax}$ (Figure 6d). A corresponding large
increase in leaf N content (Figure 6b) is, however, not seen because vegetation (and therefore
leaf) N (Figure 6a,b) is also lost through deforestation. In addition, $V_{cmax}$ is a per unit area rate
that is averaged over the whole year while leaf and vegetation N pools are sampled at the end of
each year and all crops in the northern hemisphere above 30° N are harvested before the year
end. Vegetation N mass, in fact, decreases in conjunction with vegetation C mass (Figure 5b).
Plant N demand reduces (Figure 7a) and plant N uptake increases (Figure 7b) driven by crop PFTs
in response to fertilizer input, as would be intuitively expected.  The increase in plant N uptake
comes from the increase in passive N uptake, in response to increases in pool sizes of $NH_4^+$ and
$NO_3^-$ over crop areas, while active plant N uptake decreases since passive uptake can more than
keep up with the demand over cropland area. While the C:N ratio of vegetation biomass
decreases over cropland area in response to fertilizer input (not shown) this is not seen in the
globally-averaged C:N ratio of vegetation (Figure 8a) and its components because C and N are
also lost through deforestation and the fact that crop biomass is harvested. The C:N of the global
litter pool, however, decreases in response to litter from crops which gets rich in N as fertilizer





application rates increase. Finally, in Figure 9, global net N mineralization, nitrification, leaching,
volatilization, and gaseous losses associated with nitrification and denitrification all increase by
a large amount in response to an increase in fertilizer input.
**5.2.5 Response to all forcings**
We can now evaluate and understand the simulated response of the FULL simulation to
all forcings (red line in Figures 5 through 9). The increase in GPP in the FULL simulation (14.5 Pg
C/yr) in Figure 5a over the historical period is driven by GPP increase associated with increase in
$CO_2$ (7.5 Pg C/yr), changing climate (3.6 Pg C/yr), and N deposition (2.0 Pg C/yr). The increases
associated with these individual forcings add up to 13.1 Pg C/yr indicating that synergistic effects
between forcings contribute to the additional 1.4 Pg C/yr increase in GPP. The changes in
vegetation and soil plus litter carbon mass (Figures 5b and 5d) in the FULL simulation are similarly
driven by these three factors but, in addition, LUC contributes to decreases in vegetation and soil
carbon mass as natural vegetation is deforested to accommodate for increases in crop area.
Vegetation and leaf N mass (Figures 6a and 6b) decrease in the FULL simulation driven primarily
by the response to increasing $CO_2$ (orange line compared to the red line) while changes in litter
and soil N mass are affected variably by all forcings (Figure 6c). Changes in $V_{cmax}$ (Figure 6d) are
similarly affected by all forcings: increasing $CO_2$ leads to a decrease in globally-averaged $V_{cmax}$
values while changes in climate, N deposition, and fertilizer inputs lead to increases in $V_{cmax}$
values with the net result being a small decrease over the historical period. The increase in global
$NH_4^+$ mass (Figure 6e) in the FULL simulation is driven primarily by the increase in fertilizer input
while the changes in $NO_3^-$ mass are the net result of all forcings with no single forcing dominating
the response. The increase in N demand (Figure 7a) over the historical period is also driven





primarily by the increase in atmospheric $CO_2$. Plant N uptake (Figure 7b) decreases in response
to increasing $CO_2$ but increases in response to changes in climate, N deposition, and fertilizer
inputs such that the net change over the historical is a small decrease. The increase in the C:N
ratio of vegetation (Figure 8a) and its components (leaves, stem, and root) is driven primarily by
an increase in atmospheric $CO_2$. Changes in litter C:N in the FULL simulation, in contrast, do not
experience dominant influence from any one of the forcings. The simulated change in net N
mineralization (Figure 9a) in the FULL simulation, over the historical period, is small since the
decrease in net N mineralization due to increasing $CO_2$ is compensated by the increase caused by
changes in climate, N deposition, and fertilizer inputs. The remaining fluxes of nitrification, $NO_3^-$
leaching, $NH_3$ volatilization, and gaseous losses associated with nitrification and denitrification in
the FULL simulation (Figure 9) are all strongly influenced by fertilizer input (green line compared
to red line).
Table 2 compares simulated values of all primary N pools and fluxes from the FULL
simulation with other modelling and quasi observation-based studies. Simulated values are
averaged over the 1998-2017 period. Where available, time-periods for other modelling and
quasi observation-based studies to which estimates correspond are also noted. For the most part
simulated pools and fluxes lie within the range of existing studies with the exception of $N_2$ and
NO emissions that are somewhat higher.
**5.2.6 Response to all forcings except LUC**
The FULL-no-LUC simulation includes all forcings except LUC (blue line in Figures 5
through 9) and corroborates several of the points mentioned above. In this simulation crop area



stays at its 1850 value. Figure 3b (blue line) shows increasing global fertilizer input in this
simulation despite crop area staying at its 1850 value since fertilizer application rates per unit
area increase over the historical period. In the absence of the LUC, vegetation C mass (Figure 5b)
and soil plus litter C (Figure 5d) and N (Figure 6c) are higher in the FULL-no-LUC compared to the
FULL simulation. N demand (Figure 7a) is slightly higher in FULL-no LUC than in FULL simulation
because there is more standing vegetation biomass that is responding to increasing $CO_2$. The
increase in volatilization, leaching, and gaseous losses associated with nitrification and
denitrification (Figures 9c-9f) are all primarily caused by increased fertilizer input over the
specified 1850 crop area. The increase in N losses associated with these processes, over the
historical period, is much lower in the FULL-no-LUC simulation than in the FULL simulation since
crop area stays at its 1850 values.
**5.3 Comparison of FULL and ORIGINAL simulations**

We now compare the results from the FULL simulation that includes the N cycle with that

from the ORIGINAL simulation that does not include the N cycle. Both simulations are driven with
all forcings over the historical period. Figure 5a shows that the global GPP values in the FULL (red
line) and ORIGINAL (purple line) simulations are quite similar although the rate of increase of GPP
in the FULL simulation is slightly higher than in the ORIGINAL simulation. As a result, simulated
global vegetation biomass is somewhat higher in the FULL simulation (Figure 5b). The simulated
global litter and soil carbon mass (Figure 5d) is, however, lower in the FULL simulation (1073 Pg
C) compared to the ORIGINAL simulation (1142 Pg C) and this decrease mainly comes from a
decrease at higher latitudes (not shown) due to a decrease in GPP (Figure 11a). The lower GPP in
the FULL simulation, combined with the slow decomposition at cold high latitudes, results in a





lower equilibrium for litter and soil carbon compared with the ORIGINAL simulation. Overall both
these estimates are somewhat lower than the bulk density corrected estimate of 1230 Pg C based
on the Harmonized World Soil Database (HWSD) v.1.2 (Köchy et al., 2015). Figure 11a shows that
the zonal distribution of GPP from the FULL and ORIGINAL simulations, for the 1998-2017 period,
compares reasonably well to the observation-based estimate from Beer et al. (2010). The FULL
simulation has slightly lower productivity at high-latitudes than the ORIGINAL simulation, as
mentioned above. Overall, however, the inclusion of the N cycle does not change the zonal
distribution of GPP in the model substantially. Figure 11b compares the zonal distribution of GPP
from the pre-industrial simulation (corresponding to 1850s) from the FULL and FULL-with-no-
implicit-P-limitation simulations to illustrate the high GPP in the tropics where P and not N
limitation affects GPP.

Figure 12a compares globally-summed net atmosphere-land $CO_2$ flux from the FULL,

FULL-no-LUC, and ORIGINAL simulations with quasi observation-based estimates from the 2019
Global Carbon Project (Friedlingstein et al., 2019). There are two kinds of estimates in Figure 12a
from Friedlingstein et al. (2019): the first is the net atmosphere-land $CO_2$ flux for the decades
spanning the 1960s to the 2000s which are shown as rectangular boxes with their corresponding
mean values and ranges, and the second is the terrestrial sink from 1959 to 2018 (dark yellow
line). Positive values indicate a sink of carbon over land and negative values a source. The
difference between the net atmosphere-land $CO_2$ flux and the terrestrial sink two is that the
terrestrial sink minus the LUC emissions yields the net atmosphere-land $CO_2$ flux. The
atmosphere-land $CO_2$ flux from the FULL-no-LUC simulation (blue line) is directly comparable to
the terrestrial sink since 1959, since the FULL-no-LUC simulation includes no LUC, and shows that



the simulated terrestrial sink  compares fairly well to the estimates from Friedlingstein et al.
(2019). Averaged over the period 1959-2017, the modelled and Global Carbon Project values are
2.0 and 2.1 Pg C/yr, respectively. The net atmosphere-land $CO_2$ flux from the FULL simulation
mostly lies within the uncertainty range for the five decades considered, although it is on the
higher side compared to estimates from Friedlingstein et al. (2019). The reason for this is that
LUC emissions in CLASSIC are much lower than observation-based estimates, as discussed below
in context of Figure 12c. CLASSIC simulates LUC emissions only in response to changes in crop
area whereas changes in pasture area and wood harvesting also contribute to LUC emissions. The
net-atmosphere land $CO_2$ flux from the ORIGINAL simulation compares better with the estimates
from Friedlingstein et al. (2019), than the FULL simulation, because the photosynthesis down-
regulation parameter in the ORIGINAL simulation has been adjusted despite discrepancies in
simulated LUC processes.

Figure 12b compares the zonal distribution of simulated net atmosphere-land $CO_2$ flux

from the FULL and ORIGINAL simulations with the model-mean and range from the terrestrial
ecosystem models that participated in the 2019 TRENDY model intercomparison and contributed
results to 2019 Global Carbon Project (Friedlingstein et al., 2019). The carbon sink simulated by
CLASSIC in the northern hemisphere is broadly comparable to the model-mean estimate from
the TRENDY models. However, in the tropics CLASSIC simulates a much stronger sink than the
model-mean, likely because of its lower LUC emissions.
**5.4 Contribution of forcings to land C sink and sources**



Figure 12c shows cumulative net atmosphere-land $CO_2$ flux for the 1850-2017 period from
the six primary simulations with N cycle. These simulations facilitate the attribution of carbon
uptake and release over the historical period to various forcings. The cumulative terrestrial sink
in the FULL-no-LUC simulation for the period 1850-2017 is simulated to be ~153 Pg C and this
compares reasonably well with the estimate of 185 ± 50 Pg C for the period 1850-2014 from Le
Quéré et al. (2018). Increase in $CO_2$ (~115 Pg C), change in climate (~3 Pg C), and N deposition
(~19 Pg C) all contribute to this terrestrial sink. These three contributions add up to 137 Pg C so
the additional 16 Pg C is contributed by the synergistic effects between the three forcings.
Quantified in this way, the contribution of increasing $CO_2$ (15 out of 137 Pg C), climate change (3
out of 137 Pg C), and N deposition (19 out of 137 Pg C) to carbon uptake by land over the historical
period (1850-2017) is calculated to be 84%, 2%, and 14%., respectively. Cumulative LUC emissions
simulated for the period 1850-2017 by CLASSIC can be estimated using a negative cumulative
net-atmosphere-land $CO_2$ flux of ~66 Pg C from the LUC+FERT-only simulation or by the
differencing the FULL and FULL-no-LUC simulations (~71 Pg C). While LUC emissions are highly
uncertain, both of these estimates are much lower than the 195 ± 75 Pg C estimate from Le Quéré
et al. (2018).
**6.0 Discussion and conclusions**
The interactions between terrestrial C and N cycles are complex and our understanding
of these interactions, and their representation in models, is based on empirical observations of
various terrestrial ecosystem processes. In this paper, we have evaluated the response of these
interactions by perturbing the coupled C and N cycle processes in CLASSIC with one forcing at a
time over the historical period: 1) increase in $CO_2$, 2) change in climate, 3) increase in N





deposition, and 4) LUC with increasing fertilizer input. These simulations are easier to interpret
and the model response can be evaluated against both our conceptual knowledge as well as
empirical observation-based data. Our assumption is that, if the model response to individual
forcings is realistic and consistent with expectations based on empirical observations then the
response of the model to all forcings combined will also be realistic and easier to interpret,
although we do expect and see synergistic effects between forcings.

The simulated response of coupled C and N cycles in CLASSIC to increasing atmospheric

$CO_2$ is an increase in the C:N ratio of vegetation components due to an increase in their C content
but also a decrease in their N content. This model response is conceptually consistent with a
meta-analysis of 75 field experiments of elevated $CO_2$ as reported in Cotrufo et al. (1998) who
find an average reduction in tissue N concentration of 14%. Most studies analyzed in the Cotrufo
et al. (1998) meta-analysis used ambient $CO_2$ of around 350 ppm and elevated $CO_2$ of around
650-700 ppm. In comparison, the vegetation N content in CLASSIC reduces by 18% in response
to a gradual increase in atmospheric $CO_2$ from 285 ppm to 407 ppm (an increase of 122 ppm)
over the 1850-2017 period. These two estimates cannot be compared directly - the majority
(59%) of Free-Air Carbon dioxide Enrichment (FACE) experiments last less than 3 years (Jones et
al., 2014) and the vegetation experiences a large $CO_2$ change of around 300-350 ppm while the
duration of our historical simulation is 167 years and the gradual increase in $CO_2$ of 122 ppm over
the historical period is much smaller.

The response of our model to elevated $CO_2$ is also consistent with the meta-analysis of

McGuire et al. (1995) who report an average decrease in leaf N concentration of 21% in response
to elevated $CO_2$ based on 77 studies, which is the primary reason for downregulation of





photosynthetic capacity. The simulated decrease in leaf N mass in our study for the CO2-only
experiment is also 21% (Figure 6b). Although, the same caveats that apply to the comparison
with the Cotrufo et al. (1998) study also apply to this comparison. The decrease in whole plant
and leaf N in our results is conceptually consistent with the meta-analyses of McGuire et al.
(1995) and Cotrufo et al. (1998). This decrease is, in fact, necessary in our modelling framework
to induce the required downregulation of photosynthesis to simulate the land carbon sink
realistically over the historical period. However, the decrease in plant N in response to elevated
$CO_2$, found by McGuire et al. (1995) and Cotrufo et al. (1998), is inconsistent with the meta-
analysis of Liang et al. (2016) who, in contrast, report an increase in above and belowground
plant N pools in response to elevated $CO_2$ associated with increase in BNF. We are unable to
reconcile this difference between the meta-analysis of Liang et al. (2016) and those from McGuire
et al. (1995) and Cotrufo et al. (1998). Liang et al. (2016) also report results from short-term ($\leq$ 3
years) and long-term (between 3 to 15 years) studies separately (their Figure 3). They show that
the increase in total plant and litter N pools become smaller for long-term studies. The difference
in time scales of empirical studies and the real world is a caveat that will always make it difficult
to evaluate model results over long time scales.

The response of C and N cycles to changes in climate in our model is also conceptually

realistic. Globally, GPP increases in response to climate that gradually gets warmer and wetter
and as a result vegetation biomass increases. Soil carbon mass, however, decreases (despite
increase in NPP inputs) since warmer temperatures also increase heterotrophic respiration (not
shown). As a result of increased decomposition of soil organic matter, net N mineralization
increases and together with increased BNF the overall C:N ratio of vegetation and leaves





decreases, which leads to a $V_{cmax}$ increase. The small increase in $V_{cmax}$, due to the change in
climate, thus also contributes to an increase in GPP over and above that due to an increase in
temperature solely, and therefore compensates for the amount of carbon lost due to increased
soil organic matter decomposition associated with warmer temperatures. This behaviour is
consistent with land C cycle models showing a reduction in the absolute value of the strength of
the carbon-climate feedback when they include coupling of C and N cycles (Arora et al., 2019).
Modelled GPP increases in response to N deposition through an increase in leaf N content
and therefore $V_{cmax}$ values. Finally, changes in land use associated with an increase in crop area,
and the associated increase in fertilizer application rates lead to the largest increase in $NO_3^-$
leaching, $NH_3$ volatilization, and gaseous losses associated with nitrification and denitrification
among all forcings. Overall, the model response to perturbation by all individual forcings is
realistic, conceptually expected, and of the right sign (positive or negative) although it is difficult
to evaluate the magnitude of these responses in the absence of directly comparable observation-
based estimates.
Despite the model responses to individual forcings that appears consistent with our
conceptual understanding of coupled C and N cycles, our modelling framework misses an
important feedback process that has been observed in the FACE and other experiments related
to changes in natural BNF. FACE sites and other empirical studies report an increase in natural
BNF rates at elevated $CO_2$ (McGuire et al., 1995; Liang et al., 2016) and a decrease in natural BNF
rates when additional N is applied to soils (Salvagiotti et al., 2008; Ochoa-Hueso et al., 2013). On
a broad scale this is intuitively expected but the biological processes behind changes in BNF rates
remain largely unclear. A response can still be parameterized even if the underlying physical and



biological processes are not well understood. For instance, Goll et al. (2012) parameterize BNF as
an increasing and saturating function of NPP, $BNF = 1.8\left(1.0 - exp(-0.003\,NPP)\right)$. This
approach, however, does not account for the driver behind the increase in NPP - increasing
atmospheric $CO_2$, change in environmental conditions (e.g. wetter and warmer conditions), or
increased N deposition. Clearly, increasing BNF if the NPP increase is due to N deposition is
inconsistent with empirical observations. Over the historical period an increase in atmospheric
$CO_2$ has been associated with an increase in N deposition so to some extent changes in BNF due
to both forcings will cancel each other. We realize the importance of changes in BNF, given it is
the single largest natural flux of N into the coupled soil-vegetation system, and aim to address
this in a future version of the model.
Our framework assumes a constant C:N ratio of 13 for soil organic matter ($C\!:\!N_H$), an
assumption also made in other models (e.g. (Wania et al., 2012; Zhang et al., 2018). This
assumption is also broadly consistent with Zhao et al. (2019) who attempt to model C:N of soil
organic matter, among other soil properties, as a function of mean annual temperature and
precipitation using machine learning algorithms (their Figure 2h). It is difficult to currently
establish if increasing atmospheric $CO_2$ is changing $C\!:\!N_H$ given the large heterogeneity in soil
organic C and N densities, and the difficulty in measuring small trends for such large global pools.
A choice of a somewhat different value or had we chosen PFT-dependent values of $C\!:\!N_H$ is of
relatively less importance in this context since the model is spun to equilibrium for 1850
conditions anyway. It is the change in $C\!:\!N_H$ over time that is of importance. The assumption of
constant $C\!:\!N_H$ is key to yielding a decrease in vegetation N mass, and therefore leaf N mass and
$V_{cmax}$, as $CO_2$ increases in our framework. Without a decrease in $V_{cmax}$ in our modelling



framework, in response to elevated $CO_2$, we cannot achieve the downregulation noted by
McGuire et al. (1995) in their meta-analysis,  and the simulated carbon sink over the historical
period would be greater than observed as noted above. It is possible that we are simulating the
reduction in leaf N mass, in response to elevated $CO_2$, for a wrong reason in which case our model
processes need to be revisited based on additional empirical data. If our assumption of constant
or extremely slowly changing $C{:}N_H$ is indeed severely unrealistic, this necessitates a point of
caution that a realistic land carbon sink can be simulated over the historical period with such an
assumption.

Related to this assumption is also the fact that we cannot make decomposition rates of

soil organic matter a function of its C:N ratio since it is assumed to be a constant. It is well known
that after climate, litter and soil organic matter decomposition rates are controlled by their C:N
ratio (Manzoni et al., 2008). Litter decomposition rates can still be made a function of its C:N ratio
and we aim to do this for a future model version. Since the C:N ratio of litter increases over the
historical period, one implication of inclusion of this model feature will be an enhanced land
carbon sink over the historical period due to decreasing litter decomposition rates.

The work presented in this study of coupling C and N cycles in CLASSIC yields a framework

that we can build upon to make model processes more realistic, test the effect of various model
assumptions, parameterize existing processes in other ways, include additional processes, and
evaluate model response at FluxNet sites to constrain model parameters.






## Appendix



**A1. N inputs**
**A1.1 Biological N fixation**

Biological N fixation (BNF, $B_{NH4}$) is caused by both free living bacteria in the soil and by

bacteria symbiotically living within nodules of host plants' roots. Here, the bacteria convert free
nitrogen from the atmosphere to ammonium, which is used by the host plants. Like any other
microbial activity, BNF is limited both by drier soil moisture conditions and cold temperatures.
Cleveland et al. (1999) attempt to capture this by parameterizing BNF as a function of actual
evapotranspiration (AET). AET is a function primarily of soil moisture (through precipitation and
soil water balance) and available energy. In places where vegetation exists, AET is also affected
by vegetation characteristics including LAI and rooting depth. Here, we parameterize BNF ($B_{NH4}$,
gN m$^{-2}$ day$^{-1}$) as a function of modelled soil moisture and temperature to depth of 0.5 m
following Xu-Ri and Prentice (2008) which yields a very similar geographical distribution of BNF
as the Cleveland et al. (1999) approach as seen in Figure 4c.

$$
\begin{aligned}
B_{NH4} &= \left( \sum_c \alpha_c\, f_c + \sum_n \alpha_n\, f_n + \right) f(T_{0.5})\, f(\theta_{0.5}) \\
f(T_{0.5}) &= 2^{(T_{0.5}-25)/10} \\
f(\theta_{0.5}) &= \min\left(0, \max\left(1, \frac{\theta_{0.5}-\theta_w}{\theta_{fc}-\theta_w}\right)\right)
\end{aligned}
\tag{A1}
$$


where $\alpha_c$ and $\alpha_n$ (gN m$^{-2}$ day$^{-1}$) are BNF coefficients for crop (c) and non-crop or natural (n) PFTs,
which are area weighted using the fractional coverages $f_c$ and $f_n$ of crop and non-crop PFTs that
are present in a grid cell, $f(T)$ is the dependence on soil temperature based on a Q$_{10}$ formulation
and $f(\theta)$ is the dependence on soil moisture which varies between 0 and 1. $\theta_{fc}$ and $\theta_w$ are the



soil moisture at field capacity and wilting points, respectively. $T_{0.5}$ (°C) and $\theta_{0.5}$ (m³ m⁻³) in
equation (A1) are averaged over the 0.5 m soil depth over which BNF is assumed to occur. We do
not make the distinction between symbiotic and non-symbiotic BNF since this requires explicit
knowledge of geographical distribution of N fixing PFTs which are not represented separately in
our base set of nine PFTs. A higher value of $\alpha_c$ is used compared to $\alpha_n$ to account for the use of
N fixing plants over agriculture areas. Biological nitrogen fixation has been an essential
component of many farming systems for considerable periods, with evidence for the agricultural
use of legumes dating back more than 4,000 years (O'Hara, 1998). A higher $\alpha_c$ than $\alpha_n$ is also
consistent with Fowler et al. (2013) who report BNF of 58 and 60 Tg N yr⁻¹ for natural and
agricultural ecosystems for present day. Since the area of natural ecosystems is about five times
the current cropland area it implies BNF rate per unit land area is higher for crop ecosystems than
for natural ecosystems. Values of $\alpha_c$ than $\alpha_n$ and other model parameters are summarized in
Table A1.

Similar to Cleveland et al. (1999), our approach does not lead to a significant change in

BNF with increasing atmospheric $CO_2$, other than through changes in soil moisture and
temperature. At least two meta-analyses, however, suggest that an increase in atmospheric $CO_2$
does lead to an increase in BNF through increased symbiotic activity associated with an increase
in both nodule mass and number (McGuire et al., 1995; Liang et al., 2016). Models have
attempted to capture this by simulating BNF as a function of NPP (Thornton et al., 2007; Wania
et al., 2012). The caveat with this approach and the implications of our BNF approach are
discussed in Section 6.
**A1.2 Atmospheric N deposition**



Atmospheric N deposition is externally specified. The model reads in spatially- and
temporally-varying annual deposition rates from a file.  Deposition is assumed to occur at the
same rate throughout the year so the same daily rate (gN m$^{-2}$ day$^{-1}$) is used for all days of a given
year. If separate information for ammonium (NH$_4^+$) and nitrate (NO$_3^-$) deposition rates is available
then it is used otherwise deposition is assumed to be split equally between NH$_4^+$ and NO$_3^-$
(indicated as $P_{NH4}$ and $P_{NO3}$ in equations 4 and 5).

**A1.3 Fertilizer application**

Geographically and temporally varying annual fertilizer application rates ($F_{NH4}$) are also
specified externally and read in from a file. Fertilizer application occurs over the C$_3$ and C$_4$ crop
fractions of grid cells. Agricultural management practices are difficult to model since they vary
widely between countries and even from farmer to farmer. For simplicity, we assume fertilizer is
applied at the same daily fertilizer application rate (gN m$^{-2}$ day$^{-1}$) throughout the year in the
tropics (between 30°S and 30°N), given the possibility of multiple crop rotations in a given year.
Between the 30° and 90° latitudes in both northern and southern hemispheres, we assume that
fertilizer application starts on the spring equinox and ends on the fall equinox. The annual
fertilizer application rate is thus distributed over around 180 days. This provides somewhat more
realism, than using the same treatment as in tropical regions, since extra-tropical agricultural
areas typically do not experience multiple crop rotations in a given year.

**A2. N cycling in plants and soil**

Plant roots take up mineral N from soil and then allocate it to leaves and stem to maintain
an optimal C:N ratio of each component. Litterfall from vegetation contributes to the litter pool



and decomposition of litter transfers humified litter to the soil organic matter pool.
Decomposition of litter and soil organic matter returns mineralized N back to the $NH_4^+$ pool,
closing the soil-vegetation N cycle loop. Both active and passive plant uptakes of N (from both
the $NH_4^+$ and $NO_3^-$ pools) are explicitly modelled. The modelled plant N uptake is a function of its
N demand. Higher N demand leads to higher mineral N uptake from soil.
**A2.1 Plant N demand**

Plant N demand is calculated based on the fraction of NPP allocated to leaves, stem, and

root components and their specified minimum PFT-dependent C:N ratios, similar to other models
(Xu-Ri and Prentice, 2008; Jiang et al., 2019). The assumption is that plants always want to
achieve their desired minimum C:N ratios if enough N is available.
$$\Delta_{WP} = \Delta_L + \Delta_R + \Delta_S$$
$$\Delta_i = \frac{\max\left(0, NPP \cdot a_{i,C}\right)}{C{:}N_{i,\min}}, \quad i = L, S, R \tag{A2}$$

where the whole plant N demand ($\Delta_{WP}$) is the sum of N demand for the leaves ($\Delta_L$), stem ($\Delta_S$),
and root ($\Delta_R$) components, $a_{i,C}, \ i = L, S, R$ is the fraction of NPP (i.e. carbon as indicated by
letter C in the subscript) allocated to leaf, stem, and root components, and $C{:}N_{i,\min}, i = L, S, R$
are their specified minimum C:N ratios (see Table A1 for these and all other model parameters).
A caveat with this approach when applied at the daily time step, for biogeochemical processes in
our model, is that during periods of time when NPP is negative due to adverse climatic conditions
(e.g. during winter or drought seasons), the calculated demand is negative. If positive NPP implies
there is demand for N, negative NPP cannot be taken to imply that N must be lost from
vegetation. As a result, from a plant's perspective, N demand is assumed to be zero during





periods of negative NPP. N demand is also set to zero when all leaves have been shed (i.e., when
GPP is zero). At the global scale, this leads to about 15% higher annual N demand than would be
the case if negative NPP values were taken into consideration.

**A2.2 Passive N uptake**

N demand is weighed against passive and active N uptake. Passive N uptake depends on

the concentration of mineral N in the soil and the water taken up by the plants through their
roots as a result of transpiration. We assume that plants have no control over N that comes into
the plant through this passive uptake. This is consistent with existing empirical evidence that too
much N in soil will cause N toxicity (Goyal and Huffaker, 1984), although we do not model N
toxicity in our framework. If the N demand for the current time step cannot be met by passive N
uptake then a plant compensates for the deficit (i.e., the remaining demand) through active N
uptake.

The $NH_4^+$ concentration in the soil moisture within the rooting zone, referred to as $[NH_4]$

($gN\ gH_2O^{-1}$), is calculated as
$$[NH_4] = \frac{N_{NH4}}{\sum_{i=1}^{i \le r_d} 10^6\ \theta_i\ z_i} \qquad (A3)$$
where $N_{NH4}$ is ammonium pool size ($gN\ m^{-2}$), $\theta_i$ is the volumetric soil moisture content for soil
layer $i$ ($m^3\ m^{-3}$), $z_i$ is the thickness of soil layer $i$ (m), $r_d$ is the soil layer in which the 99% rooting
depth lies as dynamically simulated by the biogeochemical module of CLASSIC following Arora
and Boer (2003). The $10^6$ term converts units of the denominator term to $gH_2O\ m^{-2}$. $NO_3^-$
concentration ($[NO_3]$, $gN\ gH_2O^{-1}$) in the rooting zone is found in a similar fashion. The



transpiration flux $q_t$ (kgH$_2$O m$^{-2}$ s$^{-1}$) (calculated in the physics module of CLASSIC) is multiplied
by [NH$_4$] and [NO$_3$] (gN gH$_2$O$^{-1}$) to obtain passive uptake of NH$_4^+$ and NO$_3^-$ (gN m$^{-2}$ day$^{-1}$) as
$$U_{p,NH4} = 86400 \times 10^3\, \beta\, q_t[\text{NH}_4]$$
$$U_{p,NO3} = 86400 \times 10^3\, \beta\, q_t[\text{NO}_3]$$
(A4)

where the multiplier 86400$\times 10^3$ converts $q_t$ to units of gH$_2$O m$^{-2}$ day$^{-1}$, and $\beta$ (see Table A1) is
the dimensionless mineral N distribution coefficient with value less than 1 that accounts for the
fact that NH$_4$ and NO$_3$ available in the soil are not well mixed in the soil moisture solution, and
not completely accessible to roots, to be taken up by plants.
**A2.3 Active N uptake**
The active plant N uptake is parameterized as a function of fine root biomass and the size
of NH$_4^+$ and NO$_3^-$ pools in a manner similar to Gerber et al. (2010) and Wania et al (2012). CLASSIC
does not explicitly models fine root biomass. We therefore calculate the fraction of fine root
biomass using an empirical relationship that is very similar to the relationship developed by Kurz
et al. (1996) (their equation 5) but also works below total root biomass of 0.33 Kg C m$^{-2}$ (the Kurz
et al. (1996) relationship yields a fraction of fine root more than 1.0 below this threshold). The
fraction of fine root biomass ($f_r$) is given by
$$f_r = 1 - \frac{C_R}{C_R + 0.6}$$
(A5)

where $C_R$ is the root biomass (KgC m$^{-2}$) simulated by the biogeochemical module of CLASSIC.
Equation (A5) yields fine root fraction approaching 1.0 as $C_R$ approaches 0, so at very low root
biomass values all roots are considered fine roots. For grasses the fraction of fine root biomass is
set to 1. The maximum or potential active N uptake is given by





$$U_{a,pot,\text{NH4}} = \frac{\varepsilon\, f_r\, C_R\, \text{N}_{\text{NH4}}}{k_{p,\frac{1}{2}}\, r_d + \text{N}_{\text{NH4}} + \text{N}_{\text{NO3}}}$$
$$U_{a,pot,\text{NO3}} = \frac{\varepsilon\, f_r\, C_R\, \text{N}_{\text{NO3}}}{k_{p,\frac{1}{2}}\, r_d + \text{N}_{\text{NH4}} + \text{N}_{\text{NO3}}}$$
(A6)

where $\varepsilon$ is the efficiency of fine roots to take up N per unit fine root mass per day (gN gC$^{-1}$ day$^-$
$^1$), $k_{p,\frac{1}{2}}$ is the half saturation constant (gN m$^{-3}$), and $\text{N}_{\text{NH4}}$ and $\text{N}_{\text{NO3}}$ are the ammonium and
nitrate pool sizes (gN m$^{-2}$) as mentioned earlier. Depending on the geographical location and the
time of the year, if passive uptake alone can satisfy the plant demand the actual active N uptake
of NH4 ($U_{a,actual,\text{NH4}}$) and NO3 ($U_{a,actual,\text{NO3}}$) is set to zero. Conversely, during other times both
passive and potential active N uptakes may not be able to satisfy the demand and in this case
actual active N uptake is equal to its potential rate. At times other than these, the actual active
uptake is lower than its potential value. This adjustment of actual active uptake is illustrated in
equation (A7).
$$\text{if} \left(\Delta_{WP} \leq U_{p,NH4} + U_{p,NO3}\right)$$
$$U_{a,actual,\text{NH4}} = 0$$
$$U_{a,actual,\text{NO3}} = 0$$

$$\text{if} \left(\Delta_{WP} > U_{p,NH4} + U_{p,NO3}\right) \wedge \left(\Delta_{WP} < U_{p,NH4} + U_{p,NO3} + U_{a,pot,\text{NH4}} + U_{a,pot,\text{NH4}}\right)$$
$$U_{a,actual,\text{NH4}} = \left(\Delta_{WP} - U_{p,NH4} - U_{p,NO3}\right)\frac{U_{a,pot,\text{NH4}}}{U_{a,pot,\text{NH4}} + U_{a,pot,\text{NH4}}}$$
$$U_{a,actual,\text{NO3}} = \left(\Delta_{WP} - U_{p,NH4} - U_{p,NO3}\right)\frac{U_{a,pot,\text{NO3}}}{U_{a,pot,\text{NH4}} + U_{a,pot,\text{NH4}}}$$
(A7)

$$\text{if} \left(\Delta_{WP} \geq U_{p,NH4} + U_{p,NO3} + U_{a,pot,\text{NH4}} + U_{a,pot,\text{NO3}}\right)$$
$$U_{a,actual,\text{NH4}} = U_{a,pot,\text{NH4}}$$
$$U_{a,actual,\text{NO3}} = U_{a,pot,\text{NO3}}$$

Finally, the total N uptake ($U$), uptake of NH4$^+$ ($U_{NH4}$) and NO3$^-$ ($U_{NO3}$), are calculated as



$$U = U_{p,NH4} + U_{p,NO3} + U_{a,actual,\text{NH4}} + U_{a,actual,\text{NO3}}$$
$$U_{NH4} = U_{p,NH4} + U_{a,actual,\text{NH4}}$$
$$U_{NO3} = U_{p,NO3} + U_{a,actual,\text{NO3}}$$
(A8)


**A2.4 Litterfall**
Nitrogen litterfall from the vegetation components is directly tied to the carbon litterfall
calculated by the phenology module of CLASSIC through their current C:N ratios.
$$LF_i = \frac{(1-r_L)LF_{i,C}}{C:N_i}, i = L, S, R$$
(A9)

where $LF_{i,C}$ is the carbon litterfall rate (gC day$^{-1}$) for component $i$, calculated by the phenology
module of CLASSIC, and division by its current C:N ratio yields the nitrogen litterfall rate, $r_L$ is the
leaf resorption coefficient that simulates the resorption of N from leaves of deciduous tree PFTs
before they are shed and $r_i = 0, i = R, S$. Litter from each vegetation component is proportioned
between structural and non-structural components according to their pool sizes.
**A2.5 Allocation and reallocation**
Plant N uptake by roots is allocated to leaves and stem to satisfy their N demand. When
plant N demand is greater than zero, total N uptake ($U$) is divided between leaves, stem, and root
components in proportion to their demands such that the allocation fractions for N ($a_i, i =$
$L, S, R$) are calculated as
$$a_i = \frac{\Delta_i}{\Delta_{WP}}, i = L, S, R$$
$$A_{R2L} = a_L (U_{NH4} + U_{NO3})$$
$$A_{R2S} = a_S (U_{NH4} + U_{NO3})$$
(A10)



where $A_{R2L}$ and $A_{R2S}$ are the amounts of N allocated from root to leaves and stem components,
respectively, as mentioned in the main text for equation (8). During periods of negative NPP due
to adverse climatic conditions (e.g. during winter or drought seasons) the plant N demand is set
to zero but passive N uptake, associated with transpiration, may still be occurring if the leaves
are still on. Even though there is no N demand, passive N uptake still needs to be partitioned
among the vegetation components. During periods of negative NPP allocation fractions for N are,
therefore, calculated in proportion to the minimum PFT-dependent C:N ratios of the leaves,
stem, and root components as follows.
$$a_i = \frac{1/C{:}N_{i,\min}}{1/C{:}N_{L,\min}+1/C{:}N_{S,\min}+1/C{:}N_{R,\min}} \ , i = L, S, R \qquad \text{(A11)}$$

For grasses, which do not have a stem component, equations (A10) and (A11) are modified
accordingly by removing the terms associated with the stem component.
Three additional rules override these general allocation rule specifically for deciduous
tree PFTs (or deciduous PFTs in general). First, no N allocation is made to leaves once leaf fall is
initiated for deciduous tree PFTs and plant N uptake is proportioned between stem and root
components based on their demands in a manner similar to equation (A10). Second, for
deciduous tree PFTs, a fraction of leaf N is resorbed from leaves back into stem and root as
follows
$$R_{L2R} = r_L \, LF_L \frac{N_{R,NS}}{N_{R,NS}+N_{S,NS}}$$
$$R_{L2S} = r_L \, LF_L \frac{N_{S,NS}}{N_{R,NS}+N_{S,NS}} \qquad \text{(A12)}$$




where $r_L$ is the leaf resorption coefficient, as mentioned earlier, and $LF_L$ is the leaf litter fall rate.
Third, and similar to resorption, at the time of leaf onset for deciduous tree PFTs, N is reallocated
to leaves (in conjunction with reallocated carbon as explained in Asaadi et al. (2018)) from stem
and root components.
$$R_{R2L} = \frac{R_{R2L,C}}{C:N_L} \frac{N_{R,NS}}{N_{R,NS}+N_{S,NS}}$$
$$R_{S2L} = \frac{R_{S2L,C}}{C:N_L} \frac{N_{S,NS}}{N_{R,NS}+N_{S,NS}}$$
(A13)

where $R_{R2L,C}$ and $R_{S2L,C}$ represent reallocation of carbon from non-structural stem and root
components to leaves and division by $C:N_L$ converts the flux into N units. The reallocation
demand for N, at the time of leaf onset, is proportioned between non-structural pools of stem
and root according to their sizes.
**A2.6 N mineralization, immobilization, and humification**
Decomposition of litter ($R_{h,D}$) and soil organic matter ($R_{h,H}$) releases C to the atmosphere
and this flux is calculated by the heterotrophic respiration module of CLASSIC. The amount of N
mineralized is calculated straightforwardly by division with the current C:N ratios of the
respective pools and contributes to the $NH_4^+$ pool.
$$M_{D,NH4} = \frac{R_{h,D}}{C:N_D}$$
$$M_{H,NH4} = \frac{R_{h,H}}{C:N_H}$$
(A14)

An implication of mineralization contributing to the $NH_4^+$ pool, in addition to BNF and fertilizer
inputs that also contribute solely to the $NH_4^+$ pool, is that the simulated $NH_4^+$ pool is typically



larger than the $NO_3^-$ pool. The exception is the dry and arid regions where the lack of
denitrification, as discussed below in Section A.3.2., leads to a build up of the $NO_3^-$ pool.

Immobilization of mineral N from the $NH_4^+$ and $NO_3^-$ pools into the soil organic matter

pool is meant to keep the soil organic matter C:N ratio ($C{:}N_H$) at its specified value of 13 for all
PFTs in a manner similar to Wania et al. (2012) and Zhang et al. (2018). A value of 13 is within the
range of observation-based estimates which vary from about 8 to 25 (Zinke et al., 1998; Tipping
et al., 2016). Although $C{:}N_H$ varies geographically, the driving factors behind this variability
remain unclear. It is even more difficult to establish if increasing atmospheric $CO_2$ is changing
$C{:}N_H$ given the large heterogeneity in soil organic C and N densities, and the difficulty in
measuring small trends for such large global pools. We therefore make the assumption that the
$C{:}N_H$ does not change with time. An implication of this assumption is that as GPP increases with
increasing atmospheric $CO_2$ rises, and plant litter becomes enriched in C with increasing C:N ratio
of litter, more and more N is locked up in the soil organic matter pool because its C:N ratio is
fixed. As a result, mineral N pools of $NH_4^+$ and $NO_3^-$ decrease in size and plant N content
subsequently follows. This is consistent with studies of plants grown in elevated $CO_2$
environment. For example, Cotrufo et al. (1998) summarize results from 75 studies and find an
average 14% reduction in N concentration for above-ground tissues. Wang et al. (2019) find
increased C concentration by 0.8–1.2% and a reduction in N concentration by 7.4–10.7% for rice
and winter wheat crop rotation system under elevated $CO_2$.

Immobilization from both the $NH_4^+$ and $NO_3^-$ pools is calculated in proportion to their pool

sizes, employing the fixed $C{:}N_H$ ratio as

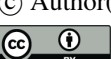



$$O_{NH4} = \max\left(0, \left(\frac{C_H}{C:N_H} - N_H\right)\frac{N_{NH4}}{N_{NH4}+N_{NO3}}\right)$$
$$O_{NO3} = \max\left(0, \left(\frac{C_H}{C:N_H} - N_H\right)\frac{N_{NO3}}{N_{NH4}+N_{NO3}}\right)$$
(A15)

Finally, the carbon flux of humified litter from the litter to the soil organic matter pool ($H_{C,D2H}$)
is also associated with a corresponding N flux that depends on the C:N ratio of the litter pool.
$$H_{N,D2H} = \frac{H_{C,D2H}}{C:N_D}$$
(A16)

**A3. N cycling in mineral pools and N outputs**
This section presents the parameterizations of nitrification (which results in transfer of N
from the $NH_4^+$ to the $NO_3^-$ pool) and the associated gaseous fluxes of $N_2O$ and NO (referred to as
nitrifier denitrification), gaseous fluxes of $N_2O$, NO, and $N_2$ associated with denitrification,
volatilization of $NH_4$ into $NH_3$, and leaching of $NO_3^-$ in runoff.
**A3.1 Nitrification**
Nitrification, the oxidative process converting ammonium to nitrate, is driven by microbial
activity and as such constrained both by high and low soil moisture (Porporato et al., 2003). At
high soil moisture content there is little aeration of soil and this constrains aerobic microbial
activity, while at low soil moisture content microbial activity is constrained by moisture
limitation. In CLASSIC, the heterotrophic respiration from soil carbon is constrained similarly but
rather than using soil moisture the parameterization is based on soil matric potential (Arora,
2003; Melton et al., 2015). Here, we use the exact same parameterization. In addition to soil
moisture, nitrification (gN m$^{-2}$ day$^{-1}$) is modelled as a function of soil temperature and the size
of the $NH_4^+$ pool as follows



$$I_{NO3} = \eta \, f_I(T_{0.5}) f_I(\psi) \, N_{NH4} \qquad \text{(A17)}$$

where $\eta$ is the nitrification coefficient (day$^{-1}$), $f_I(\psi)$ is the dimensionless soil moisture scalar that
varies between 0 and 1 and depends on soil matric potential ($\psi$), $f_I(T_{0.5})$ is the dimensionless
soil temperature scalar that depends on average soil temperature ($T_{0.5}$) over the top 0.5 m soil
depth over which nitrification is assumed to occur (following Xu-Ri and Prentice, 2008), and $N_{NH4}$
is the ammonium pool size (gN m$^{-2}$), as mentioned earlier. Both $f_I(T_{0.5})$ and $f_I(\psi)$ are
parameterized following Arora (2003) and Melton et al. (2015). $f_I(T_{0.5})$ is a Q$_{10}$ type function
with a temperature dependent Q$_{10}$
$$f_I(T_{0.5}) = Q_{10,I}^{(T_{0.5}-20)/10}, Q_{10,I} = 1.44 + 0.56 \left( \tanh\left( 0.075(46 - T_{0.5}) \right) \right) \qquad \text{(A18)}$$

The reference temperature for nitrification is set to 20 °C following Lin et al. (2000). $f_I(\psi)$ is
parameterized as a step function of soil matric potential ($\psi$) as
$$f_I(\psi) = \begin{cases} 0.5 & \text{if } \psi \leq \psi_{sat} \\ 1 - 0.5 \frac{\log(0.4) - \log(\psi)}{\log(0.4) - \log(\psi_{sat})} & \text{if } 0.4 > \psi \geq \psi_{sat} \\ 1 & \text{if } 0.6 \geq \psi \geq 0.4 \\ 1 - 0.8 \frac{\log(\psi) - \log(0.6)}{\log(100) - \log(0.6)} & \text{if } 100 > \psi > 0.6 \\ 0.2 & \text{if } \psi > 100 \end{cases} \qquad \text{(A19)}$$

where the soil matric potential ($\psi$) is found, following Clapp and Hornberger (1978), as a function
of soil moisture ($\theta$)
$$\psi(\theta) = \psi_{sat} \left( \frac{\theta}{\theta_{sat}} \right)^{-B}. \qquad \text{(A20)}$$

Saturated matric potential ($\psi_{sat}$), soil moisture at saturation (i.e. porosity) ($\theta_{sat}$), and the
parameter $B$ are calculated as functions of percent sand and clay in soil following Clapp and



Hornberger (1978) as shown in Melton et al. (2015). The soil moisture scalar $f_I(\psi)$ is calculated
individually for each soil layer and then averaged over the soil depth of 0.5 m over which
nitrification is assumed to occur.

Gaseous fluxes of NO ($I_{NO}$) and N$_2$O ($I_{N2O}$) associated with nitrification, and generated

through nitrifier denitrification, are assumed to be directly proportional to the nitrification flux
($I_{NO3}$) as

$$
\begin{aligned}
I_{NO} &= \eta_{NO}\ I_{NO3} \\
I_{N2O} &= \eta_{N2O}\ I_{NO3}
\end{aligned}. \tag{A21}
$$


where $\eta_{NO}$ and $\eta_{N2O}$ are dimensionless fractions which determine what fractions of nitrification
flux  are emitted as NO and N$_2$O.
**A3.2 Denitrification**
Denitrification is the stepwise microbiological reduction of nitrate to NO, N$_2$O, and ultimately to
N$_2$ in complete denitrification. Unlike nitrification, however, denitrification is primarily an
anaerobic process (Tomasek et al., 2017) and therefore occurs when soil is saturated. As a result,
we use a different soil moisture scalar than for nitrification. Similar to nitrification, denitrification
is modelled as a function of soil moisture, soil temperature and the size of the NO$_3^-$ pool as follows
to calculate the gaseous fluxes of NO, N$_2$O, and N$_2$.

$$
\begin{aligned}
E_{NO} &= \mu_{NO}\ f_E(T_{0.5})\ f_E(\theta)\ N_{NO3} \\
E_{N2O} &= \mu_{N2O}\ f_E(T_{0.5})\ f_E(\theta)\ N_{NO3} \\
E_{N2} &= \mu_{N2}\ f_E(T_{0.5})\ f_E(\theta)\ N_{NO3}
\end{aligned} \tag{A22}
$$


where $\mu_{NO}$, $\mu_{N2O}$, and $\mu_{N2}$ are coefficients (day$^{-1}$) that determine daily rates of emissions of NO,
N$_2$O, and N$_2$. The temperature scalar $f_E(T_{0.5})$ is exactly the same as the one for nitrification



$(f_l(T_{0.5}))$ since denitrification is also assumed to occur over the same 0.5 soil depth. The soil
moisture scalar $f_E(\theta)$ is given by
$$f_E(\theta) = 1 - \tanh\left(2.5\left(\frac{1-w(\theta)}{1-w_d}\right)^2\right)$$
$$w(\theta) = \max\left(0, \min\left(1, \frac{\theta-\theta_w}{\theta_f-\theta_w}\right)\right)$$
(A23)

where $w$ is the soil wetness that varies between 0 and 1 as soil moisture varies between wilting
point $(\theta_w)$ and field capacity $(\theta_f)$, and $w_d$ is the threshold soil wetness for denitrification below
which very little denitrification occurs. Since very little denitrification occurs when soil wetness
is below $w_d$ this leads to build up of the $NO_3^-$ pool in arid regions.
**A3.3 $NO_3^-$ leaching**

Leaching is the loss of water-soluble ions through runoff. In contrast to positively charged

$NH_4^+$ ions (i.e. cations), the $NO_3^-$ ions do not bond to soil particles because of the limited exchange
capacity of soil for negatively charged ions (i.e. anions). As a result, leaching of N in the form of
$NO_3^-$ ions is a common water quality problem, particularly over cropland regions. The leaching
flux ($L_{NO3}$, gN m$^{-2}$ day$^{-1}$) is parameterized to be directly proportional to baseflow ($b_t$, Kg m$^{-2}$ s$^-$
$^1$) calculated by the physics module of CLASSIC and the size of the $NO_3$ pool ($N_{NO3}$, gN m$^{-2}$).
Baseflow is the runoff rate from the bottommost soil layer.
$$L_{NO3} = 86400 \, \varphi \, b_t \, N_{NO3}$$
(A24)

where the multiplier 86400 converts units to per day, and $\varphi$ is the leaching coefficient (m$^2$ Kg$^{-1}$)
that can be thought of as the soil particle surface area (m$^2$) that 1 Kg of water (or about 0.001
m$^3$) can effectively wash to leach the nutrients.



### A3.4 NH₃ volatilization

NH$_3$ volatilization ($V_{NH3}$, gN m$^{-2}$ day$^{-1}$) is parametrized as a function of pool size of NH$_4^+$, soil temperature, soil pH, aerodynamic and boundary layer resistances, and atmospheric NH$_3$ concentration in a manner similar to Riddick et al. (2016) as

$$V_{NH4} = \vartheta \ 86400 \ \frac{1}{r_a+r_b}\left(\chi - [NH_{3,a}]\right) \tag{A25}$$

where $\vartheta$ is the dimensionless NH$_3$ volatilization coefficient which is set to less than 1 to account for the fact that a fraction of ammonia released from the soil is captured by vegetation, $r_a$ (s m$^{-1}$) is the aerodynamic resistance calculated by the physics module of CLASSIC, $\chi$ is the ammonia (NH$_3$) concentration at the interface of the top soil layer and the atmosphere (g m$^{-3}$), $[NH_{3,a}]$ is the atmospheric NH$_3$ concentration specified at 0.3×10$^{-6}$ g m$^{-3}$ following Riddick et al. (2016), 86400 converts flux units from gN m$^{-2}$ s$^{-1}$ to gN m$^{-2}$ day$^{-1}$, and $r_b$ (s m$^{-1}$) is the boundary layer resistance calculated following Thom (1975) as

$$r_b = 6.2 \ u_*^{-0.67} \tag{A26}$$

where $u_*$ (m/s) is the friction velocity provided by the physics module of CLASSIC. The ammonia (NH$_3$) concentration at surface ($\chi$), in a manner similar to Riddick et al. (2016), is calculated as

$$\chi = 0.26 \frac{N_{NH4}}{1+K_H+K_H[H^+]/K_{NH4}} \tag{A27}$$

where the coefficient 0.26 is the fraction of ammonium in the top 10 cm soil layer assuming exponential distribution of ammonium along the soil depth (given by $3e^{-3z}$, where $z$ is the soil depth), $K_H$ (dimensionless) is the Henry's law constant for NH$_3$, $K_{NH4}$ (mol L$^{-1}$) is the dissociation equilibrium constant for aqueous NH$_3$, and $H^+$ (mol L$^{-1}$) is the concentration of hydrogen ion



that depends on the soil pH ($H^+ = 10^{-pH}$). $K_H$ and $K_{NH4}$ are modelled as functions of soil
temperature of the top 10 cm soil layer ($T_{0.1}$) following Riddick et al. (2016) as

$$K_H = 4.59 \, T_{0.1} \exp\left( 4092 \left( \frac{1}{T_{0.1}} - \frac{1}{T_{ref,v}} \right) \right)$$

$$K_{NH4} = 5.67 \times 10^{-10} \exp\left( -6286 \left( \frac{1}{T_{0.1}} - \frac{1}{T_{ref,v}} \right) \right)$$

(A28)

where $T_{ref,v}$ is the reference temperature of 298.15 K.


**Acknowledgments**

We are grateful and thank Joe Melton and Paul Bartlett for their comments on an earlier version
of this manuscript.

**Code/Data availability**

Model code for the operational CLASSIC model can be obtained from
https://gitlab.com/cccma/classic. Changes made to the operational version to include N cycle
and the results shown here can be obtained from the second author.

**Author contributions**

A.A. implemented the N cycle in the CLASSIC code, put together all the N cycle related input
data, and performed all the simulations. V.A. and A.A. wrote the manuscript.

**Competing interests**

There are no competing interests.








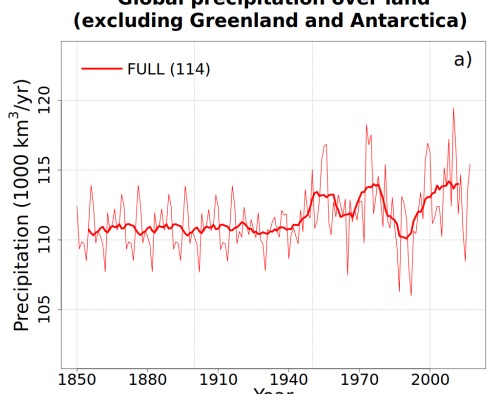 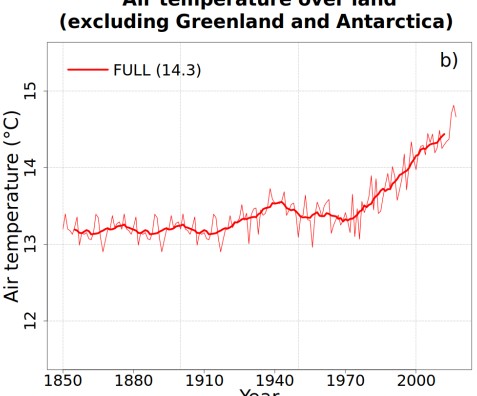



Figure A1: Annual values of global precipitation (a) and air temperature (b) over land in the
CRU-JRA reanalysis data that are used to drive the model. The data are available for the period
1901-2017. In the absence of meterological data, the period 1851-1900 uses the data from the
period 1901-1925 twice.



**Table A1**: Model parameters for various model parameterizations. Model parameters may be scalar or
an array (if they are PFT dependent) in which they are follow the following structure.

| Needleleaf evergreen | Needleleaf deciduous | |
|---|---|---|
| Broadleaf evergreen | Broadleaf deciduous cold | Broadleaf deciduous drought |
| $C_3$ crop | $C_4$ crop | |
| $C_3$ grass | $C_4$ grass | |


Corresponding equation in which the parameter appears in the main text is also noted.

| Model parameter | Eqn | Description | Units | Value(s) | | |
|---|---|---|---|---|---|---|
| *Biological N fixation* | | | | | | |
| $\alpha_c$ | A1 | BNF rate for crop PFTs | gN m$^{-2}$ day$^{-1}$ | 0.00217 | | |
| $\alpha_n$ | A1 | BNF rate for natural PFTs | gN m$^{-2}$ day$^{-1}$ | 0.00037 | | |
| *Plant N demand* | | | | | | |
| $C:N_{L,min}$ | A2 | Minimum C:N ratio for leaves | dimensionless | 25<br>20<br>16<br>13 | 22<br>18<br>20<br>18 | 18 |
| $C:N_{S,min}$ | A2 | Minimum C:N ratio for stem | dimensionless | 450<br>430<br>285<br>– | 450<br>430<br>285<br>– | 430 |
| $C:N_{R,min}$ | A2 | Minimum C:N ratio for root | dimensionless | 45<br>35<br>30<br>30 | 45<br>35<br>35<br>35 | 35 |
| *Plant uptake* | | | | | | |
| $\beta$ | A4 | Mineral N distribution coefficient | dimensionless | 0.5 | | |
| $\varepsilon$ | A6 | Fine root efficiency | gN gC$^{-1}$ day$^{-1}$ | 4.92E-5 | | |
| $k_{p,\frac{1}{2}}$ | A6 | Half saturation constant | gN m$^{-3}$ | 3 | | |
| *Litterfall* | | | | | | |
| $r_L$ | A9 | Leaf resorption coefficient | dimensionless | 0.54 | | |
| *Nitrification* | | | | | | |
| $\eta$ | A17 | Nitrification coefficient | day$^{-1}$ | 7.33E-4 | | |





| | | | | |
|---|---|---|---|---|
| $\eta_{NO}$ | A21 | Fraction of nitrification flux emitted as NO | dimensionless | 7.03E-5 |
| $\eta_{N2O}$ | A21 | Fraction of nitrification flux emitted as $N_2O$ | dimensionless | 2.57E-5 |
| *Denitrification* | | | | |
| $\mu_{NO}$ | A22 | Fraction of denitrification flux emitted as NO | $day^{-1}$ | 3.872E-4 |
| $\mu_{N2O}$ | A22 | Fraction of denitrification flux emitted as $N_2O$ | $day^{-1}$ | 1.408E-4 |
| $\mu_{N2}$ | A22 | Fraction of denitrification flux emitted as $N_2$ | $day^{-1}$ | 3.872E-3 |
| $w_d$ | A23 | Soil wetness threshold below which very little denitrification occurs | dimensionless | 0.3 |
| *Leaching* | | | | |
| $\varphi$ | A24 | Leaching coefficient | $m^2\ Kg^{-1}$ | 1.15E-3 |
| *NH₃ volatilization* | | | | |
| $\vartheta$ | A25 | NH₃ volatilization coefficient | dimensionless | 1.8 |
| *Coupling of C and N cycles* | | | | |
| $\Gamma_1$ | 15 | Parameter for calculating $V_{cmax}$ from leaf N content | $\mu mol\ CO_2\ gN^{-1}\ s^{-1}$ | 13 (all PFTs except broadleaf evergreen tree)<br>5.1 (for broadleaf evergreen tree) |
| $\Gamma_2$ | 15 | Parameter for calculating $V_{cmax}$ from leaf N content | $\mu mol\ CO_2\ m^{-2}\ s^{-1}$ | 8.5 |






**Table 1**: Historical simulations performed over the period 1851-2017 to evaluate the model's
response to various forcings. All forcings are time varying. All forcings are also spatially explicit
except atmospheric $CO_2$ for which a globally constant value is specified.


| Simulation name | Forcing that varies over the historical period | N cycle |
|---|---|---|
| *Primary simulations performed to evaluate N cycle response to various forcings* | | |
| 1. CO2-only | Atmospheric $CO_2$ concentration | Runs with N cycle |
| 2. CLIM-only | 1901-1925 meteorological data are used twice over the 1850-1900 period. For the 1901-2017 period, meteorological data for the correct year is used. | |
| 3. LUC+FERT-only | Land cover with increasing crop area, and fertilizer application rates over the crop area | |
| 4. N-DEP-only | N deposition of ammonia and nitrate | |
| 5. FULL | All forcings | |
| 6. FULL-no-LUC | All forcings except increasing crop area | |
| *Other simulations* | | |
| 7. ORIGINAL | All forcings | Runs without N cycle using the original model configuration. |
| 8. ORIG-UNCONST | All forcings but with downregulation turned off | |
| 9. FULL-no-implicit-P-limitation | All forcings but using same $\Gamma_1$ and $\Gamma_2$ globally | Run with N cycle |








**Table 2**: Comparison of simulated global N pools and fluxes, from the FULL simulation, with other
modelling and quasi observation-based studies (references for which are noted as superscripts
and listed below the table). The time-periods to which the other modelling and quasi
observation-based estimates correspond are also noted, where available. The estimates are for
land. Simulated fluxes and pool corresponds to the period 1997-2018.

| N pool and fluxes | This study  (1998-2017) | Other model and quasi observation-based estimates |
|---|---|---|
| *N inputs* (Tg N yr⁻¹) | | |
| BNF | 119 | 118[a]<br>99[b] (2001-2010)<br>138.5[c] (early 1990s)<br>128.9[d] (2000-2009)<br>104-118[e]<br>92[f] (year 2000) |
| Natural BNF | 59 | 58[a]<br>107[c] (early 1990s)<br>30-130[e]<br>39[f] (year 2000) |
| Anthropogenic BNF | 60 | 60[a]<br>31.5[c] (early 1990s)<br>14-89[e]<br>53[f] (year 2000) |
| Fertilizer input | 91 (based on TRENDY protocol) | 100[a]<br>100[b] (2001-2010)<br>100[c] (early 1990s)<br>83[f] (year 2000) |
| N deposition | 66 (based on TRENDY protocol) | 70[a]<br>56-62[b]<br>63.5[c] (early 1990s)<br>69[f] (year 2000) |
| *N pools* (Tg N yr⁻¹) | | |
| Vegetation | 3034 | 1,780[d] (2000s)<br>3,800[g] (1990s)<br>5,300[h]<br>2,940[i] (1990s) |
| Litter and soil | 77161 | 106,000[d] (2000s)<br>100,000[g] (1990s)<br>56,800[h]<br>113,000[i] (1990s) |
| Ammonia | 1924 | 163.7[d] (2000s)<br>361[h]<br>1200[i] (1990s) |
| Nitrate | 2974 | 2,778[d] (2000s)<br>580[h]<br>14,800[i] (1990s) |
| *N fluxes related to N cycling* (Tg N yr⁻¹) | | |
| Plant uptake | 940 | 618[d] (2000s)<br>1,127[g] (1990s)<br>1,084[h]<br>873[i] (1990s) |
| Net mineralization | 947 | |
| Mineralization | 2045 | 1,678[d] (2000s) |
| Immobilization | 1097 | 1,177[d] (2000s) |
| Nitrification | 239 | |





| *N losses* (Tg N yr$^{-1}$) | | | |
|---|---|---|---|
| NO$_3$- Leaching | 53.5 | | 97.1[b] (2001-2010) <br> 62.8[d] (2000s) <br> 77.0[g] (1990s) |
| NH$_3$ Volatilization | 53.9 | | 124.9[b] (2001-2010) <br> 52.6[c] (early 1990s) <br> 20.4[d] (2000s) |
| N$_2$ from denitrification | 114.2 | | 105.8[b] (2001-2010) <br> 68[f] (year 2000) |
| N$_2$O from denitrification | 4.2 | 12.6 | 8.7[b] (2001-2010) |
| N$_2$O from nitrification | 8.4 | | 10.9[c] (early 1990s) <br> 13.0[a] |
| NO from denitrification | 11.4 | 34.3 | 24.8[c] (early 1990s) |
| NO from nitrification | 22.9 | | 26.8[g] (1990s) |


[a]Fowler et al. (2013), [b]Zaehle (2013), [c]Galloway et al. (2004), [d]von Bloh et al. (2018), [e]Galloway et al.
(2013), [f]Bouwman et al. (2013), [g]Zaehle et al. (2010), [h]Xu-Ri and Prentice (2008), [i]Wania et al. (2012)





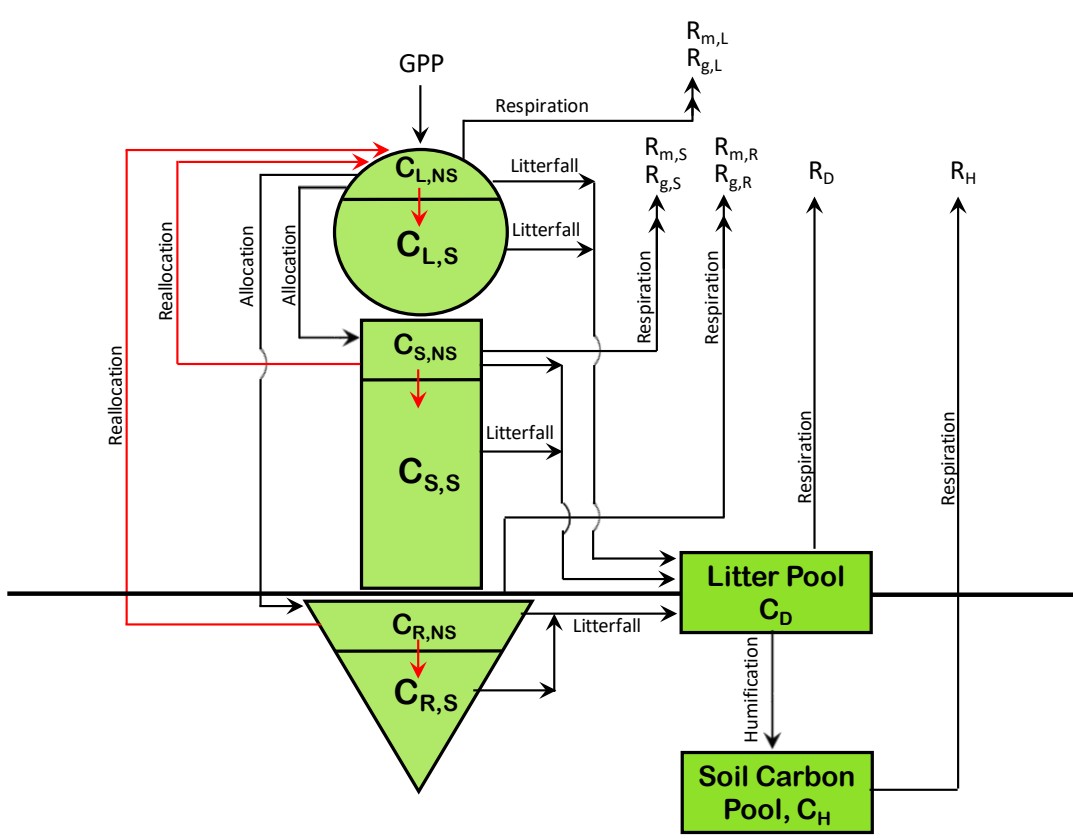

Figure 1: The structure of CLASSIC model used in this study, upon which the N cycle is implemented, with its carbon pools and fluxes. The fluxes of non-structural carbon are shown in red colour.






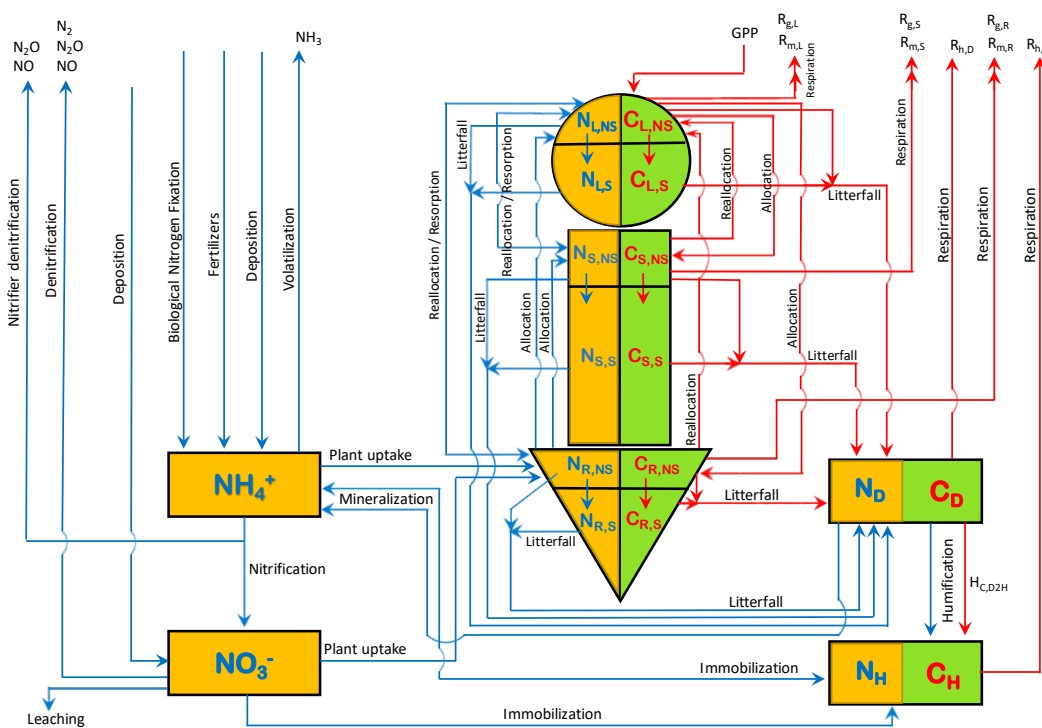


Figure 2: The structure of CLASSIC model used in this study. The eight prognostic carbon pools
are shown in green colour and carbon fluxes in red colour. The ten prognostic nitrogen pools are
shown in orange colour and nitrogen fluxes are shown in blue colour.




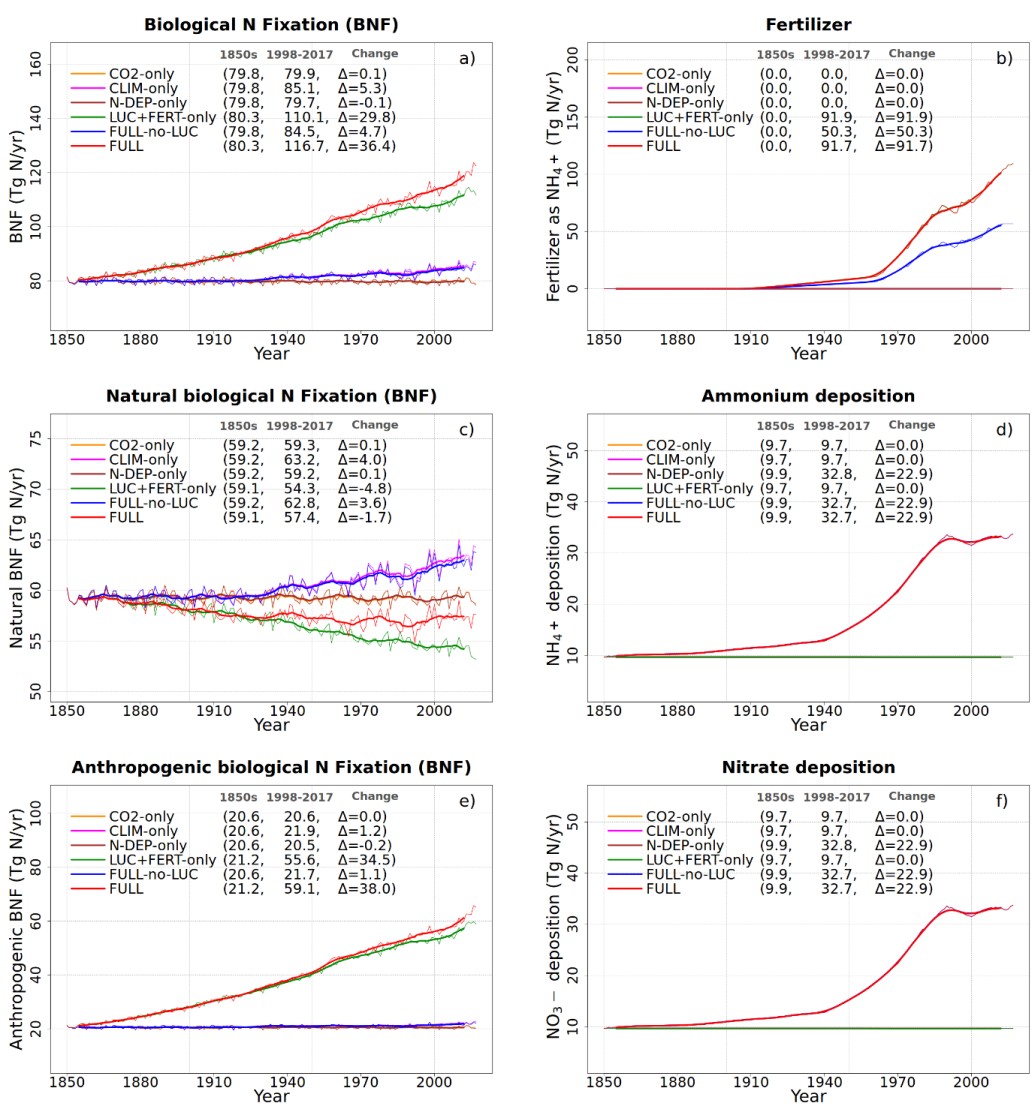


Figure 3: Global annual values of N inputs. Biological N fixation (a) and its break down into natural (c) and anthropogenic components (e). Fertilizer input (b) and atmospheric deposition of ammonium (d) and nitrate (f). The values in the parenthesis for legend entries show averages for the 1850s, the 1998-2017 period, and the change between 1850s and 1998-2017 periods. The thin lines show the annual values and the thick lines their 10-year moving average.




Figure 4: Geographical distribution of annual values of N inputs. Biological N fixation (a) and its break down into natural (c) and anthropogenic components (e). Fertilizer input (b) and atmospheric deposition of ammonium (d) and nitrate (f). The global total values shown are averaged over the 1998-2017 period. The thin lines show the annual values and the thick lines their 10-year moving average.








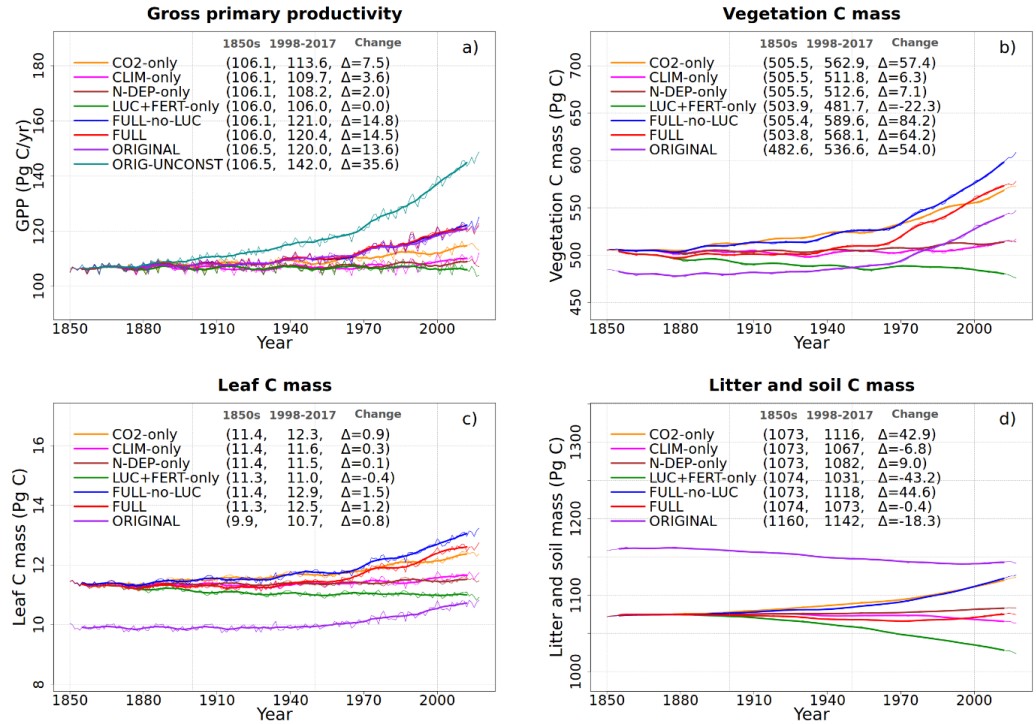


Figure 5: Global annual values of gross primary productivity (a), vegetation carbon (b), leaf
carbon (c), and litter and soil carbon (d) for the primary simulations performed. The values in
the parenthesis for legend entries show averages for the 1850s, the 1998-2017 period, and the
change between 1850s and 1998-2017 periods. The thin lines show the annual values and the
thick lines their 10-year moving average.

1331

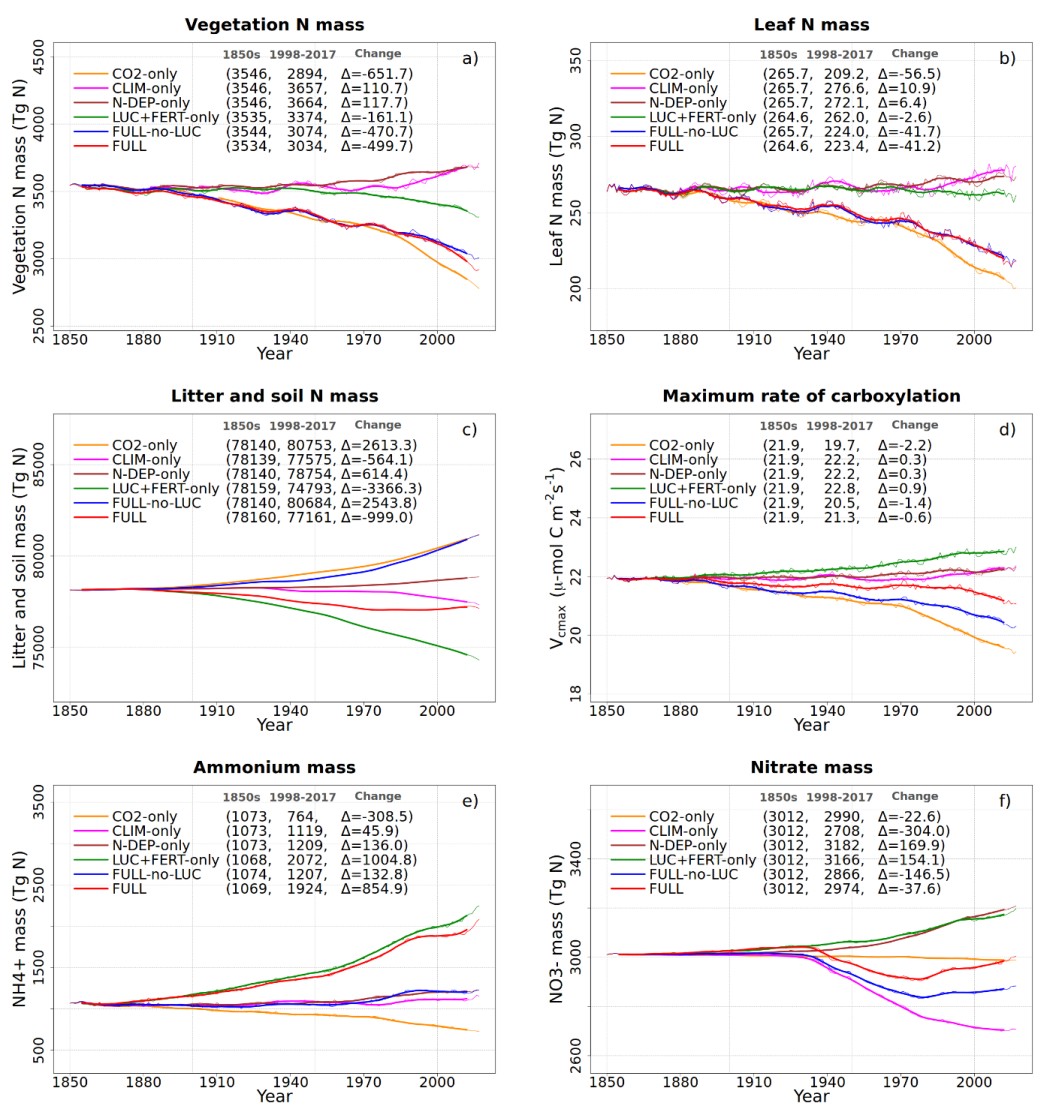

1332

Figure 6: Global annual values of N in vegetation (a), leaves (b), litter and soil organic matter (c) pools, $V_{cmax}$ (d), and ammonium (e), and nitrate (f) pools for the primary simulations performed. The values in the parenthesis for legend entries show averages for the 1850s, the 1998-2017 period, and the change between 1850s and 1998-2017 periods. The thin lines show the annual values and the thick lines their 10-year moving average.

1338


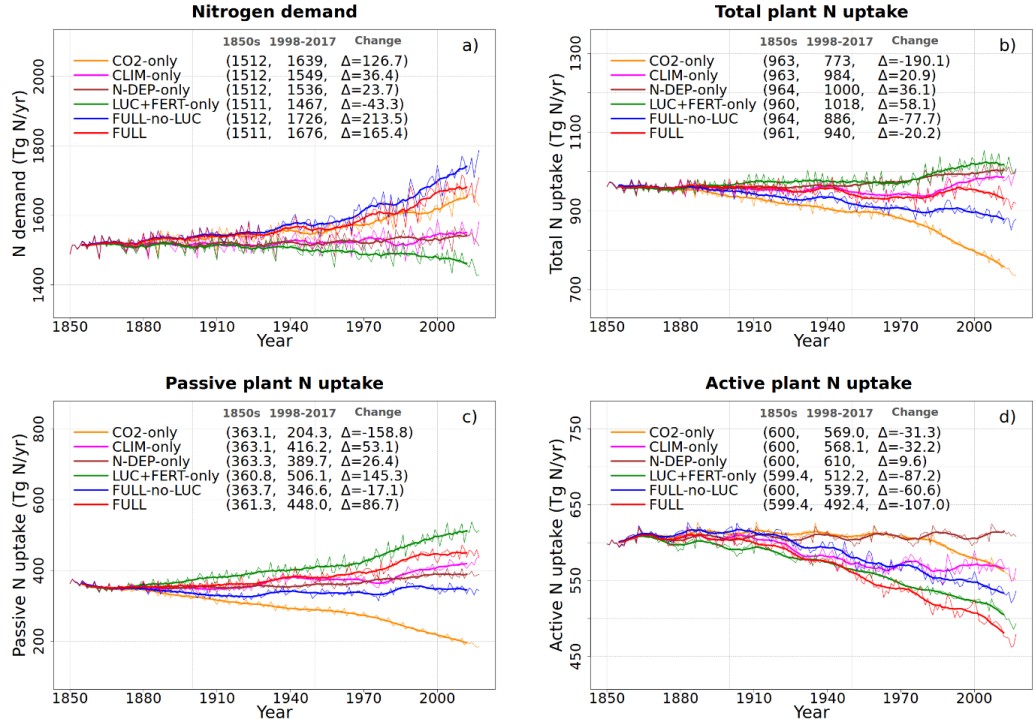

1339

1340

Figure 7: Global annual values of N demand (a), total plant N uptake (b) and its split into passive
(c) and active (d) components for the primary simulations performed. The values in the
parenthesis for legend entries show averages for the 1850s, the 1998-2017 period, and the
change between 1850s and 1998-2017 periods. The thin lines show the annual values and the
thick lines their 10-year moving average.






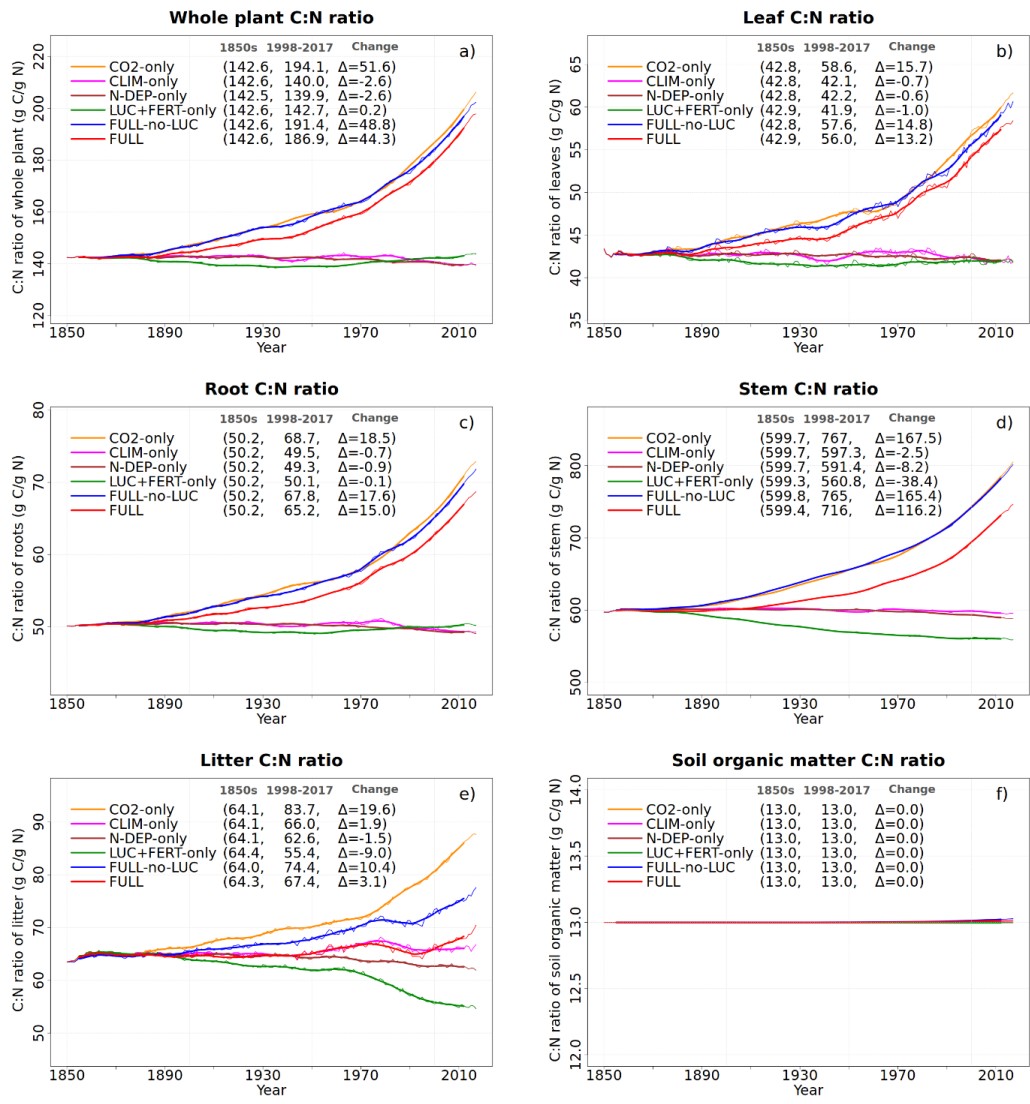


Figure 8: Global annual values of C:N ratios for whole plant (a), leaves (b) , root (c), stem (d),
litter (e) and soil organic matter (f) pools from the primary six simulations. The values in the
parenthesis for legend entries show averages for the 1850s, the 1998-2017 period, and the
change between 1850s and 1998-2017 periods. The thin lines show the annual values and the
thick lines their 10-year moving average.


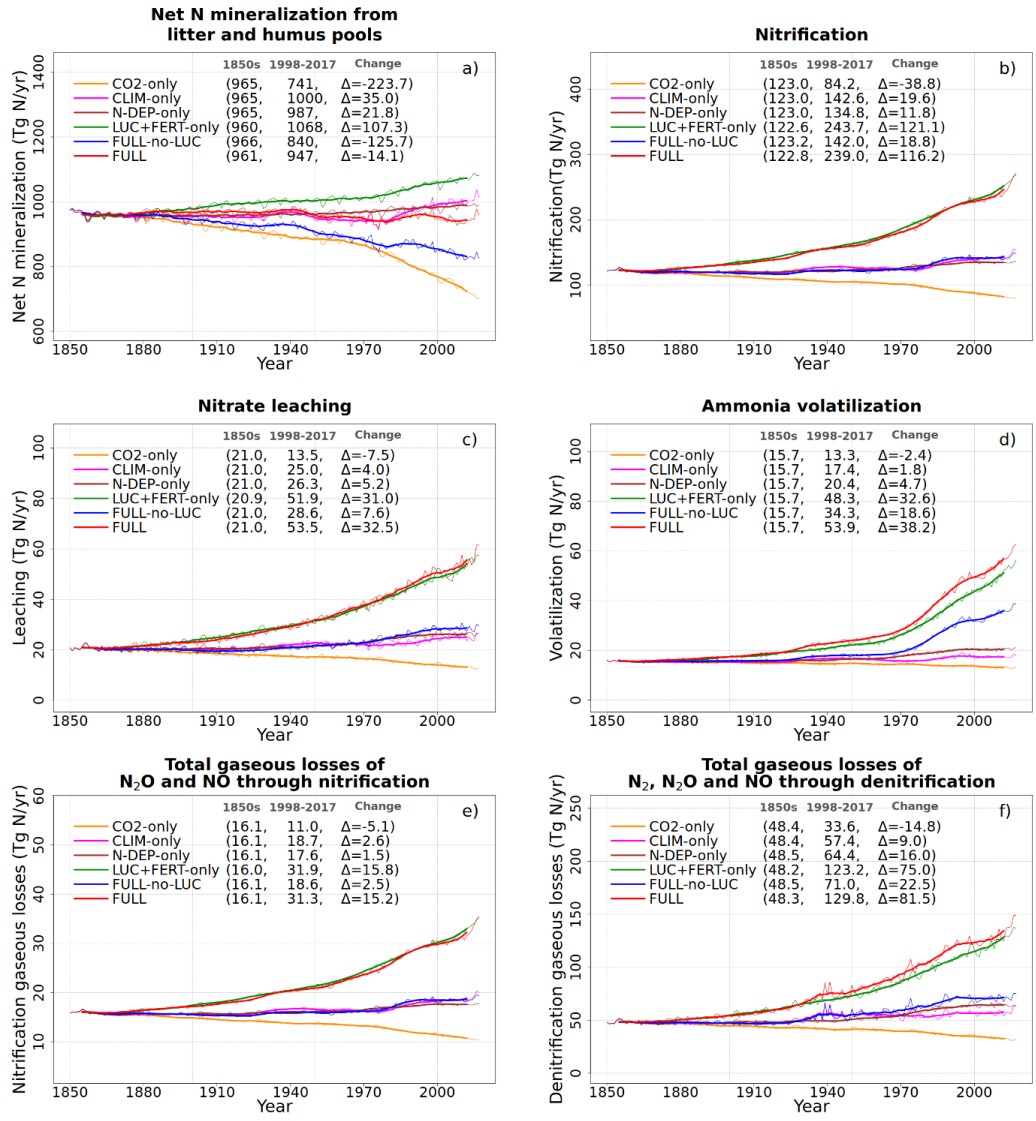


Figure 9: Global annual values of net mineralization (a), nitrification (b), NO₃- leaching (c), NH₃
volatilization (d), and gaseous losses associated with nitrification (e) and denitrification (f) from
the primary six simulations. The values in the parenthesis for legend entries show averages for
the 1850s, the 1998-2017 period, and the change between 1850s and 1998-2017 periods. The
thin lines show the annual values and the thick lines their 10-year moving average.


Figure 10: Geographical distribution of primary C and N pools. Ammonium (a), nitrate (b), vegetation C mass (c), litter and soil C mass (d), vegetation N mass (e), and litter and soil N mass (f). The global total values shown are averaged over the 1998-2017 period.








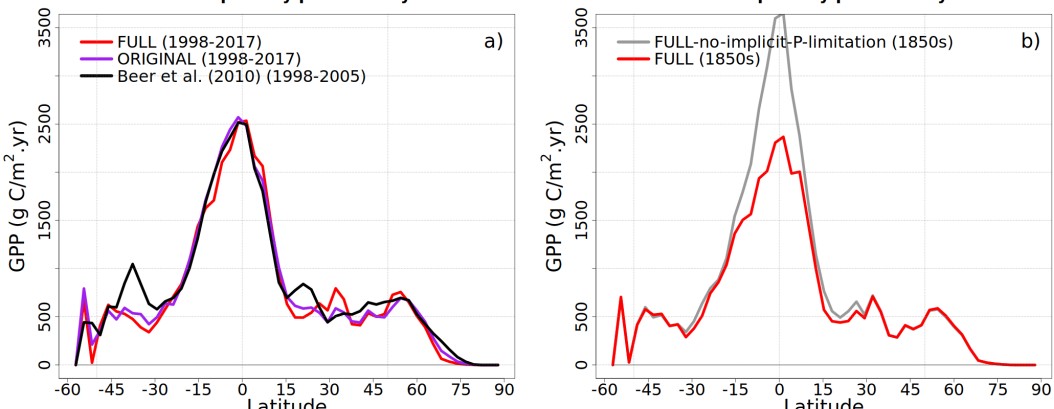


Figure 11: Comparison of zonal distribution of gross primary productivity (GPP). Panel (a) compares zonal distribution of GPP from FULL and ORIGINAL simulations with observation-based estimate from Beer at al. (2010) for the present day. Panel (b) compares the zonal distribution of GPP from the pre-industrial simulation, corresponding to 1850 conditions, from the FULL and FULL-no-implicit-P-limitation simulations to illustrate the effect of not reducing the $\Gamma_1$ parameter for calculating $V_{cmax}$ for the broadleaf evergreen tree PFT that implicitly accounts for phosphorus limitation.









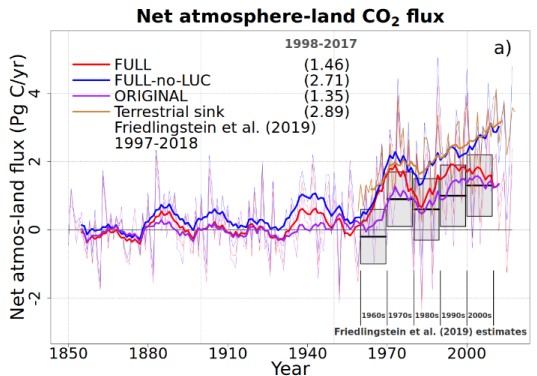

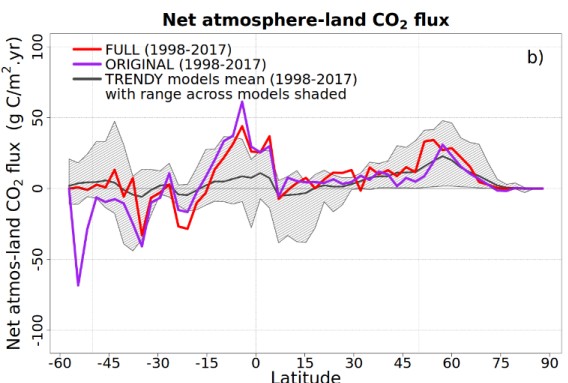

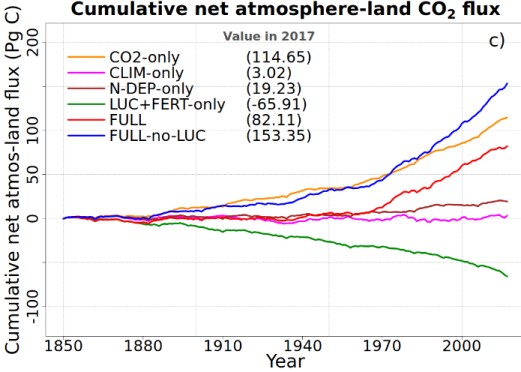


Figure 12: Comparison of simulated net atmosphere-land $CO_2$ flux from various simulations. Panel (a) compares globally-summed values of net atmosphere-land $CO_2$ flux from FULL, FULL-no-LUC simulation, and ORIGINAL simulations with estimate of terrestrial sink (dark yellow line) and net atmosphere-land CO2 flux (grey bars) from Friedlingstein et al. (2019). The thin lines show the annual values and the thick lines their 10-year moving average. Panel (b) compares zonal distribution of net atmosphere-land $CO_2$ flux from FULL and ORIGINAL simulations with the range from TRENDY models that contributed to the Friedlingstein et al. (2019) study. Panel (c) shows cumulative values of net atmosphere-land $CO_2$ flux from the six primary simulations to investigate the contribution of each forcing to the cumulative land carbon sink over the historical period.

1395



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
