# Peer review of "Implementation of nitrogen cycle in the CLASSIC land model"

_Biogeosciences, 2020_

## Referee Comment (RC1) · Anonymous Referee #1 · 8 Jun 2020

The paper introduces a land surface model with a complete prognostic nitrogen cycle. It contributes to the sentiment for a need of nutrient limitation to effectively assess anthropogenic carbon dioxide sequestration in land systems. The paper is well organized, well structured, clearly written and therefore easy and straightforward to read. The results are not surprising in that nitrogen limitation indeed curb carbon accumulation and demonstrate interactions with land-use, nitrogen deposition and climate.

Given that this is one of a growing body of models that carry a prognostic N cycle, I was a little bit disappointed with the depth of the analysis. I suppose these types of analyses are typical and perhaps even expected for the introduction of a coupled C-N model. Yet I miss the placement of this model into the suite of other models. Where do the result differ between this and other models? Where are key implementations

slightly differ than in other models, and what does this mean for the interpretation of the results?

One topic in this direction that comes to my mind is the implementation of downregulation. Clearly, N concentration in leaves lead to a decrease in Vcmax. This is caused by decreased N concentration from increased carbon, as well from an overall decrease in N. Yet, GPP increases owing to the fact of the Farquhar photosynthesis scheme, that increases the efficiency of carbon uptake with higher CO2. Is it done the same way as in other models?

It is not clear among the different sink terms of ammonium and nitrate, how the negotiation e.g. between plant uptake and microbial immobilization works. It look like the soil immobilization outcompetes plants (unfortunately I cannot glean it from the equation in the appendix), and that is plants and other sinks only have access not net mineralization? What are the sink strength of each? What would the result look like if plant have better access to N than Humic soil pool? I believe a discussion of this is central, especially if C:N ratios of the soil pool is held constant at low levels.

Consequences of allocating all GPP (no real downregulation): The way the model treats downregulation is interesting. Vcmax is mentioned, but that is the amount of photosynthesis per unit leaf area. But it seems, leaf mass and thus leaf area increase greatly with increasing CO2. As I understand there is no upper limit for C:N ratios? So this allows for considerable carbon accumulation in vegetation as C:N ratios are widening. This is different to many other models who maintain fixed C:N ratios, or keep them in a certain bound. It also may explain the strong feedback with soil nitrogen availability, where transfer into low fixed C:N ratio causes N immobilization.

With such strong potential for immobilization, there may be a need to discuss microbial immobilization vs. plant uptake competition. This is something the community grapples with and it may be worthwhile to discuss this in the context of your model setup. What if plants outcompete microbes, and have first access to the nitrogen before it fuels

immobilization?

I feel the authors could discuss other efforts to include more mechanistic BNF beyond empirical approaches used here. There are modeling approaches that also make biological sense and are mechanistic to some degree. Please take a look at BNF schemes summarized in Meyerholt (2016), and ideas put forward by Vitousek et al. (2002), and Rastetter et al. (2001), which are congruent with many observations.

Overall, I want to emphasize that the model is conceptually well conceived and described. What I am looking for is a bit more discussion of how the model hypotheses generate these results and how they contrast with other model philosophies.

If there is a need to shorten the paper, I would suggest tightening the description of the physical model. For example: It is not clear how the detailed description of soil layers down to the bedrock links up with the N cycle.

Finally, I see limited value in writing down the budget equation in the method section. The pools and flows of nitrogen are nicely depicted in Figure 2, so the equations just formally describe Figure 2. I think it is more worthwhile to use key equations in the Appendix to describe specific processes.

Detailed comments: Abstract L 35: I would appreciate a bit more tangible sentence rather than agreement. Can it be followed up?

L127 to 155: This paragraph can be shortened. Please consider describing only the mechanisms relevant for the interpretation of this study and perhaps move the rest into the appendix.

Figure 1 is redundant as all the elements in this figure are also shown in figure 2. I know that maybe Figure 2 is a bit busy, but overall, I think the existing model does not need that much of attention.

L293+. The equations in this section describe the tendency of each pool based on the fluxes. This is in my view redundant to Figure 2. I would rather like to see the characterization/equations of key processes. Nitrification, Denitrification, Plant Uptake, BNF
– similar as you described downregulation. Therefore, I ask you to consider swapping
in some of the key equations in the appendix in. That is I would like to see perhaps
equation preferred for the text from 378+

L 396: "is also modeled": Can the author be specific – i.e. constant, or varies depend-
ing on N demand, other mechanisms. I don't require a length explanation, but within
the existing sentence more information can be conveyed.

L420: "The modeled..." This sentence appears to be interpretation – part of the dis-
cussion?

L451: Can you a bit more specific how you determine equilibrium – how many years,
what is the criteria (i.e. what are drifts in total C and N at the modeled equilibrium).

L460: I don't understand what "adjusted to monthly values" means. Can you elaborate,
or are there references?

L472+ : Time varying data and maps of N deposition and fertilizer data is model in-
put, yet it is treated as model output. I am wondering showing its value in the main
manuscript, when its derived from an established protocol and used before. Perhaps
present in method section?

Figure 3: BNF is not shown for CO2 only, I assume the graph is behind "Ndep only"?

Figure 3: I appreciate adding the numbers for global baseline, global current and
change into the figures. Very useful and helpful for the reader!

L606: Sentence with "A reduction..." please reformulate, it is confusing regarding
cause and effects.

L637: On top of BNF, could also increased mineralization (reduced soil pool) contribute
to increased vegetation N pools?

L683: I assume that Vcmax is a per unit leaf area value (not ground area), please

clarify.

L837: I am not sure where the 14% is coming from N:C ratio change from 1/140 to 1/200 Figure 8a, which according to my calculation is ∼30 %.

L843: Please be careful, leaf mass and leaf concentration are not the same thing. In your simulation, there is still C accumulation in leaves owing to $CO_2$ fertilization, while N mass is reduced. This exaggerates decreases in concentration. Looking at C:N ratios, your leaf concentration decreased by 28%

L855: Again, differentiating between pool size and concentration required.

L870: Please elaborate: what is GPP in response to climate vs. GPP in response to temperature.

L901 (entire paragraph). This is a critical observation. Most of the models have an upper limit of C:N ratios for tissues, including leaves. This means that once this level is reached, photosynthesis is capped to a rate that allows maintaining C:N ratios. In contrast, your model allows C:N ratios to widen unconstrainedly. I think this is worthwhile discussion. This has also repercussion for decomposition. A wide C:N ratio in litter locks up more N in soil organic matter with a narrow constant C:N – which in turn limits N supply to vegetation.

L1144: Check the unit for immobilization, it should be g m-2 yr-1, yet the right hand side of the equation has a unit of g m-2. Please clarify.

L1562: Remove capitalization (mistake from reference software?)

No editorial comments: I congratulate the authors for putting this manuscript together so carefully.

References:

Meyerholt, J., S. Zaehle, and M. J. Smith. 2016. Variability of projected terrestrial biosphere responses to elevated levels of atmospheric $CO_2$ due to uncertainty in biological nitrogen fixation. Biogeosciences 13:1491–1518.

Rastetter, E. B., P. M. Vitousek, C. Field, G. R. Shaver, D. Herbert, and G. I. Ågren. 2001. Resource optimization and symbiotic nitrogen fixation. Ecosystems 4:369–388.

Vitousek, P. M., K. Cassman, C. Cleveland, T. Crews, C. B. Field, N. B. Grimm, R. W. Howarth, et al. 2002. Towards an ecological understanding of biological nitrogen fixation. Biogeochemistry 57–58:1–45.

---

## Referee Comment (RC2) · David Wårlind (Referee) · 18 Jun 2020

General comments

Introducing a prognostic nutrient cycle, here the nitrogen cycle, into a land surface model (LSM) is a challenging task. As the importance of nutrient limitation on productivity has been clear for a while and we have gone from one LSM with an N cycle in CMIP5 to several in CMIP6 this is a step all LSM are taking. So for undertaking this task and finishing an LSM that have included all the major N related processes I congratulate the authors. The paper goes through the steps they have taken to incorporate the N cycle processes and show how it behaves during several historical simulations where either all external factors (changes in [CO2], climate, N deposition, LUC, N fer-

tilisation) have been switched on or individual factor have been investigated separately. These simulations have then been analysed on a global scale over the historical period and the authors have come to the conclusion that the model responds to these forcing's are consistent with the conceptual understanding of the coupled C and N cycles. To be able to make such a statement I would expect a more thorough analysis of the model behaviour and also a more sophisticated representation of several of the C-N processes. This is also something you state at the end of the manuscript that many of the processes that have been incorporated have flaws and needs further development. As C-N models have been around for some 30 years now (e.g. Parton et al. 1993) it is a little disappointing that these development hasn't been done before the model goes into publication.

The first thing I miss is a real analysis of how large or small the N limitation actual is in the model. We get no number on how much GPP is limited by N limitation and nothing about if some regions are more limited than others (which we get for the implicit P limitation, Figure 11b). The only comparison we get is between the ORIGINAL and ORIG-UNCONST of 22 Pg C / yr for the period 1998-2017 (figure 5a). ORIGINAL represent N limitation with a globally fixed downregulation of GPP of 6% (line 258), hence the same N limitation everywhere. In lines 260-264 you then state that ORIGINAL is capable of simulating the realistic geographical distribution of GPP that partly comes from the specification of observation-based Vcmax rates. So it is actually Vcmax rates that give the correct GPP distribution. How has the geographical distribution of GPP changed with the N cycle? Latitudinal differences between the new model and ORIGINAL is depicted in Figure 11a where they are very similar. How do I know that the new N cycle limitation on GPP is different from ORIGINAL and N limitation isn't similar everywhere as it is for the ORIGINAL version? I would like to see some kind of analysis of how strong N limitation is in the new model and how the geographical distribution is to be sure that N limitation on GPP is strongest in areas that we expect it to be. This could either be done by turning off the N influence on GPP or in an experiment where you add a huge amount of N everywhere and see where this addition of N has an effect

on GPP. In areas of low N limitation, this additional N input shouldn't affect GPP, but in areas of strong N limitation, it should.

I'm also very disappointed that you have decided to do the study without activating competition between PFTs for space (line 170-171). Not many models have this feature and it would be very interesting to see how the inclusion of C-N interactions would influence the PFT distribution. Also if the distribution and competitive strength between PFTs changes with the N cycle it might be that you need to reparametrize some of N cycle to get an acceptable result, resulting in that your results here are not valid for simulations with active competition.

I also understand it as you have no upper limit to tissue C:N ratios. As tissue C:N ratios are already strongly influenced by historical [CO2] increase (figure 8, Section 5.2.1) I wonder how an RCP8.5 scenario would affect it, as it has a much higher [CO2]? As immobilisation is highly correlated to litter C:N ratios, then with higher C:N ratio immobilization will increase. What happens if the NH4 and NO3 pools can't meet the immobilisation demand in a future [CO2] rich world? In lines line 925-927 you state that with the increase in litter C:N ratios over the historical period litter decomposition rates decreases. Is this due to the increased immobilisation? How does the decrease in decomposition rates work and how much has it increased the soil C pools and where, as was stated in the same section? You state that your litter plus soil C is in the right ballpark compared to Köchy et al. (2015), but your geographical distribution of C is way off if compared to Köchy et al. (2015). How does it come that you have a similar amount of soil C in tropical areas as in boreal regions? Like the forested boreal regions of Russia should have a much higher soil C content than e.g. the amazon. It seems that litter and soil C is very much dictated by GPP (litter input) and not any soil processes (decomposition rates should be much slower in cold regions). Also your assumption of a constant soil C:N ratio (line 901-902) is not in line with Köchy et al. (2015) estimates. In lines 908-910 you mention the idea that soil C:N ratio should be dependent on PFT, I would sayÂăit should mostly be influenced by litter input C:N ratios.

Including a BNF representation in an LSM is one of the trickiest parts. There is no real good way of doing it that I have seen in any LSM. Most LSMs use some kind of empirical relationship which you can parameterise to get global BNF number in the right region compared to estimates. So my personal opinion when it comes to modelling BNF is then to get the right BNF response to different kinds of manipulations. One being that BNF should decrease as you add N to the system according to the meta-analysis of Zheng et al. (2019). You mention on lines 898-900 that the BNF formulation will be revised. Make sure then to capture the right responses to changing conditions.

Specific comments

L48-50: "Over land, the uptake of carbon in response to increasing anthropogenic CO2 emissions is driven by two primary factors, 1) the CO2 fertilization of the terrestrial biosphere, and 2) the increase in temperature, both of which are associated with increasing [CO2]." – How about water? How do you say that these two are the primary factors? What do you base it on? Later you write that photosynthesis can't occur without water or nutrients.

L68-72: "McGuire et al. (1995) define downregulation as a decrease in photosynthetic capacity of plants grown at elevated CO2 in comparison to plants grown at baseline CO2, although the rate of photosynthesis for plants grown and measured at elevated CO2 is still higher than the rate for plants grown and measured at baseline CO2." - This sentence doesn't make any sense, and you try and explain why in the following part of the section. But why have this at all. Why not give a proper definition of downregulation directly?

L81-83: "Comparison of land and ocean carbon uptake in C4MIP studies (Friedlingstein et al., 2006; Arora et al., 2013, 2019) indicate that the future land carbon uptake across ESMs varies widely and more than three times as much for the ocean carbon uptake." – Revise this sentence as it is unclear.

L162-164: "The biogeochemical module of CLASSIC uses this information along with

air temperature to simulate photosynthesis and prognostically calculates amount of carbon in the model's three live (leaves, stem, and root) and two dead (litter and soil) carbon pools." - Do litter and SOM exist in each soil layer or only in one total pool each? And am I assuming right that each fractional vegetation cover (needleleaf trees, broadleaf trees, crops, and grasses) has their own litter and soil C and N cycle? Not really clear in the manuscript.

L211-213: "Finally, root biomass is used to calculate rooting depth and distribution which determines the fraction of roots in each soil layer" – How does the fine root distribution between soil layers affect the N uptake capacity?

L232-259: The whole section on how the parameterizing of the downregulation of photosynthesis with increasing [CO2] for emulating nutrient constraints has been done is not needed as I guess it is described elsewhere.

Section 3.1: Whole section (Eqn 4-14) is more or less redundant as it is described in figure 2. Figure 1 is also redundant as it is present in figure 2. Better to have all eqn and description in the appendix here as they are the interesting part of the manuscript.

L421-423: "The modelled differences in PFT specific values of Vcmax, in our framework, come through differences in simulated N_L values that depend on BNF, given that BNF is the primary natural source of N input into the coupled soil-vegetation system." - Over time BNF is the main source of N to the system, but for growth in a single year/day SOM mineralisation releases much more N (figure 3a vs 9a).

L449-451: "The simulation for the pre-industrial period uses forcings that correspond to year 1850 and the model is run for thousands of years until its C and N pools come into equilibrium." - How is the initial zero N world handled in the beginning of the spin-up? Have you considered any approaches to speed up the spin-up? One approach could be to decouple the SOM pool module with constant saved litter input and soil conditions and run it for a standalone spin-up.

L490-492: "Finally, a separate pre-industrial simulation is also performed that uses the same $\Gamma 1$ and $\Gamma 2$ globally (FULL-no-implicit-P-limitation). This simulation is used to illustrate the effect of neglecting P limitation for the broadleaf evergreen tree PFT in the tropics." - Not only the tropics that is P limited (Du et al. 2020)

L621-623: "This results in a slight increase in vegetation and leaf N mass (Figures 6a and 6b) and the $NH_4^+$ (Figure 6e) pool which is the primary mineral pool in soils under vegetated regions." - How can it be that we have a total pool of NH4 and NO3 of around 4 Pg N (figure 6) at the end of the year and N demand of 1.5Pg N (figure 7). How can there be any limitations?

L635-638: "The increase in GPP due to changing climate increases the N demand (Figure 7a, magenta line) but unlike the CO2-only simulation, the plant N uptake increases since the $NH_4^+$ and $NO_3^-$ pools increase in size over the vegetated area in response to increased BNF (Figure 3a, magenta line)." - BNF increases with 5.3 Tg N / yr whereas net N mineralisation increases with 35 Tg / yr. Why do you address the increase in plant N uptake to BNF and not increases in net mineralisation?

L644-646: "The litter C:N, in contrast, shows a small increase since not all N makes its way to the litter as a fraction of leaf N is resorbed from deciduous trees leaves prior to leaf fall (Figure 8e)." - This needs a better explanation. Is the resorption fraction dynamic or is the deciduous PFT more dominant (competition between PFTs turned off, sadly)? Don't understand how this can happen.

L676-679: "An increase in crop area over the historical period results in deforestation of natural vegetation that reduces vegetation biomass but also soil carbon mass, since a higher soil decomposition rate is assumed over cropland area (Figures 5b and 5d), consistent with empirical measurements (Wei et al., 2014)." - How is this modelled? Needs an explanation. How do fig 5b and 5d show that the decomposition rate has increased? Litter input has maybe decreased but decomposition rates changed?

L714-717: "The increase in global $NH_4^+$ mass (Figure 6e) in the FULL simulation is

driven primarily by the increase in fertilizer input while the changes in NO3- mass are the net result of all forcings with no single forcing dominating the response." - This needs to be different for Crop and Natural tiles. Needs some discussion.

L720-723: "The increase in the C:N ratio of vegetation (Figure 8a) and its components (leaves, stem, and root) is driven primarily by an increase in atmospheric CO2. Changes in litter C:N in the FULL simulation, in contrast, do not experience dominant influence from any one of the forcings." - This needs some kind of explanation. How can this happen? Same as for L644-646.

L723-726: "The simulated change in net N mineralization (Figure 9a) in the FULL simulation, over the historical period, is small since the decrease in net N mineralization due to increasing CO2 is compensated by the increase caused by changes in climate, N deposition, and fertilizer inputs." - Has to be a large difference between Natural and Crop tiles. Needs to be shown.

L844: "The response of our model to elevated CO2" - Very hard to draw any conclusion in respect to elevated [CO2] from your experiments. Elevated [CO2] is usually referred to as a CO2 concentration that is higher than the present.

L865-867: "Soil carbon mass, however, decreases (despite increase in NPP inputs) since warmer temperatures also increase heterotrophic respiration (not shown)." - I wouldn't call a decrease of 0.4 Pg compared to a total pool of 1074 Pg a decrease. To say that you get a realistic response.

L986-990: "For simplicity, we assume fertilizer is applied at the same daily fertilizer application rate (gN m–2 day–1) throughout the year in the tropics (between 30°S and 30°N), given the possibility of multiple crop rotations in a given year. Between the 30° and 90° latitudes in both northern and southern hemispheres, we assume that fertilizer application starts on the spring equinox and ends on the fall equinox." - Using hard limits in a model like this will create problems. Wouldn't it be possible to do this a little more prognostic? Base it on PFT distribution and productivity? Changing climate?

Having a fixed limit like this won't adapt to a changing climate.

L1000-1001: "The modelled plant N uptake is a function of its N demand. Higher N demand leads to higher mineral N uptake from soil." - So it is independent on the amount of roots?

Section A2.6: What are the decay rate constants for litter and soil pools? Have they changed after the inclusion of the N cycle?

Section A3.3: Shouldn't the amount of NO3 in the bottom layer be estimated in a similar manner as for nitrification/denitrification/volatilization? Denitrification assumes that all NO3 is in the top 50cm.

Figure 5d: How is the split between litter and soil C?

Figure A.1: Explain what the two lines represent

Table A1: Header text is unclear

Technical corrections

L511: "FIRE-only" – remove

L779: remove "two"

References

Du et al. 2020 - doi.org/10.1038/s41561-019-0530-4

Köchy et al. 2015 - doi:10.5194/soil-1-351-2015

Parton et al. 1993 - doi.org/10.1029/93GB02042

Zheng et al. 2019 - doi:10.1111/gcb.14705

---

## Author Comment (AC1) · 21 Jul 2020

**We thank both reviewers for their comments and are grateful for the opportunity to respond. In the following pages, we have answered reviewer #1's questions and proposed how we will address his/her comments if given the opportunity to revise our manuscript.**

Reviewer #1

The paper introduces a land surface model with a complete prognostic nitrogen cycle. It contributes to the sentiment for a need of nutrient limitation to effectively assess anthropogenic carbon dioxide sequestration in land systems. The paper is well organized, well structured, clearly written and therefore easy and straightforward to read. The re-

sults are not surprising in that nitrogen limitation indeed curb carbon accumulation and demonstrate interactions with land-use, nitrogen deposition and climate. Given that this is one of a growing body of models that carry a prognostic N cycle, I was a little bit disappointed with the depth of the analysis. I suppose these types of analyses are typical and perhaps even expected for the introduction of a coupled C-N model. Yet I miss the placement of this model into the suite of other models. Where do the result differ between this and other models? Where are key implementations slightly differ than in other models, and what does this mean for the interpretation of the results?

**We thank reviewer #1 for his/her overall positive comments. We agree with reviewer's comment that placing our model parameterizations in context of existing models will help a reader. We will do this for all primary processes when revising our manuscript.**

One topic in this direction that comes to my mind is the implementation of downregulation. Clearly, N concentration in leaves lead to a decrease in Vcmax. This is caused by decreased N concentration from increased carbon, as well from an overall decrease in N. Yet, GPP increases owing to the fact of the Farquhar photosynthesis scheme, that increases the efficiency of carbon uptake with higher CO2. Is it done the same way as in other models?

**Different models parameterize Vcmax in different ways as a function of leaf N content. Some parameterize it as a function of leaf N content directly (Zaehle and Friend, 2010, and von Bloh et al., 2018) and some as a function of leaf C:N ratio (e.g., Cox, 2001, and Wania et al., 2012). We found that the latter approach results in an incorrect seasonal variation of Vcmax since C:N ratio of leaves increases during growing season which leads to reduction in Vcmax in contrast to observations which show an increase in Vcmax during the growing season (e.g., see Fig. 1a of Bauerle et al., 2012). The essence is the same that Vcmax has to be able to vary with changes in leaf N. We will place our parameterization in context of existing approaches when revising our manuscript.**

It is not clear among the different sink terms of ammonium and nitrate, how the negotiation e.g. between plant uptake and microbial immobilization works. It look like the soil immobilization outcompetes plants (unfortunately I cannot glean it from the equation in the appendix), and that is plants and other sinks only have access not net mineralization? What are the sink strength of each? What would the result look like if plant have better access to N than Humic soil pool? I believe a discussion of this is central, especially if C:N ratios of the soil pool is held constant at low levels.

**The reviewer is correct that in the current framework, and the order in which calculations are performed imply that, immobilization gets priority over N uptake by plants. Please note here that over the long term it is this feature of the model that locks up N in the soil organic matter and yields the desired downregulation of photosynthesis. Giving plants a priority in accessing mineralized N over the long term may not yield the desired downregulation. Although we do discuss the implications of constant C:N ratio of soil organic matter in our manuscript, we will modify that discussion to reflect this related aspect as well.**

**There is also another subtle point in this context. Let's assume that plant N uptake is given priority over immobilization during a given year. As a result, plant C:N ratio will decrease, and relatively more N enriched litter will be generated which will enter the soil and get locked up in the soil organic matter reducing the need for immobilization in the next year. So regardless of whether immobilization or plant N uptake gets a priority, over the long term N will get locked up in the soil organic matter given the assumption of its constant C:N. The fact that eventually all NPP has to become litter implies that all N has to return to the soil organic matter where it will get locked up.**

Consequences of allocating all GPP (no real downregulation): The way the model treats downregulation is interesting. Vcmax is mentioned, but that is the amount of photosynthesis per unit leaf area. But it seems, leaf mass and thus leaf area increase greatly with increasing CO2. As I understand there is no upper limit for C:N ratios?

So this allows for considerable carbon accumulation in vegetation as C:N ratios are widening. This is different to many other models who maintain fixed C:N ratios, or keep them in a certain bound. It also may explain the strong feedback with soil nitrogen availability, where transfer into low fixed C:N ratio causes N immobilization.

**Thanks you for your attention to this point. Indeed, we have upper C:N ratios for leaves, stem, and root components for all model PFTs. Over the simulated historical period, however, these thresholds are generally not crossed. When a given plant component of a PFT reaches this threshold then at the next model time step GPP is constrained to limit the C:N ratio of newly sequestered biomass such that it doesn't exceed the max C:N ratio. We didn't mention this aspect of the model and will report the upper C:N ratio limits and details of this processes in the revised manuscript.**

With such strong potential for immobilization, there may be a need to discuss microbial immobilization vs. plant uptake competition. This is something the community grapples with and it may be worthwhile to discuss this in the context of your model setup. What if plants outcompete microbes, and have first access to the nitrogen before it fuels immobilization?

**As mentioned above in the current model structure preference is given to immobilization over plant N uptake and this process is key to obtaining the downregulation due to N limitation.**

I feel the authors could discuss other efforts to include more mechanistic BNF beyond empirical approaches used here. There are modeling approaches that also make biological sense and are mechanistic to some degree. Please take a look at BNF schemes summarized in Meyerholt (2016), and ideas put forward by Vitousek et al. (2002), and Rastetter et al. (2001), which are congruent with many observations.

**Thank you for pointing these references which discuss BNF in the light of the costs and benefits of N uptake vs N fixation. We will discuss these approaches**

**in revising our manuscript.**

Overall, I want to emphasize that the model is conceptually well conceived and described. What I am looking for is a bit more discussion of how the model hypotheses generate these results and how they contrast with other model philosophies.

**Thank you. We agree that the manuscript will benefit from additional discussion and revise our manuscript accordingly.**

If there is a need to shorten the paper, I would suggest tightening the description of the physical model. For example: It is not clear how the detailed description of soil layers down to the bedrock links up with the N cycle.

**We struggled with partitioning the manuscript text between the main text and appendix. As both reviewers have suggested reorganization of the text from appendix to main text and vice-versa, we will do this when revising our manuscript.**

Finally, I see limited value in writing down the budget equation in the method section. The pools and flows of nitrogen are nicely depicted in Figure 2, so the equations just formally describe Figure 2. I think it is more worthwhile to use key equations in the Appendix to describe specific processes.

**Thank you for your feedback. We will move the budget equations into the appendix when reorganizing the manuscript text.**

Detailed comments: Abstract L 35: I would appreciate a bit more tangible sentence rather than agreement. Can it be followed up?

**Yes, we will expand on why the model response is consistent with expectations.**

L127 to 155: This paragraph can be shortened. Please consider describing only the mechanisms relevant for the interpretation of this study and perhaps move the rest into the appendix.

**We agree to move the description of the physical model to the appendix along**

with other reorganization of the manuscript.

Figure 1 is redundant as all the elements in this figure are also shown in figure 2. I know that maybe Figure 2 is a bit busy, but overall, I think the existing model does not need that much of attention. L293+. The equations in this section describe the tendency of each pool based on the fluxes. This is in my view redundant to Figure 2. I would rather like to see the characterization/equations of key processes. Nitrification, Denitrification, Plant Uptake, BNF similar as you described downregulation. Therefore, I ask you to consider swapping in some of the key equations in the appendix in. That is I would like to see perhaps equation preferred for the text from 378+.

**Agreed. We will remove Figure 1, move the budget equations to the appendix, and move the process parameterizations to the main text.**

L 396: "is also modeled": Can the author be specific – i.e. constant, or varies depending on N demand, other mechanisms. I don't require a length explanation, but within the existing sentence more information can be conveyed.

**This will become more clear after the text is reorganized with the detailed description of all processes and their parameterizations moved to the main text.**

L420: "The modeled: : :" This sentence appears to be interpretation – part of the discussion?

**Agreed, we will move this sentence to the discussion section.**

L451: Can you a bit more specific how you determine equilibrium – how many years, what is the criteria (i.e. what are drifts in total C and N at the modeled equilibrium).

**Global thresholds of net atmosphere-land C flux of 0.05 Pg/yr and net atmosphere-land N flux of 0.5 Tg N/yr are used to ensure the model pools have reached equilibrium. When modelled fluxes are less than these thresholds, model pools vary very little over time. We will mention these in the revised manuscript.**

L460: I don't understand what "adjusted to monthly values" means. Can you elaborate, or are there references?

**The CRUJRA meteorological data set is a blended product based on the 6 hourly Japanese reanalysis (JRA) and the Climate Research Unit's (CRU) observation-based monthly data (which are in turn based on ground based stations). Since reanalysis data typically do match observations, they are adjusted such that their monthly means/sums for various meteorological variables match the CRU data. This yields the fine temporal resolution that comes from the reanalysis and monthly means/sums that match the CRU data to yield a meteorological product that can be used by models that require sub-daily or daily meteorological forcing.**

L472+ : Time varying data and maps of N deposition and fertilizer data is model input, yet it is treated as model output. I am wondering showing its value in the main manuscript, when its derived from an established protocol and used before. Perhaps present in method section?

**Agreed we will move N deposition and fertilizer to section 4.1 so that Section 5.1 will report just the actual model results.**

Figure 3: BNF is not shown for CO2 only, I assume the graph is behind "Ndep only"?

**Yes, the "CO2-only" curve is hidden behind the "N-Dep-only" curve, as it can be inferred from the mean values for 1850s and 1998-2017 indicate. We will make this clear in the figure caption when revising our manuscript.**

Figure 3: I appreciate adding the numbers for global baseline, global current and change into the figures. Very useful and helpful for the reader!

**Thank you! That was the intention.**

L606: Sentence with "A reduction: : :" please reformulate, it is confusing regarding cause and effects.

**Thanks. We will rephrase this sentence.**

L637: On top of BNF, could also increased mineralization (reduced soil pool) contribute to increased vegetation N pools?

**Yes, in addition to BNF, increased mineralization also contributes to increasing the size of N mineral pools and therefore vegetation N pools. We will mention this in the revised manuscript.**

L683: I assume that Vcmax is a per unit leaf area value (not ground area), please clarify.

**Yes. Vcmax is per unit leaf area and we will clarify this in the revised manuscript.**

L837: I am not sure where the 14% is coming from N:C ratio change from 1/140 to 1/200 Figure 8a, which according to my calculation is 30 %.

**Actually, we calculated this using Figure 6a which reports the absolute N in Tg N. Percentages of ratios, like the C:N ratio, are always tricky. Based on Figure 6a, red line for the FULL simulation, vegetation N content reduces from 3534 Tg N for 1850s to 3034 Tg N for the period 1998-2017 which is a 14.14% reduction. We will clarify this calculation in revising our manuscript.**

L843: Please be careful, leaf mass and leaf concentration are not the same thing. In your simulation, there is still C accumulation in leaves owing to $CO_2$ fertilization, while N mass is reduced. This exaggerates decreases in concentration. Looking at C:N ratios, your leaf concentration decreased by 28%

**Thanks for pointing this out. Yes, we will ensure that all comparisons are consistent. Typically, since concentrations are a ratio, it can be misleading to report percentage change in a ratio.**

L855: Again, differentiating between pool size and concentration required.

**Noted.**

L870: Please elaborate: what is GPP in response to climate vs. GPP in response to temperature.

**The words climate and temperature were used synonymously in this sentence. We will reword this sentence.**

L901 (entire paragraph). This is a critical observation. Most of the models have an upper limit of C:N ratios for tissues, including leaves. This means that once this level is reached, photosynthesis is capped to a rate that allows maintaining C:N ratios. In contrast, your model allows C:N ratios to widen unconstrainedly. I think this is worthwhile discussion. This has also repercussion for decomposition. A wide C:N ratio in litter locks up more N in soil organic matter with a narrow constant C:N – which in turn limits N supply to vegetation.

**As mentioned above we do have an upper limit on modelled C:N ratios although this limit is generally not reached over the historical simulation. We will include these limits and the discussion around how the upper limit is maintained in our modelling framework when revising our manuscript.**

L1144: Check the unit for immobilization, it should be g m-2 yr-1, yet the right hand side of the equation has a unit of g m-2. Please clarify.

**Thank you for noticing this. Yes, we are missing a per unit time constant in this equation. The implied value of this constant is 1 per day since the model time step is one day for its biogeochemical processes. We will clarify this.**

L1562: Remove capitalization (mistake from reference software?)

**Noted.**

No editorial comments: I congratulate the authors for putting this manuscript together so carefully.

**References**

Bauerle, W. L., Oren, R., Way, D. A., Qian, S. S., Stoy, P. C., Thornton, P. E., Bowden, J. D., Hoffman, F. M., Reynolds, R. F.: Photoperiodic regulation of the seasonal pattern of photosynthetic capacity and the implications for carbon cycling. Proc. Natl. Acad. Sci., USA, 109, 8612–8617, 2012.

Cox, P. M.: Description of the TRIFFID dynamic global vegetation model, Tech. Rep. 24, Hadley Centre, Met office, London Road, Bracknell, Berks, RG122SY, UK, 2001.

von Bloh, W., Schaphoff, S., Müller, C., Rolinski, S., Waha, K., and Zaehle, S.: Implementing the nitrogen cycle into the dynamic global vegetation, hydrology, and crop growth model LPJmL (version 5.0), Geosci. Model Dev., 11, 2789–2812, https://doi.org/10.5194/gmd-11-2789-2018, 2018.

Wania, R., Meissner, K. J., Eby, M., Arora, V. K., Ross, I. and Weaver, A. J.: Carbon-nitrogen feedbacks in the UVic ESCM, Geosci. Model Dev., 5(5), 1137–1160, doi:10.5194/gmd-5-1137-2012, 2012.

Zaehle, S., and Friend, A. D.: Carbon and nitrogen cycle dynamics in the O-CN land surface model: 1. Model description, site‐scale evaluation, and sensitivity to parameter estimates. Global Biogeochemical Cycles, 24, GB1005, 2010. https://doi.org/10.1029/2009GB003521.
* * *

---

## Author Comment (AC2) · 21 Jul 2020

**We thank both reviewers for their comments and are grateful for the opportunity to respond. In the following pages, we have answered reviewer #2's questions and proposed how we will address their comments if given the opportunity to revise our manuscript.**

Reviewer #2

General comments

Introducing a prognostic nutrient cycle, here the nitrogen cycle, into a land surface model (LSM) is a challenging task. As the importance of nutrient limitation on productivity has been clear for a while and we have gone from one LSM with an N cycle in

[Figure]

CMIP5 to several in CMIP6 this is a step all LSM are taking. So for undertaking this task and finishing an LSM that have included all the major N related processes I congratulate the authors. The paper goes through the steps they have taken to incorporate the N cycle processes and show how it behaves during several historical simulations where either all external factors (changes in [CO2], climate, N deposition, LUC, N fertilisation) have been switched on or individual factor have been investigated separately. These simulations have then been analysed on a global scale over the historical period and the authors have come to the conclusion that the model responds to these forcing's are consistent with the conceptual understanding of the coupled C and N cycles. To be able to make such a statement I would expect a more thorough analysis of the model behaviour and also a more sophisticated representation of several of the C-N processes. This is also something you state at the end of the manuscript that many of the processes that have been incorporated have flaws and needs further development. As C-N models have been around for some 30 years now (e.g. Parton et al. 1993) it is a little disappointing that these development hasn't been done before the model goes into publication.

**We thank David Wårlind for his critical comments. Inclusion of N cycle in a land surface scheme is a challenging task, as the reviewer himself notes. Modelling is always done in increments. Models become complex over time due both to increased scientific understanding of the processes (contributed by the broader scientific community) and an improved understanding of the models' behaviour themselves. We respectfully disagree that a model structure and processes should be as complex as possible at the early stages. Instead, we believe that as long as the broad conceptual structure of a model is reasonable and its limitations are addressed, it lays the groundwork for further model development. For example, the wide variety of climate models out there has been illustrated as a healthy aspect of the community. In contrast, if all models were to be based on the same parameterizations there would be very little model diversity.**

The first thing I miss is a real analysis of how large or small the N limitation actual is in the model. We get no number on how much GPP is limited by N limitation and nothing about if some regions are more limited than others (which we get for the implicit P limitation, Figure 11b). The only comparison we get is between the ORIGINAL and ORIG-UNCONST of 22 Pg C / yr for the period 1998-2017 (figure 5a). ORIGINAL represent N limitation with a globally fixed downregulation of GPP of 6% (line 258), hence the same N limitation everywhere. In lines 260-264 you then state that ORIGINAL is capable of simulating the realistic geographical distribution of GPP that partly comes from the specification of observation-based Vcmax rates. So it is actually Vcmax rates that give the correct GPP distribution. How has the geographical distribution of GPP changed with the N cycle? Latitudinal differences between the new model and ORIGINAL is depicted in Figure 11a where they are very similar. How do I know that the new N cycle limitation on GPP is different from ORIGINAL and N limitation isn't similar everywhere as it is for the ORIGINAL version? I would like to see some kind of analysis of how strong N limitation is in the new model and how the geographical distribution is to be sure that N limitation on GPP is strongest in areas that we expect it to be. This could either be done by turning off the N influence on GPP or in an experiment where you add a huge amount of N everywhere and see where this addition of N has an effect on GPP. In areas of low N limitation, this additional N input shouldn't affect GPP, but in areas of strong N limitation, it should.

**While the manuscript shows the effect of N limitation at the global scale on realized GPP (in Figure 5, the GPP increase in the FULL simulation is 14.5 PgC/yr compared to 35.6 PgC/yr in the ORIG-UNCONST simulation), we agree with the reviewer that showing the geographical or zonal distribution of N limitation will be helpful for readers. A small caveat here is that the unconstrained GPP also depends on specified Vcmax rates so if original Vcmax rates are too low or too high the GPP ratios will reflect that. Regardless, we agree that such a plot will be helpful and we will add it when revising our manuscript. Also, as the reviewer noted, in the original version of the model, parameterized downregulation is specified**

**as a constant globally.**

**In regards to whether correct geographical distribution of GPP in the original model comes from specification of Vcmax, we would like to emphasize that this is not completely the case. Specified Vcmax rates for the 9 PFTs in CLASSIC vary by only about 2 times, from about 35 to 75 u-mol CO2 m-2 s-1, whereas GPP in the model varies from zero in the Sahara desert to about 3000 gC m-2 year-1 in the Amazonian rainforests. As already mentioned in the manuscript, the climate still plays a primary role in determining geographical distribution of GPP. This is additionally corroborated by Figure 1 (at the end of this response) which shows geographical distribution of GPP for the period 1998-2017 from the FULL, ORIGINAL, and ORIG-UNCONST simulations.**

I'm also very disappointed that you have decided to do the study without activating competition between PFTs for space (line 170-171). Not many models have this feature and it would be very interesting to see how the inclusion of C-N interactions would influence the PFT distribution. Also if the distribution and competitive strength between PFTs changes with the N cycle it might be that you need to reparametrize some of N cycle to get an acceptable result, resulting in that your results here are not valid for simulations with active competition.

**As mentioned above, modelling is a challenging exercise. This is especially true when there are multiple interacting processes. For example, the response of a C cycle model without N cycle interactions, to any perturbation, is relatively easier to interpret than when C and N cycles are interactive. The reason we performed multiple simulations and perturbed the model with only one forcing at a time was to be able to interpret the model behaviour in a straightforward manner. While interesting, had we turned on the competition between PFTs we would have had to run simulations with and without competition to evaluate the model response to a given forcing without interacting feedbacks from competition between PFTs. However, we plan to perform such an analysis in future.**

I also understand it as you have no upper limit to tissue C:N ratios. As tissue C:N ratios are already strongly influenced by historical [CO2] increase (figure 8, Section 5.2.1) I wonder how an RCP8.5 scenario would affect it, as it has a much higher [CO2]? As immobilisation is highly correlated to litter C:N ratios, then with higher C:N ratio immobilization will increase. What happens if the NH4 and NO3 pools can't meet the immobilisation demand in a future [CO2] rich world?

**This issue is raised by reviewer #1 as well. Indeed, we have upper C:N ratios for leaves, stem, and root components for all model PFTs. Over the simulated historical period, however, these thresholds are generally not crossed. When a given plant component of a PFT reaches this threshold then at the next model time step GPP is constrained to limit the C:N ratio of newly sequestered biomass such that it doesn't exceed the max C:N ratio. We missed to mention this aspect of the model and will report the upper C:N ratio limits and details of this processes in the revised manuscript.**

In lines line 925-927 you state that with the increase in litter C:N ratios over the historical period litter decomposition rates decreases. Is this due to the increased immobilisation? How does the decrease in decomposition rates work and how much has it increased the soil C pools and where, as was stated in the same section?

**It seems these lines in the discussion section were misinterpreted. Making litter decomposition rates a function of litter C:N ratio was mentioned as a proposal for a future version of the model. This is not something that we have already done.**

You state that your litter plus soil C is in the right ballpark compared to Köchy et al. (2015), but your geographical distribution of C is way off if compared to Köchy et al. (2015). How does it come that you have a similar amount of soil C in tropical areas as in boreal regions? Like the forested boreal regions of Russia should have a much higher soil C content than e.g. the amazon. It seems that litter and soil C is very much

dictated by GPP (litter input) and not any soil processes (decomposition rates should be much slower in cold regions).

**Similar to several other models the current version of CLASSIC does not yet include permafrost related C cycle processes and peatlands. Work is underway to include both these processes to improve soil C estimates. Of course, the N cycle rides on top of the C cycle model so any limitations in the C cycle processes also appear in the model when C and N cycle processes are coupled. Comparing Figure 1 and Figure 4 of Köchy et al. (2015) the areas with high soil C amounts are also those which are identified with histosols soils which are rich in organic matter. In addition, the observation-based data sets of soil carbon show considerable mismatch amongst themselves and with ground truthing data as a recent Tifafi et al. (2018) paper shows. The largest difference between observation-based data sets is in boreal regions where differences are related to large disparities in soil organic carbon concentration. We will clarify this when revising our manuscript.**

Also your assumption of a constant soil C:N ratio (line 901-902) is not in line with Köchy et al. (2015) estimates.

**This statement is unclear since Köchy et al. (2015) paper doesn't mention nitrogen in any way. Regardless, we do mention in our manuscript that our assumption of constant C:N ratio (13 in our case) is the same as some other models and broadly consistent with Zhao et a. (2019) (for most land the values in their Figure 2h vary in the range 10-15, and hence the orange and yellow colours at high latitudes). But as already mentioned in the text, this remains as an assumption and we will discuss its implications in more detail including that it is this assumption of constant soil C:N ratio that yields the desired downregulation of photosynthesis in our model.**

In lines 908-910 you mention the idea that soil C:N ratio should be dependent on PFT,
I would say it should mostly be influenced by litter input C:N ratios.

**This statement was made in the context of using PFT dependent fixed soil C:N ratio rather than using the number 13 for all PFTs. In the same sentence we argued that a choice of a value different than 13 or had we chosen PFT-dependent values are of relatively less importance since the model is spun to equilibrium for 1850 conditions anyway. In a framework where soil C:N ratio is modelled dynamically and in the real world, of course, soil C:N depends on C:N ratio of litter inputs and immobilization.**

Including a BNF representation in an LSM is one of the trickiest parts. There is no real good way of doing it that I have seen in any LSM. Most LSMs use some kind of empirical relationship which you can parameterise to get global BNF number in the right region compared to estimates. So my personal opinion when it comes to modelling BNF is then to get the right BNF response to different kinds of manipulations. One being that BNF should decrease as you add N to the system according to the metaanalysis of Zheng et al. (2019). You mention on lines 898-900 that the BNF formulation will be revised. Make sure then to capture the right responses to changing conditions.

**Thank you for raising this point and we agree with your assessment of how BNF is modelled across models. Unfortunately, like several other processes in terrestrial ecosystem models, BNF remains parameterized. As the land component of the Canadian Earth System model, CLASSIC is continuously under development and when BNF parameterization is revised we will ensure that BNF decreases in response to N input and increases in response to increasing CO2 as meta-analyses suggest.**

Specific comments

L48-50: "Over land, the uptake of carbon in response to increasing anthropogenic CO2 emissions is driven by two primary factors, 1) the CO2 fertilization of the terrestrial biosphere, and 2) the increase in temperature, both of which are associated with increasing [CO2]." – How about water? How do you say that these two are the primary factors? What do you base it on? Later you write that photosynthesis can't occur without water or nutrients.

**This statement was made in the context of the current land C sink and not in context of photosynthesis in general. We, however, see how this can be confusing and will revise this part of the text accordingly.**

L68-72: "McGuire et al. (1995) define downregulation as a decrease in photosynthetic capacity of plants grown at elevated CO2 in comparison to plants grown at baseline CO2, although the rate of photosynthesis for plants grown and measured at elevated CO2 is still higher than the rate for plants grown and measured at baseline CO2." – This sentence doesn't make any sense, and you try and explain why in the following part of the section. But why have this at all. Why not give a proper definition of downregulation directly?

**Agreed. We can reword this sentence to define downregulation directly and then mention the McGuire et al. (1995) reference beside it.**

L81-83: "Comparison of land and ocean carbon uptake in C4MIP studies (Friedlingstein et al., 2006; Arora et al., 2013, 2019) indicate that the future land carbon uptake across ESMs varies widely and more than three times as much for the ocean carbon uptake." – Revise this sentence as it is unclear.

**Noted. We will revise this sentence.**

L162-164: "The biogeochemical module of CLASSIC uses this information along with air temperature to simulate photosynthesis and prognostically calculates amount of carbon in the model's three live (leaves, stem, and root) and two dead (litter and soil) carbon pools." - Do litter and SOM exist in each soil layer or only in one total pool each? And am I assuming right that each fractional vegetation cover (needleleaf trees,

broadleaf trees, crops, and grasses) has their own litter and soil C and N cycle? Not really clear in the manuscript.

**Individual PFTs present in each grid cell have their own prognostic litter and soil C pools and we will clarify this. These two pools, however, are not yet tracked per soil layer. This work is in progress.**

L211-213: "Finally, root biomass is used to calculate rooting depth and distribution which determines the fraction of roots in each soil layer" – How does the fine root distribution between soil layers affect the N uptake capacity?

**The model doesn't make the distinction between fine and coarse roots and only the total root biomass is simulated. Fraction of fine root biomass is, however, diagnostically calculated as a function of total root biomass using equation A5. For calculating N uptake only the fine root biomass is taken into consideration in equation A6 and not its distribution across the layers. We will clarify this when revising our manuscript.**

L232-259: The whole section on how the parameterizing of the downregulation of photosynthesis with increasing [CO2] for emulating nutrient constraints has been done is not needed as I guess it is described elsewhere.

**We will shorten this part of the text considerably but feel that it does provide the reader an idea of how downregulation is parameterized in the original model.**

Section 3.1: Whole section (Eqn 4-14) is more or less redundant as it is described in figure 2. Figure 1 is also redundant as it is present in figure 2. Better to have all eqn and description in the appendix here as they are the interesting part of the manuscript.

**Thank for you this suggestion that is also made by reviewer # 1. We will remove Figure 1, move budget equations to the appendix and description of primary processes to the main text as we reorganize our manuscript following both reviewers' comments.**

L421-423: "The modelled differences in PFT specific values of Vcmax, in our framework, come through differences in simulated $N_L$ values that depend on BNF, given that BNF is the primary natural source of N input into the coupled soil-vegetation system."- Over time BNF is the main source of N to the system, but for growth in a single year/day SOM mineralisation releases much more N (figure 3a vs 9a).

**Yes, the distinction here is in input versus cycling of N in the coupled plant and soil system. We will revise the text accordingly.**

L449-451: "The simulation for the pre-industrial period uses forcings that correspond to year 1850 and the model is run for thousands of years until its C and N pools come into equilibrium." -

**This question was also raised by reviewer #1. Global thresholds of net atmosphere-land C flux of 0.05 Pg/yr and net atmosphere-land N flux of 0.5 Tg N/yr are used to ensure the model pools have reached equilibrium. When modelled fluxes are less than these thresholds, model pools vary very little over time. We will mention these in the revised manuscript.**

How is the initial zero N world handled in the beginning of the spinup?

**The initial spin up in the zero N world was handled by turning off the plant N uptake and other organic processes, and using specified values of Vcmax, i.e. only the inorganic part of N cycle was operative. Once the inorganic soil N pools had built up the organic processes were turned on. We will mention this when revising our manuscript.**

Have you considered any approaches to speed up the spin-up? One approach could be to decouple the SOM pool module with constant saved litter input and soil conditions and run it for a standalone spin-up.

**We do have an accelerated spin up procedure where the sum of all inputs and outputs which modifies a pool is multiplied by a factor. This procedure is, how-**

ever, used only for the pools with the longest of turnover times such as soil organic matter and the inorganic N pools. We will mention this when revising our manuscript.

L490-492: "Finally, a separate pre-industrial simulation is also performed that uses the same $\Gamma_1$ and $\Gamma_2$ globally (FULL-no-implicit-P-limitation). This simulation is used to illustrate the effect of neglecting P limitation for the broadleaf evergreen tree PFT in the tropics." - Not only the tropics that is P limited (Du et al. 2020)

**Yes, this is true that there are other regions of the world where productivity is limited by phosphorus but, of course, the tropics are the regions where productivity is limited the most and we have addressed this by using a smaller value of $\Gamma_2$. We will clarify this when revising our manuscript. As we develop confidence in our model, in future we will include phosphorus cycle in our modelling framework as well.**

L621-623: "This results in a slight increase in vegetation and leaf N mass (Figures 6a and 6b) and the NH4+ (Figure 6e) pool which is the primary mineral pool in soils under vegetated regions." - How can it be that we have a total pool of NH4 and NO3 of around 4 Pg N (figure 6) at the end of the year and N demand of 1.5Pg N (figure 7). How can there be any limitations?

**There are two aspects to be considered here. First, is the mismatch between where the pools are high and where the vegetation grows and N is needed. For example, the NO3 pool is very high in the dry arid regions of the world where no vegetation grows. Second, even if there is sufficient N available in pools at a given location the rate of uptake determines how much N can be taken up. N demand has to be weighed against the rate of uptake not the size of available pools. We will clarify this subtlety when revising our manuscript.**

L635-638: "The increase in GPP due to changing climate increases the N demand (Figure 7a, magenta line) but unlike the CO2-only simulation, the plant N uptake increases since the NH4+ and NO3- pools increase in size over the vegetated area in response to increased BNF (Figure 3a, magenta line)." - BNF increases with 5.3 Tg N / yr whereas net N mineralisation increases with 35 Tg / yr. Why do you address the increase in plant N uptake to BNF and not increases in net mineralisation?

**Thank you for noticing this. Yes, mineralization also contributes to increased plant N uptake in the simulation with changing climate. We will reword this sentence accordingly.**

L644-646: "The litter C:N, in contrast, shows a small increase since not all N makes its way to the litter as a fraction of leaf N is resorbed from deciduous trees leaves prior to leaf fall (Figure 8e)." - This needs a better explanation. Is the resorption fraction dynamic or is the deciduous PFT more dominant (competition between PFTs turned off, sadly)? Don't understand how this can happen.

**The cold deciduous broadleaf and needleleaf PFT covers a substantial fraction of the vegetation area (around 16 out of 84 million km2 in our land cover data, so around 19%). The resorption fraction is not dynamic but specified as 0.54 (Table A1) for deciduous PFTs so a lot of N gets resorbed. The result is that although the leaf C:N ratio decreases in the CLIM only simulation, in response to increased BNF and increased mineralization, the C:N ratio of litter increases, since the decrease in C:N ratio of leaves is not large enough to overcome the effect of resorption. We will clarify this when revising our manuscript.**

L676-679: "An increase in crop area over the historical period results in deforestation of natural vegetation that reduces vegetation biomass but also soil carbon mass, since a higher soil decomposition rate is assumed over cropland area (Figures 5b and 5d), consistent with empirical measurements (Wei et al., 2014)." - How is this modelled? Needs an explanation. How do fig 5b and 5d show that the decomposition rate has increased? Litter input has maybe decreased but decomposition rates changed?

**The increased soil decomposition rate over croplands is modelled via a higher**

**soil carbon decomposition rate to account for tillage. In Figure 5a the GPP in the simulation does not change substantially from its pre-industrial value so a decrease in GPP and thus NPP is not the cause of decrease seen in soil carbon in the LUC+FERT only simulation (Fig. 5d). Vegetation carbon in the LUC+FERT only simulation (Fig. 5b) decreases because of deforestation associated with land use. Despite this increased litter input into soils associated with deforestation, soil carbon decreases because of increased soil carbon decomposition rates as cropland area expands. This is by design to simulate the effects of land use and cover change.**

L714-717: "The increase in global NH4+ mass (Figure 6e) in the FULL simulation is driven primarily by the increase in fertilizer input while the changes in NO3- mass are the net result of all forcings with no single forcing dominating the response." – This needs to be different for Crop and Natural tiles. Needs some discussion.

**Yes, this is indeed different for fractions of grid cell covered with cropland and natural vegetation and this insight is gained by comparing LUC+FERT only simulation (green line) with the FULL simulation (red line) in Figures 6e and 6f. This was indeed the purpose of doing simulations perturbed with only selected forcings at a time to be able to understand the model response. We will clarify this when revising our manuscript.**

L720-723: "The increase in the C:N ratio of vegetation (Figure 8a) and its components (leaves, stem, and root) is driven primarily by an increase in atmospheric CO2. Changes in litter C:N in the FULL simulation, in contrast, do not experience dominant influence from any one of the forcings." - This needs some kind of explanation. How can this happen? Same as for L644-646.

**We agree that just like lines 644-646 this needs additional explanation. The reason that litter C:N doesn't show large changes here can be inferred from Figure 8e. While the litter C:N does indeed increase in the CO2-only simulation as**

**would be intuitively expected, in the full simulation it doesn't increase as much because it is compensated by the decrease over croplands (see green line in Figure 8e) due to increased fertilizer input. We will clarify this when revising our manuscript.**

L723-726: "The simulated change in net N mineralization (Figure 9a) in the FULL simulation, over the historical period, is small since the decrease in net N mineralization due to increasing CO2 is compensated by the increase caused by changes in climate, N deposition, and fertilizer inputs." - Has to be a large difference between Natural and Crop tiles. Needs to be shown.

**This is actually shown using our different simulations. The purpose of doing LUC+FERT-only simulation is indeed to highlight the changes over crop area. Figure 9a clearly shows that mineralization increases in CLIM-only, NDEP-only, and LUC+FERT-only simulations and decreases in CO2-only simulation. The net result of all these forcings is that in the FULL simulation mineralization doesn't change substantially compared to its pre-industrial value.**

L844: "The response of our model to elevated CO2" - Very hard to draw any conclusion in respect to elevated [CO2] from your experiments. Elevated [CO2] is usually referred to as a CO2 concentration that is higher than the present.

**We assume you are referring to the subtlety between the "historical increase in CO2" which is gradual and "elevated CO2" in FACE experiments in which CO2 is instantaneously elevated. We will reword this sentence.**

L865-867: "Soil carbon mass, however, decreases (despite increase in NPP inputs) since warmer temperatures also increase heterotrophic respiration (not shown)." – I wouldn't call a decrease of 0.4 Pg compared to a total pool of 1074 Pg a decrease. To say that you get a realistic response.

**Small misinterpretation here. You looked at the results from the full simulation**

**in Figure 5d to infer the changes in soil and litter carbon pool in response to change in climate. Whereas this sentence refers to the CLIM-only simulation in which soil carbon decreases by 6.8 Pg C. We will clarify this when revising our manuscript. While not a large decrease (since NPP inputs also increase somewhat in the CLIM-only simulation) the change is of the right sign which is what we were implying. We will reword this sentence.**

L986-990: "For simplicity, we assume fertilizer is applied at the same daily fertilizer application rate (gN m–2 day–1) throughout the year in the tropics (between 30 S and 30 N), given the possibility of multiple crop rotations in a given year. Between the 30 and 90 latitudes in both northern and southern hemispheres, we assume that fertilizer application starts on the spring equinox and ends on the fall equinox." - Using hard limits in a model like this will create problems. Wouldn't it be possible to do this a little more prognostic? Base it on PFT distribution and productivity? Changing climate? Having a fixed limit like this won't adapt to a changing climate.

**Yes, of course, it is possible to make fertilizer application rules more realistic by making them country specific or developing a parameterization that would apply fertilizer as a function of climate. However, the problem here is that whatever parameterization one comes up with, care has to be taken that the annual amount of fertilizer that is read from the input file for a given year is indeed applied. Fixed application dates make it easier ahead of time to figure out how much fertilizer is to be applied each day. A parameterization with prognostic start and end days will have more technical difficultly because it is not known ahead of time when the fertilizer application will end. Since the purpose of this first attempt at N cycle in CLASSIC is to ensure that correct amount of fertilizer, as read from input files, is indeed applied a simple approach is more robust.**

L1000-1001: "The modelled plant N uptake is a function of its N demand. Higher N demand leads to higher mineral N uptake from soil." - So it is independent on the amount of roots?

**Some misinterpretation again here. There are two types of plant N uptakes. Passive N uptake depends on inorganic N amounts in the soil and plant water uptake through transpiration. In our model, plants have no control on this N uptake. If plant N demand cannot be met via passive N uptake, active N uptake kicks in and this depends on fine root biomass (Section A2.3). So in summary, plant N uptake depends on the demand but its active component depends on fine root biomass. We will ensure to clarify this even more when revising our manuscript.**

Section A2.6: What are the decay rate constants for litter and soil pools? Have they changed after the inclusion of the N cycle?

**The decay rates of litter and soil C are described in the C cycle part of CLASSIC (Melton and Arora, 2016) and as mentioned in the discussion section, they are yet to be made as a function of C:N ratio of litter and soil C (though the soil organic matter's C:N is fixed anyway). We will make this even more clear.**

Section A3.3: Shouldn't the amount of NO3 in the bottom layer be estimated in a similar manner as for nitrification/denitrification/volatilization? Denitrification assumes that all NO3 is in the top 50cm.

**It seems the reviewer is referring to nitrogen leaching (since he refers to Section A3.3). While the model uses the soil temperature and moisture from the top 50 cm soil layer for nitrification and denitrification it still uses the entire pool size. For leaching, it would be unrealistic to assume a distribution of NO3 in soil and use NO3 only from the bottom layer for leaching. As water enters from the top and leaves at the bottom of the permeable soil layer all NO3 is available for leaching.**

Figure 5d: How is the split between litter and soil C? Figure A.1: Explain what the two lines represent

**We will mention the split between litter and soil carbon when revising our**

**manuscript. Typically, litter amount is around 80 Pg C out of the total 1100 Pg C of soil C and litter combined. As Figure A1 caption mentions, the thin lines are the annual values and the thick lines are their 10 year moving averages.**

Table A1: Header text is unclear

**Thank you. We will reword this text.**

Technical corrections

L511: "FIRE-only" – remove. L779: remove "two"

**Thank you for catching these.**

**References**

Köchy, M., Hiederer, R. and Freibauer, A.: Global distribution of soil organic carbon – Part 1: Masses and frequency distributions of SOC stocks for the tropics, permafrost regions, wetlands, and the world, SOIL, 1(1), 351–365, doi:10.5194/soil-1-351-2015, 2015.

McGuire, A. D., Melillo, J. M. and Joyce, L. A.: The role of nitrogen in the response of forest net rimary production to elevated atmospheric carbon dioxide, Annu. Rev. Ecol. Syst., 26(1), 473–503, doi:10.1146/annurev.es.26.110195.002353, 1995.

Melton, J. R. and Arora, V. K.: Competition between plant functional types in the Canadian Terrestrial Ecosystem Model (CTEM) v. 2.0, Geosci Model Dev, 9(1), 323–361, doi:10.5194/gmd-9-323-2016, 2016.

Tifafi, M., Guenet, B., Hatté, C. (2018). Large differences in global and regional total soil carbon stock estimates based on SoilGrids, HWSD, and NCSCD: Intercomparison and evaluation based on field data from USA, England, Wales, and France. Global Biogeochemical Cycles, 32, 42–56. https://doi.org/10.1002/2017GB005678

Zhao, X., Yang, Y., Shen, H., Geng, X. and Fang, J.: Global soil–climate–biome di-

agram: linking surface soil properties to climate and biota, Biogeosciences, 16(14), 2857–2871, doi:10.5194/bg-16-2857-2019, 2019.

[Figure]

GPP (1998-2017)

Simulation name
FULL

Global total = 120 Pg C/yr

50  100  200  300  500  1000 1500 2000 2500 3300
g C m$^{-2}$ year$^{-1}$

GPP (1998-2017)

Simulation name
ORIGINAL

Global total = 120 Pg C/yr

50  100  200  300  500  1000 1500 2000 2500 3300
g C m$^{-2}$ year$^{-1}$

GPP (1998-2017)

Simulation name
ORIG-UNCONST

Global total = 142 Pg C/yr

50  100  200  300  500  1000 1500 2000 2500 3300
g C m$^{-2}$ year$^{-1}$

**Fig. 1.**

[Figure]

[Figure]

---

## Author Response (AR1)

Dear handling editor,

Thanks for giving us the opportunity to revise our manuscript following reviewers' comments. We have taken into account all of reviewers' comments in revising our manuscript.

Among their major comments,

1) we have rearranged the manuscript by moving budget equations to the appendix, and moving the description of all parameterizations to the main text,
2) moved Figure 1, which showed the original structure of the model, from main text to the appendix
3) added additional discussion of various BNF parameterizations, and parameterizations that related leaf N to Vcmax, in the discussion section
4) added equations that describe how the model treats upper limit of C:N ratios of different plant components,
5) included an additional plot (now Figure 10c) which shows zonally-averaged GPP ratios to illustrate how downregulation works differently in the original model and the model version with N cycle, and
6) included a sentence about prioritizing immobilization versus plant N uptake in section 3.3.6 (on page 27 of this document).
7) Reorganized figures such that fertilizer input and N deposition are considered inputs, plotted in a separate figure, and discussed in inputs while BNF is considered a model output, plotted separately than specified N inputs of fertilizer and N deposition, and discussed in the model results section.

In addition, we also took into account all of minor comments that reviewers made as mentioned in our reply to the reviewers in the discussion section.

This document tracks all the changes that we have made since our last version.

Best regards,
Vivek.

[revised manuscript text omitted]

**Global precipitation over land**
**(excluding Greenland and Antarctica)**

[Figure]

**Air temperature over land**
**(excluding Greenland and Antarctica)**

[Figure]

Figure A2: Annual values of global precipitation (a) and air temperature (b) over land in the
CRU-JRA reanalysis data that are used to drive the model. The data are available for the period
1901-2017. In the absence of meterological data, the period 1851-1900 uses the data from the
period 1901-1925 twice. The thin lines are the annual values and the thick line their 10 year
running mean.

**Table A1**: Model parameters for various model parameterizations. Corresponding equation in which the
parameter appears in the main text is also noted. Model parameters may be scalar or an array (if they
are PFT dependent) in which case they are written according to the following structure in the table
below.

| Needleleaf evergreen | Needleleaf deciduous | |
|---|---|---|
| Broadleaf evergreen | Broadleaf deciduous cold | Broadleaf deciduous drought |
| C$_3$ crop | C$_4$ crop | |
| C$_3$ grass | C$_4$ grass | |

| Model parameter | Eqn | Description | Units | Value(s) | | |
|---|---|---|---|---|---|---|
| *Biological N fixation* | | | | | | |
| $\alpha_c$ | 3 | BNF rate for crop PFTs | gN m$^{-2}$ day$^{-1}$ | 0.00217 | | |
| $\alpha_n$ | 3 | BNF rate for natural PFTs | gN m$^{-2}$ day$^{-1}$ | 0.00037 | | |
| *Plant N demand* | | | | | | |
| $C{:}N_{L,min}$ | 4 | Minimum C:N ratio for leaves | dimensionless | 25
| 22
| 18 |
| $C{:}N_{S,min}$ | 4 | Minimum C:N ratio for stem | dimensionless | 450
− | 450
− | 430 |
| $C{:}N_{R,min}$ | 4 | Minimum C:N ratio for root | dimensionless | 45
| 45
| 35 |
| *Plant uptake* | | | | | | |
| $\beta$ | 6 | Mineral N distribution coefficient | dimensionless | 0.5 | | |
| $\varepsilon$ | 8 | Fine root efficiency | gN gC$^{-1}$ day$^{-1}$ | 4.92E-5 | | |
| $k_{p,\frac{1}{2}}$ | 8 | Half saturation constant | gN m$^{-3}$ | 3 | | |
| *Litterfall* | | | | | | |
| $r_L$ | 11 | Leaf resorption coefficient | dimensionless | 0.54 | | |
| *Nitrification* | | | | | | |
| $\eta$ | 19 | Nitrification coefficient | day$^{-1}$ | 7.33E-4 | | |

¶

| | | | | | | |
|---|---|---|---|---|---|---|
| $\eta_{NO}$ | 23 | Fraction of nitrification flux emitted as NO | dimensionless | 7.03E-5 | | |
| $\eta_{N2O}$ | 23 | Fraction of nitrification flux emitted as N$_2$O | dimensionless | 2.57E-5 | | |
| *Denitrification* | | | | | | |
| $\mu_{NO}$ | 24 | Fraction of denitrification flux emitted as NO | day$^{-1}$ | 3.872E-4 | | |
| $\mu_{N2O}$ | 24 | Fraction of denitrification flux emitted as N$_2$O | day$^{-1}$ | 1.408E-4 | | |
| $\mu_{N2}$ | 24 | Fraction of denitrification flux emitted as N$_2$ | day$^{-1}$ | 3.872E-3 | | |
| $w_d$ | 24 | Soil wetness threshold below which very little denitrification occurs | dimensionless | 0.3 | | |
| *Leaching* | | | | | | |
| $\varphi$ | 26 | Leaching coefficient | m$^2$ Kg$^{-1}$ | 1.15E-3 | | |
| *NH$_3$ volatilization* | | | | | | |
| $\vartheta$ | 27 | NH$_3$ volatilization coefficient | dimensionless | 0.54 | | |
| *Coupling of C and N cycles* | | | | | | |
| $\Gamma_1$ | 31 | Parameter for calculating V$_{cmax}$ from leaf N content | µmol CO$_2$ gN$^{-1}$ s$^{-1}$ | 13 (all PFTs except broadleaf evergreen tree) 5.1 (for broadleaf evergreen tree) | | |
| $\Gamma_2$ | 31 | Parameter for calculating V$_{cmax}$ from leaf N content | µmol CO$_2$ m$^{-2}$ s$^{-1}$ | 8.5 | | |
| $k_\Lambda$ | 32 | Parameter for constraining V$_{cmax}$ increase when C:N ratios exceed their maximum limit | dimensionless | 0.05 | | |
| $C{:}N_{L,max}$ | 33 | Maximum C:N ratio for leaves | dimensionless | 60 55 40 35 | 50 40 50 50 | 40 |
| $C{:}N_{S,max}$ | 33 | Maximum C:N ratio for stem | dimensionless | 800 670 500 | 800 670 500 | 670 |

| | | | | $=$ | $=$ | |
|---|---|---|---|---|---|---|
| $C\!:\!N_{R,max}$ | 33 | Maximum C:N ratio for root | dimensionless | 90
| 90
| 70 |

**Table 1**: Historical simulations performed over the period 1851-2017 to evaluate the model's
response to various forcings. All forcings are time varying. All forcings are also spatially explicit
except atmospheric $CO_2$ for which a globally constant value is specified.

| Simulation name | Forcing that varies over the historical period | N cycle |
|---|---|---|
| *Primary simulations performed to evaluate N cycle response to various forcings* | | |
| 1. CO2-only | Atmospheric $CO_2$ concentration | Runs with N cycle |
| 2. CLIM-only | 1901-1925 meteorological data are used twice over the 1850-1900 period. For the 1901-2017 period, meteorological data for the correct year is used. | |
| 3. LUC+FERT-only | Land cover with increasing crop area, and fertilizer application rates over the crop area | |
| 4. N-DEP-only | N deposition of ammonia and nitrate | |
| 5. FULL | All forcings | |
| 6. FULL-no-LUC | All forcings except increasing crop area | |

| | | |
|---|---|---|
| *Other simulations* | | |
| 7. ORIGINAL | All forcings | Runs without N cycle using the original model configuration. |
| 8. ORIG-UNCONST | All forcings but with downregulation turned off | |
| 9. FULL-no-implicit-P-limitation | All forcings but using same $\Gamma_1$ and $\Gamma_2$ globally | Run with N cycle |

**Table 2**: Comparison of simulated global N pools and fluxes, from the FULL simulation, with other
modelling and quasi observation-based studies (references for which are noted as superscripts
and listed below the table). The time-periods to which the other modelling and quasi
observation-based estimates correspond are also noted, where available. The estimates are for
land. Simulated fluxes and pool corresponds to the period 1997-2018.

| N pool and fluxes | This study (1998-2017) | Other model and quasi observation-based estimates |
|---|---|---|
| *N inputs* (Tg N yr⁻¹) | | |
| BNF | 119 | 118[a]
99[b] (2001-2010)
138.5[c] (early 1990s)
128.9[d] (2000-2009)
104-118[e]
92[f] (year 2000) |
| Natural BNF | 59 | 58[a]
107[c] (early 1990s)
30-130[e]
39[f] (year 2000) |
| Anthropogenic BNF | 60 | 60[a]
31.5[c] (early 1990s)
14-89[e]

[revised manuscript text omitted]

---

## Author Response (AR2)

Dear handling editor,

Thank you for giving us the opportunity to include technical corrections suggested by Reviewer # 2 following the second round of reviews for our manuscript. We have included all of his/her suggestions.

In particular we have clarified the units of leaf N amount and consistently used the word "amount" instead of the word "content" through the manuscript. These changes make the units clear and consistent in all equations.

Thank you,

Vivek Arora.

[revised manuscript text omitted]